# Adaptive Discretization for Model-Based Reinforcement Learning

**Sean R. Sinclair**
Cornell University
srs429@cornell.edu

**Tianyu Wang**
Duke University
tianyu@cs.duke.edu

**Gauri Jain**
Cornell University
gauri.g.jain@gmail.com

**Siddhartha Banerjee**
Cornell University
sbanerjee@cornell.edu

**Christina Lee Yu**
Cornell University
cleeyu@cornell.edu

## Abstract

We introduce the technique of adaptive discretization to design an efficient model-based episodic reinforcement learning algorithm in large (potentially continuous) state-action spaces. Our algorithm is based on optimistic one-step value iteration extended to maintain an adaptive discretization of the space. From a theoretical perspective we provide worst-case regret bounds for our algorithm which are competitive compared to the state-of-the-art model-based algorithms. Moreover, our bounds are obtained via a modular proof technique which can potentially extend to incorporate additional structure on the problem.

From an implementation standpoint, our algorithm has much lower storage and computational requirements due to maintaining a more efficient partition of the state and action spaces. We illustrate this via experiments on several canonical control problems, which shows that our algorithm empirically performs significantly better than fixed discretization in terms of both faster convergence and lower memory usage. Interestingly, we observe empirically that while fixed discretization model-based algorithms vastly outperform their model-free counterparts, the two achieve comparable performance with adaptive discretization. [1]

## 1 Introduction

Reinforcement learning (RL) is a paradigm modeling an agent's interactions with an unknown environment with the goal of maximizing their cumulative reward throughout the trajectory [43]. In online settings the dynamics of the system are unknown and the agent must learn the optimal policy only through interacting with the environment. This requires the agent to navigate the *exploration exploitation trade-off*, between exploring unseen parts of the system and exploiting historical high-reward decisions. Most algorithms for learning the optimal policy in these online settings can be classified as either *model-free* or *model-based*. Model-free algorithms construct estimates for the $Q$-function of the optimal policy, the expected sum of rewards obtained from playing a specific action and following the optimal policy thereafter, and create upper-confidence bounds on this quantity [38, 16]. In contrast, model-based algorithms instead estimate unknown system parameters, namely the average reward function and the dynamics of the system, and use this to learn the optimal policy based on full or one-step planning [5, 13].

| Algorithm | Regret | Time Complexity | Space Complexity |
|---|---|---|---|
| ADAMB (Alg. 1) ($d_{\mathcal{S}} > 2$) | $H^{1+\frac{1}{d+1}}K^{1-\frac{1}{d+d_{\mathcal{S}}}}$ | $HK^{1+\frac{d_{\mathcal{S}}}{d+d_{\mathcal{S}}}}$ | $HK$ |
| ($d_{\mathcal{S}} \leq 2$) | $H^{1+\frac{1}{d+1}}K^{1-\frac{1}{d+d_{\mathcal{S}}+2}}$ | $HK^{1+\frac{d_{\mathcal{S}}}{d+d_{\mathcal{S}}+2}}$ | $HK^{1-\frac{2}{d+d_{\mathcal{S}}+2}}$ |
| ADAPTIVE Q-LEARNING [38] | $H^{5/2}K^{1-\frac{1}{d+2}}$ | $HK\log_d(K)$ | $HK^{1-\frac{2}{d+2}}$ |
| KERNEL UCBVI [11] | $H^3\ \ K^{1-\frac{1}{2d+1}}$ | $HAK^2$ | $HK$ |
| NET-BASED $Q$-LEARNING [42] | $H^{5/2}K^{1-\frac{1}{d+2}}$ | $HK^2$ | $HK$ |
| LOWER-BOUNDS [40] | $H\ \ K^{1-\frac{1}{d+2}}$ | N/A | N/A |

Table 1: *Comparison of our bounds with several state-of-the-art bounds for RL in continuous settings. Here, $d$ is the covering dimension of the state-action space, $d_{\mathcal{S}}$ is the covering dimension of the state space, $H$ is the horizon of the MDP, and $K$ is the total number of episodes. Implementing KERNEL UCBVI [11] is unclear under general action spaces, so we specialize the time complexity under a finite set of actions of size $A$. As running UCBVI with a fixed discretization is a natural approach to this problem, we include a short discussion of this algorithm in Appendix G.1. Since the results are informal, we do not include them in the table here. We include 'N/A' under the time and space complexity lower bound as there is no prior work in this domain to our knowledge.*

RL has received a lot of interest in the design of algorithms for large-scale systems using parametric models and function approximation. For example, the AlphaGo Zero algorithm that mastered Chess and Go from scratch trained their algorithm over 72 hours using 4 TPUs and 64 GPUs [36]. These results show the intrinsic power of RL in learning complex control policies, but are computationally infeasible for applying algorithms to RL tasks in computing systems or operations research. The limiting factor is implementing regression oracles or gradient steps on computing hardware. For example, RL approaches have received much interest in designing controllers for memory systems [1] or resource allocation in cloud-based computing [15]. Common to these examples are computation and storage limitations on the devices used for the controller, requiring algorithms to compete on three major facets: efficient learning, low computation, and low storage requirements.

Motivated by these requirements we consider discretization techniques which map the continuous problem to a discrete one as these algorithms are based on simple primitives easy to implement in hardware (and has been tested heuristically in practice [32, 22]). A challenge is picking a discretization to manage the trade-off between the discretization error and the errors accumulated from solving the discrete problem. As a fixed discretization wastes computation and memory by forcing the algorithm to explore unnecessary parts of the space, we develop an adaptive discretization of the space, where the discretization is only refined on an *as-needed* basis. This approach reduces unnecessary exploration, computation, and memory by only keeping a fine-discretization across important parts of the space [38].

Adaptive discretization techniques have been successfully applied to multi-armed bandits [40] and model-free RL [38]. The key idea is to maintain a non-uniform partition of the space which is refined based on the density of samples. These techniques do not, however, directly extend to model-based RL, where the main additional ingredient lies in maintaining transition probability estimates and incorporating these in decision-making. Doing so is easy in tabular RL and $\epsilon$-net based policies, as simple transition counts concentrate well enough to get good regret. This is much less straightforward, though, when the underlying discretization changes in an online, data-dependent way.

**Our Contributions**. We design and analyze a *model-based* RL algorithm, ADAMB, that discretizes the state-action space in a data-driven way so as to minimize regret. ADAMB requires the underlying state and action spaces to be embedded in compact metric spaces, and the reward function and transition kernel to be Lipschitz continuous with respect to this metric. This encompasses discrete and continuous state-action spaces with mild assumptions on the transition kernel, and deterministic systems with Lipschitz continuous transitions. Our algorithm only requires access to the metric, unlike prior algorithms which require access to simulation oracles [18], strong parametric assumptions [17], or impose additional assumptions on the action space to be computationally efficient [11].

Our policy achieves near-optimal dependence of the regret on the covering dimension of the metric space when compared to other model-based algorithms. In particular, we show that for a $H$-step

MDP played over $K$ episodes, our algorithm achieves a regret bound

$$R(K) \lesssim \begin{cases} H^{1+\frac{1}{d+1}} K^{\frac{d+d_S-1}{d+d_S}} & d_S > 2 \\ H^{1+\frac{1}{d+1}} K^{\frac{d+d_S+1}{d+d_S+2}} & d_S \leq 2 \end{cases}$$

where $d_S$ and $d_A$ are the covering dimensions of the state and action space respectively, and $d = d_S + d_A$. As Table 1 illustrates, our bounds are uniformly better (in terms of dependence on $K$ and $H$, in all dimensions) than the best existing bounds for model-based RL in continuous-spaces [21, 11]. In addition to having lower regret, ADAMB is also simple and practical to implement, with low query complexity and storage requirements (see Table 1) compared to other model-based techniques.

To highlight this, we complement our theory with experiments comparing model-free and model-based algorithms, using both fixed and adaptive discretization. Our experiments show that with a fixed discretization, model-based algorithms outperform model-free ones; however, when using an adaptive partition of the space, model-based and model-free algorithms perform similarly. This provides an interesting contrast between practice (where model-based algorithms are thought to perform much better) and theory (where regret bounds in continuous settings are currently worse for model-based compared to model-free algorithms), and suggests more investigation is required for ranking the two approaches.

**Related Work**. There is an extensive literature on model-based reinforcement learning; below, we highlight the work which is closest to ours, but for more extensive references, see [43] for RL, and [6, 41] for bandits (and a longer discussion in Appendix B).

Recently there has been a surge in theoretical analysis of nonparametric algorithms for RL in continuous spaces. These algorithms all require mild local assumptions on the underlying process, most commonly that the $Q$ function is Lipschitz continuous with respect to a given metric. For example, some work considers nearest-neighbour methods for deterministic, and infinite horizon discounted settings [48, 34]. Other work assumes access to a generative model instead of the online setting considered here [18, 14]. The work closest to ours concerns algorithms with provable guarantees for continuous state-action settings (see also Table 1). In model-free settings, tabular algorithms have been adapted to continuous state-action spaces via fixed discretization (i.e., $\epsilon$-nets) [42]. In model-based settings, researchers have tackled continuous spaces via kernel methods, based on either a fixed discretization of the space [21], or more recently, without resorting to discretization [11]. While the latter does learn a data-driven representation of the space via kernels, it requires solving a complex optimization problem at each step, and hence is efficient mainly for finite action sets (more discussion on this is in Section 4). Finally, adaptive discretization has been successfully implemented in model-free settings [38, 7], and this provides a good benchmark for our algorithm, and for comparing model-free and model-based algorithms.

## 2   Preliminaries

**MDP and Policies**. We consider an agent interacting with an underlying finite-horizon Markov Decision Processes (MDP) over $K$ sequential episodes, denoted $[K] = \{1, \ldots, K\}$. The underlying MDP is given by a five-tuple $(\mathcal{S}, \mathcal{A}, H, T, R)$ where horizon $H$ is the number of steps (indexed $[H] = \{1, 2, \ldots, H\}$) in each episode, and $(\mathcal{S}, \mathcal{A})$ denotes the set of states and actions in each step. When needed for exposition, we use $\mathcal{S}_h, \mathcal{A}_h$ to explicitly denote state/action sets at step $h$. When the step $h$ is clear we omit the subscript for readability.

Let $\Delta(\mathcal{X})$ denote the set of probability measures over a set $\mathcal{X}$. State transitions are governed by a collection of transition kernels $T = \{T_h(\cdot \mid x, a)\}_{h \in [H], x \in \mathcal{S}, a \in \mathcal{A}}$, where $T_h(\cdot \mid x, a) \in \Delta(\mathcal{S}_{h+1})$ gives the distribution over states in $\mathcal{S}_{h+1}$ if action $a$ is taken in state $x$ at step $h$. The instantaneous rewards are bounded in $[0, 1]$, and their distributions are specified by a collection of parameterized distributions $R = \{R_h\}_{h \in [H]}$, $R_h : \mathcal{S}_h \times \mathcal{A}_h \to \Delta([0, 1])$. We denote $r_h(x, a) = \mathbb{E}_{r \sim R_h(x,a)}[r]$.

A policy $\pi$ is a sequence of functions $\{\pi_h \mid h \in [H]\}$ where each $\pi_h : \mathcal{S}_h \to \mathcal{A}_h$ is a mapping from a given state $x \in \mathcal{S}_h$ to an action $a \in \mathcal{A}_h$. At the beginning of each episode $k$, the agent fixes a policy $\pi^k$ for the entire episode, and is given an initial (arbitrary) state $X_1^k \in \mathcal{S}_1$. In each step $h \in [H]$, the agent receives the state $X_h^k$, picks an action $A_h^k = \pi_h^k(X_h^k)$, receives reward $R_h^k \sim R_h(X_h^k, A_h^k)$, and transitions to a random state $X_{h+1}^k \sim T_h(\cdot \mid X_h^k, \pi_h^k(X_h^k))$. This continues until the final transition

to state $X_{H+1}^k$, at which point the agent chooses policy $\pi^{k+1}$ for the next episode after incorporating observed rewards and transitions in episode $k$, and the process is repeated.

**Value Function and Bellman Equations**. For any policy $\pi$, let $A_h^\pi$ denote the (random) action taken in step $h$ under $\pi$, i.e., $A_h^\pi = \pi_h(X_h^k)$. We define $V_h^\pi : \mathcal{S} \to \mathbb{R}$ to denote the *policy value function* at step $h$ under policy $\pi$, i.e., the expected sum of future rewards under policy $\pi$ starting from $X_h = x$ in step $h$ until the end of the episode. Formally,

$$V_h^\pi(x) := \mathbb{E}\Big[\sum_{h'=h}^H R_{h'} \mid X_h = x\Big] \quad \text{for} \quad R_{h'} \sim R_h(X_{h'}, A_{h'}^\pi).$$

We define the state-action value function (or $Q$-function) $Q_h^\pi : \mathcal{S} \times \mathcal{A} \to \mathbb{R}$ at step $h$ as the sum of the expected rewards received after taking action $A_h = a$ at step $h$ from state $X_h = x$, and then following policy $\pi$ in all subsequent steps of the episode. Formally,

$$Q_h^\pi(x, a) := r_h(x, a) + \mathbb{E}\Big[\sum_{h'=h+1}^H R_{h'} \mid X_{h+1} \sim T_h(\cdot \mid x, a)\Big] \quad \text{for} \quad R_{h'} \sim R_{h'}(X_{h'}, A_{h'}^\pi).$$

Under suitable assumptions on $\mathcal{S} \times \mathcal{A}$ and reward functions [31], there exists an optimal policy $\pi^\star$ which gives the optimal value $V_h^\star(x) = \sup_\pi V_h^\pi(x)$ for all $x \in \mathcal{S}$ and $h \in [H]$. For ease of notation we denote $Q^\star = Q^{\pi^\star}$. The Bellman equations [31] state that,

$$\begin{aligned}
V_h^\pi(x) &= Q_h^\pi(x, \pi_h(x)) & \forall\, x \in \mathcal{S} \\
Q_h^\pi(x, a) &= r_h(x, a) + \mathbb{E}\big[V_{h+1}^\pi(X_{h+1}) \mid X_h = x, A_h = a\big] & \forall\, (x, a) \in \mathcal{S} \times \mathcal{A} \quad (1) \\
V_{H+1}^\pi(x) &= 0 & \forall\, x \in \mathcal{S}.
\end{aligned}$$

For the optimal policy $\pi^\star$, it additionally holds that $V_h^\star(x) = \max_{a \in \mathcal{A}} Q_h^\star(x, a)$.

In each episode $k \in [K]$ the agent selects a policy $\pi^k$, and is given an arbitrary starting state $X_1^k$. The goal is to maximize the total expected reward $\sum_{k=1}^K V_1^{\pi^k}(X_1^k)$. We benchmark the agent on their regret: the additive loss over all episodes the agent experiences using their policy instead of the optimal one. In particular, the *regret $R(K)$* is defined as:

$$R(K) = \sum_{k=1}^K \Big(V_1^\star(X_1^k) - V_1^{\pi^k}(X_1^k)\Big). \quad (2)$$

Our goal is to show that the regret $R(K)$ is sublinear with respect to $K$.

**Metric Space and Lipschitz Assumptions**. We assume the state space $\mathcal{S}$ and the action space $\mathcal{A}$ are each separable compact metric spaces, with metrics $\mathcal{D}_\mathcal{S}$ and $\mathcal{D}_\mathcal{A}$, and covering dimensions $d_\mathcal{S}$ and $d_\mathcal{A}$ respectively. This imposes a metric structure $\mathcal{D}$ on $\mathcal{S} \times \mathcal{A}$ via the product metric, or any sub-additive metric such that

$$\mathcal{D}((x, a), (x', a')) \le \mathcal{D}_\mathcal{S}(x, x') + \mathcal{D}_\mathcal{A}(a, a').$$

This also ensures that the covering dimension of $\mathcal{S} \times \mathcal{A}$ is at most $d = d_\mathcal{S} + d_\mathcal{A}$. We assume that the algorithm has oracle access to the metrics $\mathcal{D}_\mathcal{S}$ and $\mathcal{D}_\mathcal{A}$ through several queries, which are explained in more detail in Appendix G.1. We also need that $T_h(\cdot \mid x, a)$ is Borel with respect to the metric $\mathcal{D}_\mathcal{S}$ for any $(x, a) \in \mathcal{S} \times \mathcal{A}$.

We assume w.l.o.g. that $\mathcal{S} \times \mathcal{A}$ has diameter 1, and we denote the diameter of $\mathcal{S}$ as $\mathcal{D}(\mathcal{S}) = \sup_{a \in \mathcal{A}, (x,y) \in \mathcal{S}^2} \mathcal{D}((x, a), (y, a)) \le 1$. For more information on metrics and covering dimension, see [40, 20, 38] for a summary.

To motivate the discretization approach, we also assume non-parametric Lipschitz structure on the transitions and rewards of the underlying process [38].

**Assumption 1** (Lipschitz Rewards and Transitions). *For every $x, x', h \in \mathcal{S} \times \mathcal{S} \times [H]$ and $a, a' \in \mathcal{A} \times \mathcal{A}$, the average reward function $r_h(x, a)$ is Lipschitz continuous with respect to $\mathcal{D}$, i.e.:*

$$|r_h(x, a) - r_h(x', a')| \le L_r \mathcal{D}((x, a), (x', a'))$$

*For every $(x, x', h) \in \mathcal{S} \times \mathcal{S} \times [H]$ and $(a, a') \in \mathcal{A} \times \mathcal{A}$, the transition kernels $T_h(x' \mid x, a)$ are Lipschitz continuous in the 1-Wasserstein metric $d_W$ with respect to $\mathcal{D}$, i.e.:*

$$d_W(T_h(\cdot \mid x, a), T_h(\cdot \mid x', a')) \le L_T \mathcal{D}((x, a), (x', a')).$$

*We further assume that $Q_h^\star$ and $V_h^\star$ are also $L_V$-Lipschitz continuous for some constant $L_V$.*

See [38, 11] for conditions that relate $L_V$ to $L_r$ and $L_T$.

---

**Algorithm 1** Model-Based Reinforcement Learning with Adaptive Partitioning (ADAMB)

---
1: **procedure** ADAMB($\mathcal{S}, \mathcal{A}, \mathcal{D}, H, K, \delta$)
2:     Initialize partitions $\mathcal{P}_h^0 = \mathcal{S} \times \mathcal{A}$ for $h \in [H]$, estimates $\overline{\mathbf{Q}}_h^0(\cdot) = \overline{\mathbf{V}}_h^k(\cdot) = H - h + 1$
3:     **for** each episode $k \leftarrow 1, \ldots K$ **do**
4:         Receive starting state $X_1^k$
5:         **for** each step $h \leftarrow 1, \ldots, H$ **do**
6:             Observe $X_h^k$ and determine $\text{RELEVANT}_h^k(X_h^k) = \{B \in \mathcal{P}_h^{k-1} | X_h^k \in B\}$
7:             Greedy selection rule: pick $B_h^k = \text{argmax}_{B \in \text{RELEVANT}_h^k(X_h^k)} \overline{\mathbf{Q}}_h^{k-1}(B)$
8:             Play action $A_h^k = \tilde{a}(B_h^k)$ associated with ball $B_h^k$; receive $R_h^k$ and transition to $X_{h+1}^k$
9:             Update counts for $n_h^k(B_h^k), \bar{\mathbf{r}}_h^k(B_h^k)$, and $\overline{\mathbf{T}}_h^k(\cdot \mid B_h^k)$
10:            **if** $n_h^k(B_h^k) + 1 \geq n_+(B_h^k)$ **then** REFINE PARTITION($B_h^k$)
            COMPUTE ESTIMATES$(R_h^k, B_h^k)_{h=1}^H$
11: **procedure** REFINE PARTITION($B, h, k$)
12:     Construct $\mathcal{P}(B) = \{B_1, \ldots, B_{2^d}\}$ a $2^{-(\ell(B)+1)}$-dyadic partition of $B$
13:     Update $\mathcal{P}_h^k = \mathcal{P}_h^{k-1} \cup \mathcal{P}(B) \setminus B$
14:     For each $B_i$, initialize $n_h^k(B_i) = n_h^k(B), \bar{\mathbf{r}}_h^k(B_i) = \bar{\mathbf{r}}_h^k(B)$ and $\overline{\mathbf{T}}_h^k(\cdot \mid B_i) \sim \overline{\mathbf{T}}_h^k(\cdot \mid B)$
15: **procedure** COMPUTE ESTIMATES$((B_h^k, R_h^k, X_{h+1}^k)_{h=1}^H)$
16:     **for** each $h \leftarrow 1, \ldots H$ and $B \in \mathcal{P}_h^k$ **do** : Update $\overline{\mathbf{Q}}_h^k(B)$ and $\overline{\mathbf{V}}_h^k(\cdot)$ via Eq. (4) and Eq. (6)

---

## 3   Algorithm

We now present our ***Model-Based RL with Adaptive Partitioning*** algorithm, which we refer to as ADAMB. At a high level, ADAMB maintains an adaptive partition of $\mathcal{S}_h \times \mathcal{A}_h$ for each step $h$, and uses *optimistic value-iteration* over this partition. It takes as input the number of episodes $K$, metric $\mathcal{D}$ over $\mathcal{S} \times \mathcal{A}$, and the Lipschitz constants. It maintains *optimistic estimates* for $r_h(x, a)$ and $T_h(\cdot \mid x, a)$ (i.e. high-probability uniform upper bounds $\forall h, x, a$). These are used for performing value iteration to obtain optimistic estimates $\overline{\mathbf{Q}}_h$ and $\overline{\mathbf{V}}_h$ via one-step updates in Eq. (4) and Eq. (6). For full pseudocode of the algorithm, and a discussion on implementation details, see Appendix G.

**Adaptive State-Action Partitioning**: For each step $h \in [H]$, ADAMB maintains a partition of the space $\mathcal{S}_h \times \mathcal{A}_h$ into a collection of 'balls' which is refined over episodes $k \in [K]$. We denote $\mathcal{P}_h^k$ to be the partition for step $h$ at the end of episode $k$; the initial partition is set as $\mathcal{P}_h^0 = \mathcal{S} \times \mathcal{A} \; \forall h \in [H]$. Each element $B \in \mathcal{P}_h^k$ is a ball of the form $B = \mathcal{S}(B) \times \mathcal{A}(B)$, where $\mathcal{S}(B) \subset \mathcal{S}$ (respectively $\mathcal{A}(B) \subset \mathcal{A}$) is the projection of ball $B$ onto its corresponding state (action) space. We let $(\tilde{x}(B), \tilde{a}(B))$ be the center of $B$ and denote $\mathcal{D}(B) = \sup\{\mathcal{D}((x, a), (y, b)) \mid (x, a), (y, b) \in B\}$ to be the diameter of a ball $B$. The partition $\mathcal{P}_h^k$ can also be represented as a tree, with leaf nodes representing *active* balls, and inactive *parent* balls of $B \in \mathcal{P}_h^k$ corresponding to $\{B' \in \mathcal{P}_h^{k'} | B' \supset B, k' < k\}$; moreover, $\ell(B)$ is the depth of $B$ in the tree (with the root at level 0). See Figure 1 for an example partition and tree generated by the algorithm. Let

$$\mathcal{S}(\mathcal{P}_h^k) := \bigcup_{B \in \mathcal{P}_h^k \text{ s.t. } \nexists B' \in \mathcal{P}_h^k, \mathcal{S}(B') \subset \mathcal{S}(B)} \mathcal{S}(B) \tag{3}$$

denote the partition over the state spaced induce by the current state-action partition $\mathcal{P}_h^k$. We can verify that the above constructed $\mathcal{S}(\mathcal{P}_h^k)$ is indeed a partition of $\mathcal{S}$ because the partition $\mathcal{P}_h^k$ is constructed according to a dyadic partitioning.

While our partitioning works for any compact metric space, a canonical example to keep in mind is $\mathcal{S} = [0, 1]^{d_\mathcal{S}}, \mathcal{A} = [0, 1]^{d_\mathcal{A}}$ with the infinity norm $\mathcal{D}((x, a), (x', a')) = ||(x, a) - (x', a')||_\infty$ (which was used in some of the simulations). We illustrate this in Fig. 1 for $d_\mathcal{S} = d_\mathcal{A} = 1$. We define $\ell(B) = -\log_2(\mathcal{D}(B))$ to be the *level* of a ball $B$, and construct $B$ as a level-$\ell(B)$ *dyadic cube* in the metric-space $(\mathcal{S} \times \mathcal{A}, \mathcal{D})$. In our example of $([0, 1]^2, || \cdot ||_\infty)$, a ball $B$ is an axis-aligned cube of length $2^{-\ell(B)}$ and corners in $2^{-\ell(B)}\mathbb{Z}^2$, as depicted in Fig. 1.

At the end of each episode, for each active ball $B \in \mathcal{P}_h^k$ ADAMB maintains three statistics:

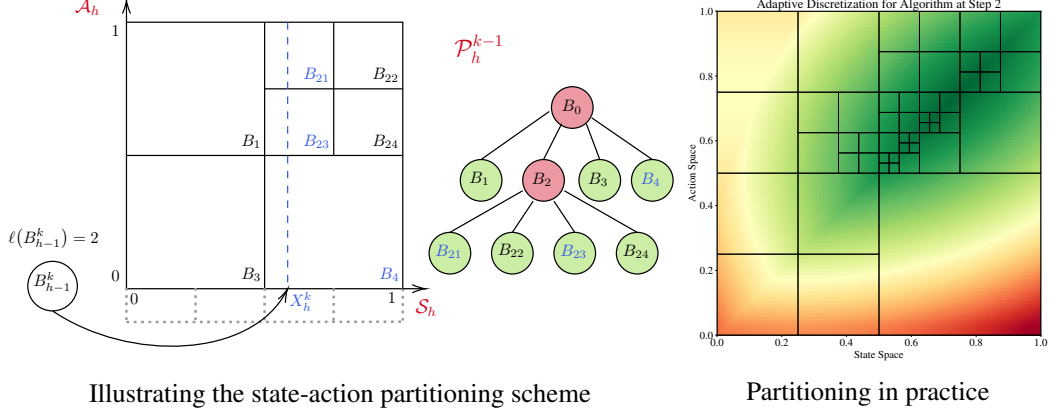

Illustrating the state-action partitioning scheme                Partitioning in practice

Figure 1: *Partitioning scheme for $\mathcal{S} \times \mathcal{A} = [0,1]^2$: On the left, we illustrate our scheme. Partition $\mathcal{P}_h^{k-1}$ is depicted with corresponding tree (showing active balls in green, inactive parents in red). The algorithm plays ball $B_{h-1}^k$ in step $h-1$, leading to new state $X_h^k$. Since $\ell(B_{h-1}^k) = 2$, we store transition estimates $\overline{\mathbf{T}}_{h-1}^k(\cdot \mid B_{h-1}^k)$ for all subsets of $\mathcal{S}_h$ of diameter $2^{-2}$ (depicted via dotted lines). The set of relevant balls $\mathrm{RELEVANT}_h^k(X_h^k) = \{B_4, B_{21}, B_{23}\}$ are highlighted in blue. On the right, we show the partition $\mathcal{P}_2^K$ from one of our synthetic experiments (See 'Oil Discovery' in Appendix H). The colors denote the true $Q_2^\star(\cdot)$ values, with green corresponding to higher values. Note that the partition is more refined in areas which have higher $Q_2^\star(\cdot)$.*

- $n_h^k(B)$: the number of times the ball $B$ has been *selected* up to and including episode $k$.

- $\hat{\mathbf{r}}_h^k(B)$: the empirical (instantaneous) reward earned from playing actions in $B$.
  $\bar{\mathbf{r}}_h^k(B)$: the empirical reward earned from playing actions in $B$ and its ancestors.

- $\{\hat{\mathbf{T}}_h^k(\cdot \mid B)\}$: the empirical fractions of transitions to sets in a $2^{-\ell(B)}$-coarse partition of $\mathcal{S}_{h+1}$ (which we denote as $\square_{\ell(B)}$) after playing actions in $B$.
  $\{\overline{\mathbf{T}}_h^k(\cdot \mid B)\}$: the empirical fractions of transitions from playing actions in $B$ and its ancestors.

These estimates are used to construct *optimistic Q-function estimates* $\overline{\mathbf{Q}}_h^k(B)$ for each $B \in \mathcal{P}_h^k$. Each ball $B \in \mathcal{P}_h^k$ has an *associated action* $\tilde{a}(B) \in \mathcal{A}(B)$ (we take this to be the center of the ball $\mathcal{A}(B)$).

**The ADAMB Algorithm**: Given the above partitions and statistics, the algorithm proceeds as follows. In each episode $k$ and step $h$, ADAMB observes state $X_h^k$, and finds all *relevant* balls $\mathrm{RELEVANT}_h^k(X_h^k) = \{B \in \mathcal{P}_h^{k-1} \mid X_h^k \in B\}$ (see Fig. 1). It then selects an action according to a *greedy selection rule*, picking $B_h^k \in \mathrm{RELEVANT}_h^k(X_h^k)$ with highest $\overline{\mathbf{Q}}_h^{k-1}(B)$, and plays action $\tilde{a}(B_h^k)$. Note that the algorithm can also play any action $a$ such that $(X_h^k, a) \in B_h^k$ uniformly at random and the theory still applies. Next, the algorithm *updates counts* for $\hat{\mathbf{r}}_h^k(B_h^k)$ and $\hat{\mathbf{T}}_h^k(\cdot \mid B_h^k)$ based on the observed reward $R_h^k$ and transition to $X_{h+1}^k$. Following this, it *refines the partition* if needed. Finally, at the end of the episode, ADAMB *updates estimates* by solving for $\overline{\mathbf{Q}}_h^k(\cdot)$ which are used in the next episode. We now describe the last three subroutines in more detail; see Algorithm 2 for the full pseudocode, and Appendix G.1 for implementation, space, and run-time analysis.

**Update Counts**: After playing active ball $B_h^k$ and observing $(R_h^k, X_{h+1}^k)$ for episode $k$ step $h$,
– Increment counts and reward estimates according to

$$n_h^k(B_h^k) = n_h^{k-1}(B_h^k) + 1 \quad \text{and} \quad \hat{\mathbf{r}}_h^k(B_h^k) = \frac{n_h^{k-1}(B_h^k)\hat{\mathbf{r}}_h^{k-1}(B_h^k) + R_h^k}{n_h^k(B_h^k)}.$$

– Update $\hat{\mathbf{T}}_h^k(\cdot \mid B_h^k)$ as follows: For each set $A$ in a $2^{-\ell(B)}$-coarse partition of $\mathcal{S}_{h+1}$ denoted by $\square_{\ell(B)}$, we set

$$\hat{\mathbf{T}}_h^k(A \mid B) = \frac{n_h^{k-1}(B_h^k)\hat{\mathbf{T}}_h^{k-1}(A \mid B) + \mathbb{1}_{\{X_{h+1}^k \in A\}}}{n_h^k(B_h^k)}.$$

This is maintaining an empirical estimate of the transition kernel for a ball $B$ at a level of granularity proportional to its diameter $\mathcal{D}(B) = 2^{-\ell(B)}$.

**Refine Partition**: To refine the partition over episodes, we split a ball when the confidence in its estimate is smaller than its diameter. Formally, for any ball $B$, we define a *splitting threshold* $n_+(B) = \phi 2^{\gamma \ell(B)}$, and partition $B$ once we have $n_h^k(B) + 1 \geq n_+(B)$. Note the splitting threshold grows exponentially with the level. More concretely the splitting threshold is defined via

$$n_+(B) = \phi 2^{d_{\mathcal{S}} \ell(B)} \qquad d_{\mathcal{S}} > 2$$
$$n_+(B) = \phi 2^{(d_{\mathcal{S}}+2)\ell(B)} \qquad d_{\mathcal{S}} \leq 2$$

where the difference in terms comes from the Wasserstein concentration. This is in contrast to the splitting threshold for the model-free algorithm where $n_+(\mathcal{B}) = 2^{2\ell(B)}$ [38]. The $\phi$ term is chosen to minimize the dependence on $H$ in the final regret bound where $\phi = H^{(d+d_{\mathcal{S}})/(d+1)}$.

In episode $k$ step $h$, if we need to split $B_h^k$, then we partition $\mathcal{S}(B_h^k) \times \mathcal{A}(B_h^k)$ using new balls each of diameter $\frac{1}{2}\mathcal{D}(B_h^k)$. This partition $\mathcal{P}(B)$ can be constructed by taking a cross product of a level $(\ell(B) + 1)$-dyadic partition of $\mathcal{S}(B_h^k)$ and a level-$(\ell(B) + 1)$ dyadic partition of $\mathcal{A}(B_h^k)$. We then remove $B$ and add $\mathcal{P}(B)$ to $\mathcal{P}_h^{k-1}$ to form the new partition $\mathcal{P}_h^k$. In practice, each child ball can inherit all estimates from its parent, and counts for the parent ball are not updated from then on. However, for ease of presentation and analysis we assume each child ball starts off with fresh estimates of $\hat{\mathbf{r}}_h^k(\cdot)$ and $\hat{\mathbf{T}}_h^k(\cdot)$ and use $\bar{\mathbf{r}}_h^k(\cdot)$ and $\overline{\mathbf{T}}_h^k(\cdot)$ to denote the aggregate statistics.

**Compute Estimates**: At the end of the episode we set

$$\bar{\mathbf{r}}_h^k(B) = \frac{\sum_{B' \supseteq B} \hat{\mathbf{r}}_h^k(B') n_h^k(B')}{\sum_{B' \supseteq B} n_h^k(B')}$$

$$\overline{\mathbf{T}}_h^k(A \mid B) = \frac{\sum_{B' \supseteq B} \sum_{A' \in \square_{\ell(B')}; A \subset A'} 2^{-d_{\mathcal{S}}(\ell(B')-\ell(B))} n_h^k(B') \hat{\mathbf{T}}_h^k(A' \mid B')}{\sum_{B' \supseteq B} n_h^k(B')}$$

When aggregating the estimates of the transition kernel, we need to multiply by a factor to ensure we obtain a valid distribution. This is because any ancestor $B'$ of $B$ maintain empirical estimates of the transition kernel to a level $\square_{\ell(B')}$. Thus, we need to split the mass in order to construct a distribution over $\square_{\ell(B)}$. We also define confidence terms typically used in multi-armed bandits which are defined via:

$$\text{RUCB}_h^k(B) = \sqrt{\frac{8\log(2HK^2/\delta)}{\sum_{B' \supseteq B} n_h^k(B')}} + 4L_r \mathcal{D}(B)$$

$\text{TUCB}_h^k(B)$

$$= \begin{cases} L_V\left((5L_T + 4)\mathcal{D}(B) + 4\sqrt{\frac{\log(HK^2/\delta)}{\sum_{B' \subseteq B} n_h^k(B')}} + c\left(\sum_{B' \subseteq B} n_h^k(B')\right)^{-1/d_{\mathcal{S}}}\right) & \text{if } d_{\mathcal{S}} > 2 \\ L_V\left((5L_T + 6)\mathcal{D}(B) + 4\sqrt{\frac{\log(HK^2/\delta)}{\sum_{B' \supseteq B} n_h^k(B')}} + c\sqrt{\frac{2^{d_{\mathcal{S}}\ell(B)}}{\sum_{B' \supseteq B} n_h^k(B')}}\right) & \text{if } d_{\mathcal{S}} \leq 2 \end{cases}$$

The difference in definitions of $\text{TUCB}_h^k(\cdot)$ comes from the Wasserstein concentration in Appendix D. With these in place we set

$$\overline{\mathbf{Q}}_h^k(B) := \begin{cases} \bar{\mathbf{r}}_H^k(B) + \text{RUCB}_H^k(B) & \text{if } h = H \\ \bar{\mathbf{r}}_h^k(B) + \text{RUCB}_h^k(B) + \mathbb{E}_{A \sim \overline{\mathbf{T}}_h^k(\cdot|B)}\left[\overline{\mathbf{V}}_{h+1}^{k-1}(A)\right] + \text{TUCB}_h^k(B) & \text{if } h < H \end{cases} \tag{4}$$

mimicing the Bellman equations by replacing the true unknown quantities with their estimates. The value function estimates are computed in a two-stage process. For each ball $A \in \mathcal{S}(\mathcal{P}_h^k)$ we have that

$$\widetilde{\mathbf{V}}_h^k(A) := \min\{\widetilde{\mathbf{V}}_h^{k-1}(A), \max_{B \in \mathcal{P}_h^k : \mathcal{S}(B) \supseteq A} \overline{\mathbf{Q}}_h^k(B)\}. \tag{5}$$

For technical reasons we need to construct a Lipschitz continuous function to estimate the value function in order to show concentration of the transition kernel estimates. For each point $x \in \mathcal{S}_h$ we define

$$\overline{\mathbf{V}}_h^k(x) = \min_{A' \in \mathcal{S}(\mathcal{P}_h^k)} \left( \widetilde{\mathbf{V}}_h^k(A') + L_V \mathcal{D}_S(x, \tilde{x}(A')) \right). \tag{6}$$

However, as the support of $\overline{\mathbf{T}}_h^k(\cdot \mid B)$ is only over sets in $\square_{\ell(B)}$ we overload notation to let $\overline{\mathbf{V}}_h^k(A) = \overline{\mathbf{V}}_h^k(\tilde{x}(A))$. We equivalently overload notation so that $x \sim \overline{\mathbf{T}}_h^k(\cdot \mid B)$ refers to sampling over the centers associated to balls in $\square_{\ell(B)}$.

This corresponds to a value-iteration step, where we replace the true rewards and transitions in the Bellman Equations (Eq. (1)) with their (optimistic) estimates. We only compute one-step updates as in [13], which reduces computational complexity as opposed to solving the full Bellman update.

Note that at the end of the episode, for each step $h$, we only need to update $\overline{\mathbf{Q}}_h^k(B)$ for $B = B_h^k$ and $\widetilde{\mathbf{V}}_h^k(A)$ for each $A \in \mathcal{S}(\mathcal{P}_h^k)$ such that $A \subseteq B_h^k$. $\overline{\mathbf{V}}_h^k$ is only used to compute the expectation in Eq. (4), and thus it is only evaluated in episode $k+1$ for balls $A$ in the $2^{-\ell(B_{h-1}^{k+1})}$-coarse partition of $\mathcal{S}_h$.

## 4 Main Results

We start with giving worst-case regret guarantees for ADAMB.

**Theorem 4.1.** *Let $d = d_A + d_S$, then the regret of* ADAMB *for any sequence of starting states $\{X_1^k\}_{k=1}^K$ is upper bounded with probability at least $1 - \delta$ by*

$$R(K) \lesssim \begin{cases} LH^{1+\frac{1}{d+1}} K^{\frac{d+d_S-1}{d+d_S}} & d_S > 2 \\ LH^{1+\frac{1}{d+1}} K^{\frac{d+d_S+1}{d+d_S+2}} & d_S \le 2 \end{cases}$$

*where $L = 1 + L_r + L_V + L_V L_T$ and $\lesssim$ omits poly-logarithmic factors of $\frac{1}{\delta}, H, K, d$, and any universal constants.*

**Comparison to Model-Free Methods**: Previous model-free algorithms achieve worst-case bounds scaling via $H^{5/2} K^{(d+1)/(d+2)}$, which achieve the optimal worst-case dependence on the dimension $d$ [38]. The bounds presented here have better dependence on the number of steps $H$. This is expected, as current analysis for model-free and model-based algorithms under tabular settings shows that model-based algorithms achieve better dependence on $H$. However, under the Lipschitz assumptions here the constant $L$ also scales with $H$ so the true dependence is somewhat masked. A modification of our algorithm that uses full planning instead of one-step planning will achieve linear dependence on $H$, with the negative effect of increased run-time. When we compare the dependence on the number of episodes $K$ we see that the dependence is worse - primarily due to the additional factor of $d_S$, the covering dimension of the state-space. This term arises as model-based algorithms maintain an estimate of the transition kernel, whose complexity depends on $d_S$.

**Comparison to Model-Based Methods**: Current state of the art model-based algorithms (KERNEL-UCBVI) achieve regret scaling like $H^3 K^{2d/(2d+1)}$ [11]. We achieve better scaling with respect to both $H$ and $K$, and our algorithm has lower time and space complexity. However, we require additional oracle assumptions on the metric space to be able to construct packings and coverings efficiently, whereas KERNEL-UCBVI uses the data and the metric itself. Better dependence on $H$ and $K$ is primarily achieved by using recent work on concentration for the Wasserstein metric. These guarantees allow us to construct tighter confidence intervals which are independent of $H$, obviating the need to construct a covering of $H$-uniformly bounded Lipschitz functions like prior work (see Appendix D).

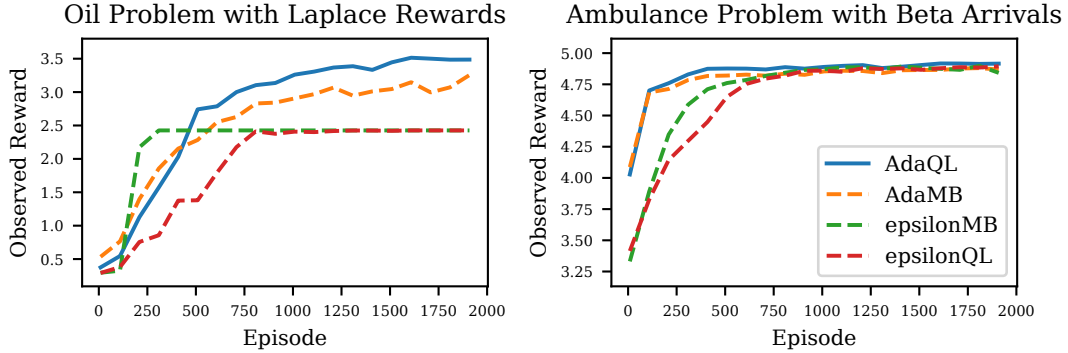

Figure 2: *Here we compare four algorithms:* EPSQL *and* EPSMB, *uniform discretization versions of model-free and model-based algorithms, and* ADAQL *and* ADAMB, *their adaptive discretization counterparts. More simulation results are in Appendix H.*

In addition, KERNEL-UCBVI uses a fixed bandwidth parameter in their kernel interpolation. We instead keep an adaptive partition of the space, helping our algorithm maintain a smaller and more efficient discretization. This technique also lends itself to show instance dependent bounds, which we leave for future work.

**Discussion on Instance-Specific Bounds**: The bounds presented here are worst-case, problem independent guarantees. Recent work has shown that model-free algorithms are able to get problem dependent guarantees which depend on the zooming dimension instead of the covering dimension of the space [7]. Extending this result to model-based algorithms will be more technical, due to requiring improved concentration guarantees for the transition kernel. Most model-based algorithms require showing *uniform concentration*, in particular that the estimate of the transition kernel concentrates well when taking expectation over any Lipschitz function. Getting tighter bounds for model-based algorithms in continuous settings will require showing that the transition kernel is naturally estimated well in parts of the space that matter - as the state-visitation frequency is dependent on the policy used. In Appendix C we discuss the transition concentration in more details.

**Proof Sketch**: The high level proof is divided into three sections. First we show *concentration and clean-events*, under which our estimates $\bar{r}$ and $\overline{T}$ constitute upper bounds on the relevant quantities (Appendix D). Afterwards, we show a *regret decomposition*, which relates the difference between the estimated value and the value accumulated by the algorithm to the bonus terms (Appendix E). Lastly, we use an LP-based argument to bound the *worst-case size of the partition* and the *sum of the bonus terms* which is used for the final regret bound (Appendix F). The full proof sketch is in Appendix C.

## 5   Conclusion

We presented an algorithm using adaptive discretization for model-based online reinforcement learning based on one-step planning. In worst case instances, we showed regret bounds for our algorithm which are competitive with other model-based algorithms in continuous settings under the assumption that the underlying dynamics of the system are Lipschitz continuous with respect to a known metric on the space. We also provided simulations comparing model-based and model-free methods using an adaptive and fixed discretizations of the space on several canonical control problems. Our experiments showed that adaptive partitioning empirically performs better than fixed discretizations in terms of both faster convergence and lower memory.

One future direction for the work is analyzing the discrepancy between model-based and model-free methods in continuous settings, as model-based algorithms so far have sub-optimal dependence on the dimension of the space. Moreover, we hope to characterize problems where model-based methods using adaptive discretization are able to outperform model-free methods using a gap-dependent analysis inspired by recent gap-dependent analysis for tabular algorithms [37].

## Acknowledgements

We are grateful to the referees for their constructive input. Part of this work was done while Sean Sinclair and Christina Yu were visiting the Simons Institute for the Theory of Computing for the semester on the Theory of Reinforcement Learning. We also gratefully acknowledge funding from the NSF under grants ECCS-1847393, DMS-1839346, CCF-1948256, and CNS-1955997, the ARL under grant W911NF-17-1-0094, and the Cornell Engaged Grant: Applied Mathematics in Action.

## Broader Impact

**Exploring Memory-Computation Trade-offs in RL**

Reinforcement learning policies have enjoyed remarkable success in recent years, in particular in the context of large-scale game playing. These results, however, mask the high underlying costs in terms of computational resources and training time that the demonstrations requires [36, 26, 27, 35]. For example, the AlphaGo Zero algorithm that mastered Chess and Go from scratch trained their algorithm over 72 hours using 4 TPUs and 64 GPUs. These results, while highlighting the intrinsic power in reinforcement learning algorithms, are computationally infeasible for applying algorithms to RL tasks in computing systems. As an example, RL approaches have received much interest in several of the following problems:

- *Memory Management*: Many computing systems have two sources of memory; on-chip memory which is fast but limited, and off-chip memory which has low bandwidth and suffers from high latency. Designing memory controllers for these system require a scheduling policy to adapt to changes in workload and memory reference streams, ensuring consistency in the memory, and controlling for long-term consequences of scheduling decisions [1, 2, 8].

- *Online Resource Allocation*: Cloud-based clusters for high performance computing must decide how to allocate computing resources to different users or tasks with highly variable demand. Controllers for these systems must make decisions online to manage the trade-offs between computation cost, server costs, and delay in job-completions. Recent work has studied RL algorithms for such problems [15, 23, 28, 22].

Common to all of these examples are computation and storage limitations on the devices used for the controller.

- *Limited Memory*: On chip memory is expensive and off-chip memory access has low-bandwidth. As any reinforcement learning algorithm requires memory to store estimates of relevant quantities - RL algorithms for computing systems must manage their computational requirements.

- *Power Consumption*: Many applications require low-power consumption for executing RL policies on general computing platforms.

- *Latency Requirements*: Many problems for computing systems (e.g. memory management) have strict latency quality of service requirements that limits reinforcement learning algorithms to execute their policy quickly.

Our algorithm ADAMB takes a first step towards designing efficient reinforcement learning algorithms for continuous (or large finite) spaces, where efficient means both low-regret, but also low storage and computation complexity (see Table 1). ADAMB is motivated by recent algorithms for reinforcement learning on memory constrained devices which use a technique called cerebellar model articulation controller (CMAC). This technique uses a random-discretizations of the space at various levels of coarseness [15]. Moreover, heuristic algorithms which use discretizations (either fixed or adaptive) have been extensively studied on various tasks [32, 39, 22].

We are able to show that our algorithm achieves good dependence with respect to $K$ on all three dimensions (regret, computation, and storage complexity). With future work we hope to determine problem specific guarantees, exhibiting how these adaptive partitioning algorithms are able to extract structure common in computing systems problems.

**Societal Projects**

An important component of this research and proposed future work is the focus on building a data-driven simulator for societal systems, and use these for benchmarking and testing RL algorithms. Many problems in computing systems and operations research exhibit additional structure, whereby analyzing how algorithms are able to extract and exploit that structure is paramount to their success in real-time applications. Examples like the Behaviour Suite for Reinforcement Learning [29] are designed with toy examples to understand different components in RL algorithms (exploration, generalization, memory, etc) but is not application driven by design. Our research takes the first-step in analyzing the performance of adaptive discretization for these tasks. We summarize several problems below:

- *Ambulance Routing* (See Appendix H): This problem generalizes the canonical $k$-Server problem commonly studied in theoretical computer science. An ambulance operator controls a fleet of ambulances and must decide on locations to station the ambulances in order to minimize transportation cost and travel-time to service patients arriving from an unknown distribution.

- *Oil Problem* (See Appendix H): This problem generalizes the common discrete Grid-World environment. Here an operator controls a measurement system (say a machine used to drill for oil) and must decide on locations to station the rig in order to minimize transportation cost and maximize the probability of obtaining the resource.

- *Online Resource Allocation*: Many problems in online resource allocation can be formulated and solved through reinforcement learning. For a concrete example, consider a mobile-food pantry that needs to design allocation rules for how much resources to allocate to a specific location without any knowledge on the distribution of demands for locations to come. Reinforcement learning policies in this setting will have to balance between designing algorithms that utilize all of the resources (*pareto-optimality*), while ensuring fairness across locations (*envy-freeness*). This problem helps serve as an interdisciplinary connection between reinforcement learning, fairness, and resource allocation.

## Footnotes

[1]The code for the experiments are available at https://github.com/seanrsinclair/AdaptiveQLearning. A full report is available at https://arxiv.org/abs/2007.00717.

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
