[Supplementary Material]

# A  Table of Notation

| Symbol | Definition |
|---|---|
| Problem setting specifications | |
| $\mathcal{S}, \mathcal{A}, H, K$ | State space, action space, steps per episode, number of episodes |
| $r_h(x,a)$, $T_h(\cdot \mid x,a)$ | Average reward/transition kernel for taking action $a$ in state $x$ at step $h$ |
| $\pi_h, V_h^\pi(\cdot), Q_h^\pi(\cdot,\cdot)$ | Arbitrary step-$h$ policy, and Value/$Q$-function at step $h$ under $\pi$ |
| $\pi_h^\star, V_h^\star(\cdot), Q_h^\star(\cdot,\cdot)$ | Optimal step-$h$ policy, and corresponding Value/$Q$-function |
| $L_r, L_T, L_V$ | Lipschitz constants for $r$, $T$ and $V^\star$ respectively |
| $\mathcal{D}_\mathcal{S}, D_\mathcal{A}, \mathcal{D}$ | Metrics on $\mathcal{S}$, $\mathcal{A}$, and $\mathcal{S} \times \mathcal{A}$ respectively |
| Algorithm variables and parameters | |
| $k, h$ | Index for episode, index for step in episode |
| $(X_h^k, A_h^k, R_h^k)$ | State, action, and received reward under algorithm at step h in episode k |
| $\mathcal{P}_h^k$ | Partition tree of $\mathcal{S} \times \mathcal{A}$ for step $h$ at end of episode $k$ |
| $\text{RELEVANT}_h^k(x)$ | Set of balls relevant for $x$ at $(k,h)$ (i.e., $\{B \in \mathcal{P}_h^{k-1} \mid (x,a) \in B$ for some $a \in \mathcal{A}\}$) |
| $\tilde{x}(B), \tilde{a}(B)$ | Associated state/action for ball $B$ (i.e., 'center' of ball $B$) |
| $B_h^k$ | Ball in $\mathcal{P}_h^{k-1}$ selected at $(k,h)$ ($\text{argmax}_{B \in \text{RELEVANT}_h^k(X_h^k)} \overline{\mathbf{Q}}_h^{k-1}(B)$) |
| $n_+(B)$ | Threshold number of samples after which ball $B$ is split |
| $\overline{\mathbf{Q}}_h^k(B)$ | $Q$-function estimates for ball $B \in \mathcal{P}_h^k$, at end of episode $k$ |
| $\widetilde{\mathbf{V}}_h^k(A)$ | $V$-function estimate for a ball $A \in \mathcal{S}(\mathcal{P}_h^k)$, at end of episode $k$ |
| $\overline{\mathbf{V}}_h^k(x)$ | $V$-function estimate for a point $x \in \mathcal{S}$, at end of episode $k$ |
| $n_h^k(B)$ | Number of times $B$ has been chosen by the end of episode $k$ |
| $\hat{\mathbf{r}}_h^k(B), \hat{\mathbf{T}}_h^k(\cdot \mid B)$ | Empirical rewards and transitions from ball $B \in \mathcal{P}_h^k$ at end of episode $k$ |
| $\overline{\mathbf{r}}_h^k(B), \overline{\mathbf{T}}_h^k(\cdot \mid B)$ | Inherited reward/transition estimates for $B \in \mathcal{P}_h^k$ at end of episode $k$ |
| Definitions used in the analysis | |
| $\Delta(\mathcal{S})$ | Set of probability measures on $\mathcal{S}$ |
| $\square_\ell$ | Set of dyadic cubes of $\mathcal{S}$ of diameter $2^{-\ell}$ |
| $\mathcal{S}(\mathcal{P}_h^k)$ | Induced state partition from $\mathcal{P}_h^k$ |
| $\mathcal{S}(\mathcal{P}_h^k, x)$ | Region in $\mathcal{S}(\mathcal{P}_h^k)$ containing the point $x$ |
| $\mathcal{S}(B), \mathcal{A}(B)$ | Projection of a ball $B = B_\mathcal{S} \times B_\mathcal{A}$ to $B_\mathcal{S}$ and $B_\mathcal{A}$ accordingly |
| $\mathcal{D}(B)$ | The diameter of a ball $B$ |
| $\ell(B)$ | The depth in the tree of ball $B$, equivalent to $\log_2(\mathcal{D}(\mathcal{S} \times \mathcal{A})/\mathcal{D}(B))$ |
| $R(K)$ | The regret up to episode $K$ |
| $\mathbb{E}[V_{h+1}(\hat{x}) \mid x,a]$ | $\mathbb{E}_{\hat{x} \sim \mathbb{P}_h(\cdot \mid x,a)}[V_{h+1}(\hat{x})]$ |
| $\mathcal{F}_k$ | Sigma-field generated by all information up to start of episode $k$ |
| $\mathbb{E}^k[X]$ | Expectation conditioned on information before episode $k$, i.e., $\mathbb{E}[X \mid \mathcal{F}_k]$ |

Table 2: List of common notation

# B  Related Work

There is an extensive literature on model-based reinforcement learning; below, we highlight the work which is closest to ours, but for more extensive references, see [43] for RL, and [6, 41] for bandits.

**Tabular RL**: There is a long line of research on the sample complexity and regret for RL in tabular settings. In particular, the first asymptotically tight regret bound for tabular model-based algorithms with non-stationary dynamics of $O(H^{3/2}\sqrt{SAK})$ where $S, A$ are the size of state/action spaces respectively was established in [5]. These bounds were matched (in terms of $K$) using an 'asynchronous value-iteration' (or one-step planning) approach [4, 13], which is simpler to implement. Our work extends this latter approach to continuous spaces via adaptive discretization. More recently,

analysis was extended to develop instance-dependent instead of worst-case guarantees [49, 37] There has also been similar regret analysis for model-free algorithms [16].

**Parametric Algorithms**: For RL in continuous spaces, several recent works have focused on the use of linear function approximation [17, 12, 50, 45, 44, 30]. These works assume that the controller has a feature-extractor under which the process is well-approximated via a linear model. While the resulting algorithms can be computationally efficient, they incur linear loss when the underlying process does not meet their strict parametric assumptions. Other work has extended this approach to problems with bounded eluder dimension [44, 33].

**Nonparametric Algorithms**: In contrast, nonparametric algorithms only require mild local assumptions on the underlying process, most commonly, that the $Q$-function is Lipschitz continuous with respect to a given metric. For example, [48] and [34] consider nearest-neighbour methods for deterministic, infinite horizon discounted settings. Others assume access to a generative model [18, 14].

The works closest to ours concerns algorithms with provable guarantees for continuous state-action settings (see also Table 1). In model-free settings, tabular algorithms have been adapted to continuous state-action spaces via fixed discretization (i.e., $\epsilon$-nets) [42]. In model-based settings, researchers have tackled continuous spaces via kernel methods, based on either a fixed discretization of the space [21], or more recently, without resorting to discretization [11]. While the latter does learn a data-driven representation of the space via kernels, it requires solving a complex optimization problem at each step, and hence is efficient mainly for finite action sets (more discussion on this is in Section 4). Finally, adaptive discretization has been successfully implemented in model-free settings [38, 7], and this provides a good benchmark for our algorithm, and for comparing model-free and model-based algorithms.

**Discretization Based Approaches**: Discretization-based approaches to reinforcement learning have been investigated heuristically through many different settings. One line of work investigates adaptive basis functions, where the parameters of the functional model (e.g. neural network) are learned online while also adapting the basis functions as well [19, 25, 47]. Similar techniques are done with soft state aggregation [39]. Most similar to our algorithm, though, are tree based partitioning rules, which store a hierarchical tree based partition of the state and action space (much like ADAMB) which is refined over time [32, 22]. These were tested heuristically with various splitting rules (e.g. Gini index, etc), where instead we split based off the metric and level of uncertainty in the estimates.

## C  Proof Sketch

The high level proof is divided into three sections. First, we show *concentration and clean-events*, under which our estimates constitute upper bounds on the relevant quantities. Afterwards, we show a *regret decomposition*, which relates the difference between the estimated value and the value accumulated by the algorithm with the bonus terms. Lastly, we use an LP-based argument to bound the *worst-case size of the partition* and the *sum of the bonus terms* which is used for the final regret bound. We discuss each of them briefly before giving more technical details. As the final regret-bound is technical and mostly involves algebra and combining terms, its derivation is deferred to Appendix J.

### C.1  Concentration and Clean Events (Appendix D)

ADAMB maintains estimates $\bar{\mathbf{r}}_h^k(B)$ and $\overline{\mathbf{T}}_h^k(\cdot \mid B)$ of the unknown rewards and transitions of the underlying MDP. In order to ensure that the one-step value iteration update in Equation 4 concentrates we need to verify that these estimates provide good approximations to their true quantities. In particular, applying Azuma-Hoeffding's inequality shows that:

**Lemma C.1.** *With probability at least $1 - \delta$ we have that for any $h, k \in [H] \times [K]$ and ball $B \in \mathcal{P}_h^k$, and any $(x, a) \in B$,*

$$\left| \bar{\mathbf{r}}_h^k(B) - r_h(x, a) \right| \leq \text{RUCB}_h^k(B).$$

The next step is ensuring concentration of the transition estimates $\overline{\mathbf{T}}_h^k(\cdot \mid B)$. As the algorithm takes expectations over Lipschitz functions with respect to these distributions, we use recent work on

Wasserstein distance concentration. This is in contrast to previous work that requires using a covering argument on the space of value functions in order to show concentration guarantees for the transition kernel [10, 17, 11]. In particular, we show the following:

**Lemma C.2.** *With probability at least $1 - 2\delta$ we have that for any $h, k \in [H] \times [K]$ and ball $B \in \mathcal{P}_h^k$ with $(x, a) \in B$ that*

$$d_W(\overline{\mathbf{T}}_h^k(\cdot \mid B), T_h(\cdot \mid x, a)) \leq \frac{1}{L_V} \text{TUCB}_h^k(B)$$

The main proof uses recent work on bounding the Wasserstein distance between an empirical measure and the true measure [46]. For the case when $d_{\mathcal{S}} > 2$ the concentration inequality holds up to a level of $n^{-\frac{1}{d_{\mathcal{S}}}}$ with high probability. We use this result by chaining the Wasserstein distance of various measures together. Unfortunately, the scaling does not hold for the case when $d_{\mathcal{S}} \leq 2$. In this situation we use the fact that $\overline{\mathbf{T}}_h^k(\cdot \mid B)$ is constructed as an empirical measure with finite support $|\square_{\ell(B)}| = 2^{d_{\mathcal{S}}\ell(B)}$. Although $T_h(\cdot \mid x, a)$ is a continuous distribution, we consider "snapped" versions of the distributions and repeat a similar argument. This allows us to get the scaling of $\sqrt{2^{d_{\mathcal{S}}\ell(B)}/n}$ seen in the definition of $\text{TUCB}_h^k(B)$. The result from [46] has corresponding lower bounds, showing that in the worst case scaling with respect to $d_{\mathcal{S}}$ is inevitable. As the transition bonus terms leads to the dominating terms in the regret bounds, improving on our result necessitates creating concentration intervals around the value function instead of the model [3].

The Wasserstein concentration established in the previous lemmas allows us to forgo showing uniform convergence of the transition kernels over all value functions. Indeed, the variational definition of the Wasserstein metric between measures is $d_W(\mu, \nu) = \sup_f \int f d(\mu - \nu)$ where the supremum is taken over all 1-Lipschitz functions. Noting that $V_h^\star$ and $\overline{\mathbf{V}}_h^k(\cdot)$ are constructed to be $L_V$-Lipschitz functions we therefore get that for $V = V_h^\star$ or $V = \overline{\mathbf{V}}_h^k(\cdot)$:

$$\mathbb{E}_{X \sim \overline{\mathbf{T}}_h^k(\cdot \mid B)}[V(X)] - \mathbb{E}_{X \sim T_h(\cdot \mid x, a)}[V(X)] \leq L_V d_W(\overline{\mathbf{T}}_h^k(\cdot \mid B), T_h(\cdot \mid x, a)) \leq \text{TUCB}_h^k(B).$$

Getting improved bounds for model-based algorithms in continuous spaces will necessitate showing that the algorithm does not need to show uniform concentration over all value functions or all Lipschitz functions, but rather a subset that is constructed by the algorithm.

These concentration bounds allow us to now demonstrate a principle of *optimism* for our value-function estimates. Formally, we show that conditioned on the concentration bounds on the rewards and transitions being valid, the estimates for $Q_h^\star$ and $V_h^\star$ constructed by ADAMB are indeed upper bounds for the true quantities. This follows a common approach for obtaining regret guarantees for reinforcement learning algorithms [37].

**Lemma C.3.** *With probability at least $1 - 3\delta$, the following bounds are all simultaneously true for all $k, h \in [K] \times [H]$, and any partition $\mathcal{P}_h^k$*

$$\overline{\mathbf{Q}}_h^k(B) \geq Q_h^\star(x, a) \qquad \text{for all } B \in \mathcal{P}_h^k, \text{ and } (x, a) \in B$$
$$\widetilde{\mathbf{V}}_h^k(A) \geq V_h^\star(x) \qquad \text{for all } A \in \mathcal{S}(\mathcal{P}_h^k), \text{ and } x \in A$$
$$\overline{\mathbf{V}}_h^k(x) \geq V_h^\star(x) \qquad \text{for all } x \in \mathcal{S}$$

### C.2 Regret Decomposition (Appendix E)

Similar to [13], we use one step updates for $\overline{\mathbf{Q}}_h^k(\cdot)$ and $\overline{\mathbf{V}}_h^k(\cdot)$. We thus use similar ideas to obtain the final regret decomposition, which then bounds the final regret of the algorithm by a function of the size of the partition and the sum of the bonus terms used in constructing the high probability estimates. In particular, by expanding the update rules on $\overline{\mathbf{Q}}_h^k(B)$ and $\overline{\mathbf{V}}_h^k(x)$ we can show:

**Lemma C.4.** *The expected regret for* ADAMB *can be decomposed as*

$$\mathbb{E}[R(K)] \lesssim \sum_{k=1}^K \sum_{h=1}^H \mathbb{E}\left[\widetilde{\mathbf{V}}_h^{k-1}(\mathcal{S}(\mathcal{P}_h^{k-1}, X_h^k)) - \widetilde{\mathbf{V}}_h^k(\mathcal{S}(\mathcal{P}_h^k, X_h^k))\right]$$

$$+ \sum_{h=1}^{H} \sum_{k=1}^{K} \mathbb{E}\big[2\mathrm{RUCB}_h^k(B_h^k)\big] + \sum_{h=1}^{H} \sum_{k=1}^{K} \mathbb{E}\big[2\mathrm{TUCB}_h^k(B_h^k)\big] + \sum_{k=1}^{K} \sum_{h=1}^{H} L_V \mathbb{E}\big[\mathcal{D}(B_h^k)\big].$$

*where $\mathcal{S}(\mathcal{P}_h^{k-1}, X_h^k)$ is the region in $\mathcal{S}(\mathcal{P}_h^{k-1})$ containing the point $X_h^k$.*

The first term in this expression arises from using one-step planning instead of full-step planning, and the rest due to the bias in the estimates for the reward and transitions. Using the fact that the $\widetilde{\mathbf{V}}_h^k$ are decreasing with respect to $k$ we can show that this term is upper bounded by the size of the partition. Obtaining the final regret bound then relies on finding a bound on the size of the partition and the sum of bonus terms.

## C.3 Bounds on Size of Partition and Sums of Bonus Terms (Appendix F)

We show technical lemmas that provide bounds on terms of the form $\sum_{k=1}^{K} \frac{1}{(n_h^k(B_h^k))^\alpha}$ almost surely based on the splitting rule used in the algorithm and the size of the resulting partition. We believe that this is of independent interest as many optimistic regret decompositions involve bounding sums of bonus terms over a partition that arise from concentration inequalities.

We formulate these quantities as a linear program (LP) where the objective function is to maximize either the size of the partition or the sum of bonus terms associated to a valid partition (represented as a tree) constructed by the algorithm. The constraints follow from conditions on the number of samples required before a ball is split into subsequent children balls. To derive an upper bound on the value of the LP we find a tight dual feasible solution. This argument could be broadly useful and modified for problems with additional structures by including additional constraints into the LP. In particular, we are able to show the following:

**Corollary C.5.** *For any $h \in [H]$, consider any sequence of partitions $\mathcal{P}_h^k, k \in [K]$ induced under* ADAMB *with splitting thresholds $n_+(\ell) = \phi 2^{\gamma \ell}$. Then, for any $h \in [H]$ we have:*

- $|\mathcal{P}_h^k| \leq 4^d K^{\frac{d}{d+\gamma}} \phi^{-\frac{d}{d+\gamma}}$

- *For any $\alpha, \beta \geq 0$ s.t. $\alpha \leq 1$ and $\alpha\gamma - \beta \geq 1$, we have*

$$\sum_{k=1}^{K} \frac{2^{\beta \ell(B_h^k)}}{\big(n_h^k(B_h^k)\big)^\alpha} = O\Big(\phi^{\frac{-(d\alpha+\beta)}{d+\gamma}} K^{\frac{d+(1-\alpha)\gamma+\beta}{d+\gamma}}\Big)$$

- *For any $\alpha, \beta \geq 0$ s.t. $\alpha \leq 1$ and $\alpha\gamma - \beta/\ell^\star \geq 1$ (where $\ell^\star = 2 + \frac{1}{d+\gamma}\log_2\big(\frac{K}{\phi}\big)$), we have*

$$\sum_{k=1}^{K} \frac{\ell(B_h^k)^\beta}{\big(n_h^k(B_h^k)\big)^\alpha} = O\Big(\phi^{\frac{-d\alpha}{d+\gamma}} K^{\frac{d+(1-\alpha)\gamma}{d+\gamma}} (\log_2 K)^\beta\Big)$$

We use this result with the regret decomposition to show the final regret bound. The splitting threshold $\gamma$ is taken in order to satisfy the requirements of the corollary. As the dominating term arises from the concentration of the transition kernel, for the case when $d_\mathcal{S} > 2$ the sum is of the form when $\alpha = 1/d_\mathcal{S}$ and $\beta = 0$. This gives the $K^{(d+d_\mathcal{S}-1)/(d+d_\mathcal{S})}$ term in the regret bound. The case when $d_\mathcal{S} \leq 2$ is similar.

## D Concentration Bounds, Optimism, and Clean Events

In this section we show that the bonus terms added on, namely $\mathrm{RUCB}_h^k(\cdot)$ and $\mathrm{TUCB}_h^k(\cdot)$, ensure that the estimated rewards and transitions are upper bounds for the true quantities with high probability. This follows a proof technique commonly used for multi-armed bandits and reinforcement learning, where algorithm designers ensure that relevant quantities are estimated optimistically with a bonus that declines as the number of samples increases.

For all proofs we let $\{\mathcal{F}_k\}$ denote the filtration induced by all information available to the algorithm at the start of episode $k$, i.e. $\mathcal{F}_k = \sigma\Big((X_h^{k'}, A_h^{k'}, B_h^{k'}, R_h^{k'})_{h\in[H],k'<k} \cup X_1^k\Big)$ where we include the

starting state for the episode. With this filtration in place, all of the estimates $\overline{\mathbf{Q}}_h^{k-1}, \overline{\mathbf{V}}_h^{k-1}$, and the policy $\pi_k$ are measurable with respect to $\mathcal{F}_k$.

Before stating the concentration inequalities, we first give a technical result, which we use to simplify the upper confidence terms. The proof of this result is deferred to Appendix K.

**Lemma D.1.** *For any $h, k \in [H] \times [K]$ and ball $B \in \mathcal{P}_h^k$ we have that*

$$\frac{\sum_{B' \supseteq B} \mathcal{D}(B') n_h^k(B')}{\sum_{B' \supseteq B} n_h^k(B')} \leq 4\mathcal{D}(B).$$

### D.1 Concentration of Reward Estimates

We start by showing that with probability at least $1 - \delta$, our reward estimate $\bar{\mathbf{r}}_h^k(B) + \text{RUCB}_h^k(B)$ is a *uniform* upper bound on the true mean reward $r_h(x, a)$ for any $(x, a) \in B$.

**Lemma D.2.** *With probability at least $1 - \delta$ we have that for any $h, k \in [H] \times [K]$ and ball $B \in \mathcal{P}_h^k$, and any $(x, a) \in B$,*

$$\left| \bar{\mathbf{r}}_h^k(B) - r_h(x, a) \right| \leq \text{RUCB}_h^k(B),$$

*where we define $\text{RUCB}_h^k(B) = \sqrt{\frac{8 \log(2HK^2/\delta)}{\sum_{B' \supseteq B} n_h^k(B')}} + 4L_r \mathcal{D}(B)$.*

*Proof.* Let $h, k \in [H] \times [K]$ and $B \in \mathcal{P}_h^k$ be fixed and $(x, a) \in B$ be arbitrary. First consider the left hand side of this expression,

$$
\left| \bar{\mathbf{r}}_h^k(B) - r_h(x, a) \right| = \left| \frac{\sum_{B' \supseteq B} \hat{\mathbf{r}}_h^k(B') n_h^k(B')}{\sum_{B' \supseteq B} n_h^k(B')} - r_h(x, a) \right|
$$

$$
\leq \left| \frac{\sum_{B' \supseteq B} \sum_{k' \leq k} \mathbb{1}_{\left[B_h^{k'} = B'\right]} (R_h^{k'} - r_h(X_h^{k'}, A_h^{k'}))}{\sum_{B' \supseteq B} n_h^k(B')} \right|
$$

$$
+ \left| \frac{\sum_{B' \supseteq B} \sum_{k' \leq k} \mathbb{1}_{\left[B_h^{k'} = B'\right]} (r_h(X_h^{k'}, A_h^{k'}) - r_h(x, a))}{\sum_{B' \supseteq B} n_h^k(B')} \right|.
$$

where we use the definitions of $\bar{\mathbf{r}}_h^k(B)$ and $\hat{\mathbf{r}}_h^k(B)$ and the triangle inequality.

Next, using the fact that $r_h$ is Lipschitz continuous and that $(x, a) \in B \subseteq B'$ and $(X_h^{k'}, A_h^{k'}) \in B'$ have a distance bounded above by $\mathcal{D}(B')$, we can bound the second term by

$$
\left| \frac{\sum_{B' \supseteq B} \sum_{k' \leq k} \mathbb{1}_{\left[B_h^{k'} = B'\right]} (r_h(X_h^{k'}, A_h^{k'}) - r_h(x, a))}{\sum_{B' \supseteq B} n_h^k(B')} \right| \leq \left| \frac{\sum_{B' \supseteq B} L_r \mathcal{D}(B') n_h^k(B')}{\sum_{B' \supseteq B} n_h^k(B')} \right|.
$$

Finally we bound the first term via the Azuma-Hoeffding inequality. Let $k_1, \ldots, k_t$ be the episodes in which $B$ and its ancestors were selected by the algorithm (i.e. $B_h^{k_i}$ is an ancestor of $B$); here $t = \sum_{B' \supseteq B} n_h^k(B')$. Under this definition the first term can be rewritten as

$$
\left| \frac{1}{t} \sum_{i=1}^{t} \left( R_h^{k_i} - r_h(X_h^{k_i}, A_h^{k_i}) \right) \right|
$$

Set $Z_i = R_h^{k_i} - r_h(X_h^{k_i}, A_h^{k_i})$. Clearly $Z_i$ is a martingale difference sequence with respect to the filtration $\hat{\mathcal{F}}_i = \mathcal{F}_{k_i+1}$. Moreover, as the sum of a martingale difference sequence is a martingale then for any $\tau \leq K$, $\sum_{i=1}^{\tau} Z_i$ is a martingale, where the difference in subsequent terms is bounded by 2. Thus by Azuma-Hoeffding's inequality we see that for a fixed $\tau \leq K$ that

$$
\mathbb{P}\left( \left| \frac{1}{\tau} \sum_{i=1}^{\tau} Z_i \right| \leq \sqrt{\frac{8 \log(2HK^2/\delta)}{\tau}} \right) \geq 1 - 2 \exp\left( -\frac{\tau \frac{8 \log(2HK^2/\delta)}{\tau}}{8} \right)
$$

$$= 1 - \frac{\delta}{2HK^2}.$$

When $\tau = t = \sum_{B' \supseteq B} n_h^k(B')$ the right hand side in the concentration is precisely

$$\sqrt{\frac{8\log(2HK^2/\delta)}{\sum_{B' \supseteq B} n_h^k(B')}}.$$

We then take a union bound over all steps $H$ and episodes $K$ and all $K$ possible values of $\tau$. Note that we do not need to union bound over the balls $B \in \mathcal{P}_h^k$ as the estimate of only one ball is changed per (step, episode) pair, i.e. $\hat{\mathbf{r}}_h^k(B)$ is changed for a single ball per episode. For all balls not selected, it inherits the concentration of the good event from the previous episode because its estimate does not change. Furthermore, even if ball $B$ is "split" in episode $k$, all of its children inherit the value of the parent ball, and thus also inherits the good event, so we still only need to consider the update for $B_h^k$ itself.

Combining these we have for any $(h, k) \in [H] \times [K]$ and ball $B \in \mathcal{P}_h^k$ such that $(x, a) \in B$

$$|\bar{\mathbf{r}}_h^k(B) - r_h(x, a)| \leq \sqrt{\frac{8\log(2HK^2/\delta)}{\sum_{B' \supseteq B} n_h^k(B')}} + \frac{\sum_{B' \supseteq B} L_r \mathcal{D}(B')n_h^k(B')}{\sum_{B' \supseteq B} n_h^k(B')}$$

$$\leq \sqrt{\frac{8\log(2HK^2/\delta)}{\sum_{B' \supseteq B} n_h^k(B')}} + 4L_r \mathcal{D}(B) = \text{RUCB}_h^k(B) \qquad \text{(by Lemma D.1).}$$

$\square$

## D.2    Concentration of Transition Estimates

Next we show concentration of the estimate of the transition kernel. We use recent work on bounding the Wasserstein distance between the empirical distribution and the true distribution for arbitrary measures [46]. The proof is split into two cases, where the cases define the relevant $\text{TUCB}_h^k(\cdot)$ used. We state the result here but defer the full proof to Appendix K.

**Lemma D.3.** *With probability at least $1 - 2\delta$ we have that for any $h, k \in [H] \times [K]$ and ball $B \in \mathcal{P}_h^k$ with $(x, a) \in B$ that*

$$d_W(\overline{\mathbf{T}}_h^k(\cdot \mid B), T_h(\cdot \mid x, a)) \leq \frac{1}{L_V} \text{TUCB}_h^k(B)$$

## D.3    Optimism Principle

The concentration bounds derived in Appendices D.1 and D.2 allow us to now demonstrate a principle of *optimism* for our value-function estimates.

**Lemma D.4.** *With probability at least $1 - 3\delta$, the following bounds are all simultaneously true for all $k, h \in [K] \times [H]$, and any partition $\mathcal{P}_h^k$*

$$\overline{\mathbf{Q}}_h^k(B) \geq Q_h^\star(x, a) \qquad \text{for all } B \in \mathcal{P}_h^k, \text{ and } (x, a) \in B$$
$$\widetilde{\mathbf{V}}_h^k(A) \geq V_h^\star(x) \qquad \text{for all } A \in \mathcal{S}(\mathcal{P}_h^k), \text{ and } x \in A$$
$$\overline{\mathbf{V}}_h^k(x) \geq V_h^\star(x) \qquad \text{for all } x \in \mathcal{S}$$

*Proof.* Recall the 'good events' in Lemmas D.2 and D.3 simultaneously hold with probability $1 - 3\delta$. Conditioned on this, we show the result by forwards induction on $k$ and backwards induction on $h$.

**Base Case** ($k = 0$): Recall the estimates are initialized as $\overline{\mathbf{Q}}_h^k(\cdot) = \overline{\mathbf{V}}_h^k(\cdot) = \widetilde{\mathbf{V}}_h^k(\cdot) = H - h + 1$. Now since all the rewards lie in $[0, 1]$, we have that $Q_h^\star(\cdot, \cdot)$ and $V_h^\star(\cdot)$ are upper bounded by $H - h + 1$, and so optimism holds for any $h \in [H]$.

**Induction** ($k-1 \to k$): We first consider $h = H+1$ and then proceed via backwards induction on $h$. For $h = H+1$, optimism holds as all quantities are zero. Next, for any $B \in \mathcal{P}_h^k$ and any $(x,a) \in B$,

$$
\begin{aligned}
\overline{\mathbf{Q}}_h^k(B) &= \overline{\mathbf{r}}_h^k(B) + \mathrm{RUCB}_h^k(B) + \mathbb{E}_{Y \sim \overline{\mathbf{T}}_h^k(\cdot|B)}[\overline{\mathbf{V}}_{h+1}^{k-1}(Y)] + \mathrm{TUCB}_h^k(B) \\
&\geq r_h(x,a) + \mathbb{E}_{Y \sim \overline{\mathbf{T}}_h^k(\cdot|B)}[V_{h+1}^\star(Y)] + \mathrm{TUCB}_h^k(B) \text{ (induction hypothesis and Lemma D.2)} \\
&\geq r_h(x,a) + \mathbb{E}_{Y \sim T_h(\cdot|x,a)}[V_{h+1}^\star(Y)] = Q_h^\star(x,a) \text{ (by Lemma D.3)}
\end{aligned}
$$

where we used the fact that $V_h^\star$ is $L_V$-Lipschitz continuous and that the difference in expectation over any Lipschitz function with respect to two different distributions is bounded above by the Wasserstein distance times the Lipschitz constant.

For any $A \in \mathcal{S}(\mathcal{P}_h^k)$ and any $x \in A$, if $\widetilde{\mathbf{V}}_h^k(A) = \widetilde{\mathbf{V}}_h^{k-1}(A)$ then optimism clearly holds by the induction hypothesis, and otherwise

$$
\begin{aligned}
\widetilde{\mathbf{V}}_h^k(A) &= \max_{B \in \mathcal{P}_h^k : \mathcal{S}(B) \supseteq A} \overline{\mathbf{Q}}_h^k(B) \\
&\geq \overline{\mathbf{Q}}_h^k(B^\star) \quad \text{for } (x, \pi_h^\star(x)) \in B^\star \\
&\geq Q_h^\star(x, \pi_h^\star(x)) = V_h^\star(x).
\end{aligned}
$$

For $x \in A \in \mathcal{S}(\mathcal{P}_h^k)$, and for the ball $B^\star \in \mathcal{P}_h^k$ that satisfies $(x, \pi_h^\star(x)) \in B^\star$, it must be that $\mathcal{S}(B^\star) \supseteq A$ because of the construction of the induced partition $\mathcal{S}(\mathcal{P}_h^k)$ via Eq. (3), the dyadic partitioning of $P_h^k$ which guarantees $\mathcal{S}(\mathcal{P}_h^k)$ is a partition, and the fact that $x \in \mathcal{S}(B^\star)$.

And lastly we have that for any $x \in \mathcal{S}$,

$$
\begin{aligned}
\overline{\mathbf{V}}_h^k(x) &= \widetilde{\mathbf{V}}_h^k(A) + L_V d_{\mathcal{S}}(x, \tilde{x}(A)) \quad \text{for some ball } A \in \mathcal{S}(\mathcal{P}_h^k) \\
&\geq V_h^\star(\tilde{x}(A)) + L_V d_{\mathcal{S}}(x, \tilde{x}(A)) \quad \text{by optimism of } \widetilde{\mathbf{V}}_h^k \\
&\geq V_h^\star(x) \quad \text{by Lipschitzness of } V_h^\star.
\end{aligned}
$$

Note that when a ball $B$ is split, it inherits all estimates from its parents, and thus it inherits the optimistic properties from its parents value functions as well. $\qquad\square$

## E  Sample-Path Regret Decomposition

We next outline our sample-path regret decomposition for one-step value iteration, which uses an idea adapted from Lemma 12 in [13]. We introduce the notation $\mathcal{S}(\mathcal{P}_h^k, x)$ to refer to the state-ball in $\mathcal{S}(\mathcal{P}_h^k)$ which contains the point $x$. The proofs of both results are deferred to Appendix K.

We begin by showing a result on the one-step difference between the estimated value of the policy and the true value of the policy employed. This critically uses the one-step value-iteration update in order to express the difference as a decreasing bounded process plus the sum of bonus terms.

**Lemma E.1.** *Consider any $h, k \in [H] \times [K]$, and any dyadic partition $\mathcal{P}_h^{k-1}$ of $\mathcal{S} \times \mathcal{A}$. Then the value update of* ADAMB *in the $k$'th episode in step $h$ is upper bounded by*

$$
\begin{aligned}
&\widetilde{\mathbf{V}}_h^{k-1}(\mathcal{S}(\mathcal{P}_h^{k-1}, X_h^k)) - V_h^{\pi^k}(X_h^k) \\
&\leq \sum_{h'=h}^{H} \mathbb{E}^{k-1}\left[\widetilde{\mathbf{V}}_{h'}^{k-1}(\mathcal{S}(\mathcal{P}_{h'}^{k-1}, X_{h'}^k)) - \widetilde{\mathbf{V}}_{h'}^k(\mathcal{S}(\mathcal{P}_{h'}^k, X_{h'}^k)) \mid X_h^k\right] \\
&\quad + \sum_{h'=h}^{H} \mathbb{E}^{k-1}\left[\overline{\mathbf{r}}_{h'}^k(B_{h'}^k) - r_{h'}(X_{h'}^k, A_{h'}^k) + \mathrm{RUCB}_{h'}^k(B_{h'}^k) \mid X_h^k\right] \\
&\quad + \sum_{h'=h}^{H} \mathbb{E}^{k-1}\left[\mathbb{E}_{x \sim \overline{\mathbf{T}}_{h'}^k(\cdot|B_{h'}^k)}[\overline{\mathbf{V}}_{h+1}^{k-1}(x)] - \mathbb{E}_{x \sim T_{h'}(\cdot|X_{h'}^k, A_{h'}^k)}[\overline{\mathbf{V}}_{h'+1}^{k-1}(x)] \mid X_h^k\right] \\
&\quad + \sum_{h'=h}^{H} \mathbb{E}^{k-1}\left[\mathrm{TUCB}_{h'}^k(B_{h'}^k) \mid X_h^k\right] + L_V \sum_{h'=h+1}^{H} \mathbb{E}^{k-1}\left[\mathcal{D}(B_{h'}^k) \mid X_h^k\right]
\end{aligned}
$$

The proof follows directly by expanding and substituting the various quantities. Moreover, using this lemma, we can further decompose the expected regret using the optimism principle defined in Appendix D.

**Lemma E.2.** *The expected regret for* ADAMB *can be decomposed as*

$$\mathbb{E}[R(K)] \lesssim \sum_{k=1}^{K}\sum_{h=1}^{H} \mathbb{E}\Big[\widetilde{\mathbf{V}}_h^{k-1}(\mathcal{S}(\mathcal{P}_h^{k-1}, X_h^k)) - \widetilde{\mathbf{V}}_h^k(\mathcal{S}(\mathcal{P}_h^k, X_h^k))\Big]$$
$$+ \sum_{h=1}^{H}\sum_{k=1}^{K} \mathbb{E}\big[2\mathrm{RUCB}_h^k(B_h^k)\big] + \sum_{h=1}^{H}\sum_{k=1}^{K} \mathbb{E}\big[2\mathrm{TUCB}_h^k(B_h^k)\big] + \sum_{k=1}^{K}\sum_{h=1}^{H} L_V \mathbb{E}\big[\mathcal{D}(B_h^k)\big].$$

This again follows from the definition of regret, and uses Lemma E.1. The proof is provided in Appendix K.

Next we analyze the first term in the regret decomposition by arguing it is bounded uniformly over all sample paths.

**Lemma E.3.** *Under* ADAMB*, along every sample trajectory we have*

$$\sum_{k=1}^{K}\sum_{h=1}^{H} \widetilde{\mathbf{V}}_h^{k-1}(\mathcal{S}(P_h^{k-1}, X_h^k)) - \widetilde{\mathbf{V}}_h^k(\mathcal{S}(\mathcal{P}_h^k, X_h^k)) \le H^2 \max_h |\mathcal{S}(\mathcal{P}_h^K)|.$$

*Proof.* We show a somewhat stronger bound, namely, that for every $h \in [H]$ we have

$$\sum_{k=1}^{K} \widetilde{\mathbf{V}}_h^{k-1}(\mathcal{S}(P_h^{k-1}, X_h^k)) - \widetilde{\mathbf{V}}_h^k(\mathcal{S}(\mathcal{P}_h^k, X_h^k)) \le (H - h + 1)|\mathcal{S}(\mathcal{P}_h^k)|$$

from which the claim then follows.

Recall that by definition, we have $\widetilde{\mathbf{V}}_h^{k-1}(\mathcal{S}(P_h^{k-1}, x))$ is non-decreasing $\forall x \in \mathcal{S}$. Now we can write

$$\sum_{k=1}^{K} \widetilde{\mathbf{V}}_h^{k-1}(\mathcal{S}(P_h^{k-1}, X_h^k)) - \widetilde{\mathbf{V}}_h^k(\mathcal{S}(\mathcal{P}_h^k, X_h^k)) \le \sum_{k=1}^{K} \sum_{A \in \mathcal{S}(\mathcal{P}_h^K)} \widetilde{\mathbf{V}}_h^{k-1}(A) - \widetilde{\mathbf{V}}_h^k(A)$$

where for a set $A \in \mathcal{S}(\mathcal{P}_h^K)$ which is not in $\mathcal{P}_h^k$ we let $\widetilde{\mathbf{V}}_h^k(A)$ be the $\widetilde{\mathbf{V}}_h^k(\cdot)$ value of the ball in $\mathcal{S}(\mathcal{P}_h^k)$ which contains $A$ (i.e., we set $\widetilde{\mathbf{V}}_h^{k-1}(A) = \widetilde{\mathbf{V}}_h^{k-1}(\mathcal{S}(\mathcal{P}_h^{k-1}, \tilde{x}(A)))$ and $\widetilde{\mathbf{V}}_h^k(A) = \widetilde{\mathbf{V}}_h^k(\mathcal{S}(\mathcal{P}_h^k, \tilde{x}(A)))$). Finally, we can change the order of summations to get

$$\sum_{k=1}^{K} \sum_{A \in \mathcal{S}(\mathcal{P}_h^K)} \widetilde{\mathbf{V}}_h^{k-1}(A) - \widetilde{\mathbf{V}}_h^k(A) = \sum_{A \in \mathcal{S}(\mathcal{P}_h^K)} \sum_{k=1}^{K} \widetilde{\mathbf{V}}_h^{k-1}(A) - \widetilde{\mathbf{V}}_h^k(A)$$
$$= \sum_{A \in \mathcal{S}(\mathcal{P}_h^K)} \widetilde{\mathbf{V}}_h^0(A) - \widetilde{\mathbf{V}}_h^K(A)$$
$$\le (H - h + 1)|\mathcal{S}(\mathcal{P}_h^k)|.$$

$\square$

# F   Adversarial Bounds for Counts over Partitions

Recall that the splitting threshold is defined to be: split a ball once we have that $n_h^k(B) + 1 \ge n_+(B)$ where $n_+(B) = \phi 2^{\gamma \ell(B)}$ for parameters $\phi$ and $\gamma$. As the splitting threshold only depends on the level of the ball in the partition, we abuse notation and use $n_+(\ell) = \phi 2^{\gamma \ell}$ to denote the threshold number of samples needed by the splitting rule to trigger splitting a ball at level $\ell$. We first provide a general bound for counts over any partition $\mathcal{P}_h^k$.

**Lemma F.1.** *Consider any partition $\mathcal{P}_h^k$ for any $k \in [K], h \in [H]$ induced under* ADAMB *with splitting thresholds $n_+(\ell)$, and consider any 'penalty' vector $\{a_\ell\}_{\ell \in \mathbb{N}_0}$ that satisfies $a_{\ell+1} \geq a_\ell \geq 0$ and $2a_{\ell+1}/a_\ell \leq n_+(\ell)/n_+(\ell-1)$ for all $\ell \in \mathbb{N}_0$. Define $\ell^\star = \inf\{\ell \mid 2^{d(\ell-1)}n_+(\ell-1) \geq k\}$. Then*

$$\sum_{\ell=0}^\infty \sum_{B \in \mathcal{P}_h^k : \ell(B)=\ell} a_\ell \leq 2^{d\ell^\star} a_{\ell^\star}$$

*Proof.* For $\ell \in \mathbb{N}_0$, let $x_\ell$ denote the number of active balls at level $\ell$ in $\mathcal{P}_h^k$. Then $\sum_{B \in \mathcal{P}_h^k : \ell(B)=\ell} a_\ell = \sum_{\ell \in \mathbb{N}_0} a_\ell x_\ell$. Now we claim that under any partition, this sum can be upper bound via the following linear program (LP):

$$\text{maximize:} \quad \sum_{\ell=0}^\infty a_\ell x_\ell$$
$$\text{subject to:} \quad \sum_\ell 2^{-\ell d} x_\ell \leq 1 \,,$$
$$\sum_\ell n_+(\ell-1) 2^{-d} x_\ell \leq k \,,$$
$$x_\ell \geq 0 \,\forall\, \ell$$

The first constraint arises via the Kraft-McMillan inequality for prefix-free codes (see Chapter 5 in [9]): since each node can have at most $D = 2^d$ (where $d = d_\mathcal{S} + d_\mathcal{A}$) children by definition of the covering dimension, the partition created can be thought of as constructing a prefix-free code on a $D$-ary tree. The second constraint arises via a conservation argument on the number of samples; recall that $n_+(B)$ is the minimum number of samples required before $B$ is split into $2^d$ children – an alternate way to view this is that each ball at level $\ell$ requires a 'sample cost' of $n_+(\ell-1)/2^d$ unique samples in order to be created. The sum of this sample cost over all active balls is at most the number of samples $k$.

Next, via LP duality, we get that the optimal value for this program is upper bounded by $\alpha + \beta$ for any $\alpha$ and $\beta$ such that:

$$2^{-\ell d}\alpha + n_+(\ell-1)2^{-d}\beta \geq a_\ell \quad \forall \ell \in \mathbb{N}_0$$
$$\alpha, \beta \geq 0.$$

Recall the definition of $\ell^\star = \inf\{\ell \mid 2^{d(\ell-1)}n_+(\ell-1) \geq k\}$ and consider

$$\hat{\alpha} = \frac{2^{d\ell^\star} a_{\ell^\star}}{2} \quad \hat{\beta} = \frac{2^d a_{\ell^\star}}{2n_+(\ell^\star-1)}.$$

We claim that this pair satisfies the constraint that $2^{-\ell d}\hat{\alpha} + n_+(\ell-1)2^{-d}\hat{\beta} \geq a_\ell$ for any $\ell$, and hence by weak duality we have that

$$\sum_{B \in \mathcal{P}_h^k : \ell(B)=\ell} a_\ell \leq \hat{\alpha} + \hat{\beta} \leq 2\hat{\alpha} = 2^{d\ell^\star} a_{\ell^\star}.$$

To verify the constraints on $(\hat{\alpha}, \hat{\beta})$ we check it by cases. First note that for $\ell = \ell^\star$, we have $2^{-\ell^\star d}\hat{\alpha} + n_+(\ell^\star-1)2^{-d}\hat{\beta} = a_{\ell^\star}$.

Next, for any $\ell < \ell^\star$, note that $2^{-\ell d} \geq 2^{-(\ell^\star-1)d} > 2 \cdot (2^{-\ell^\star d})$, and hence $2^{-\ell d}\hat{\alpha} \geq 2 \cdot (2^{-\ell^\star d}\hat{\alpha}) = a_{\ell^\star} \geq a_\ell$ by construction of the penalty vector.

Similarly, for any $\ell > \ell^\star$, we have by assumption on the costs and $n_+(\ell)$ that

$$\frac{n_+(\ell-1)}{a_\ell} \geq \frac{2^{\ell-\ell^\star}n_+(\ell^\star-1)}{a_{\ell^\star}} \geq 2\frac{n_+(\ell^\star-1)}{a_{\ell^\star}}.$$

Then we get by plugging in our value of $\hat{\beta}$ that

$$n_+(\ell-1)2^{-d}\hat{\beta} = \frac{a_{\ell^\star}n_+(\ell-1)}{2n_+(\ell^\star-1)} \geq a_\ell$$

This verifies the constraints for all $\ell \in \mathbb{N}_0$. $\qquad\square$

Note also that in the above proof, we actually use the condition $2a_{\ell+1}/a_\ell \leq n_+(\ell)/n_+(\ell-1)$ for $\ell \geq \ell^\star$; we use this more refined version in Corollary F.3 below.

### F.1 Worst-Case Partition Size and Sum of Bonus Terms

One immediate corollary of Lemma F.1 is a bound on the size of the partition $|\mathcal{P}_h^k|$ for any $h, k$.

**Corollary F.2.** *For any $h$ and $k$ we have that*

$$|\mathcal{P}_h^k| \leq 4^d \left(\frac{k}{\phi}\right)^{\frac{d}{d+\gamma}}$$

*and that*

$$\ell^\star \leq \frac{1}{d+\gamma}\log_2(k/\phi) + 2.$$

*Proof.* Note that the size of the partition can be upper bounded by the sum where we take $a_\ell = 1$ for every $\ell$. Clearly this satisfies the requirements of Lemma F.1. Moreover, using the definition of $\ell^\star$ we have that $2^{d(\ell^\star-2)}n_+(\ell^\star-2) \leq k$ as otherwise $\ell^\star - 1$ would achieve the infimum. Taking this equation and plugging in the definition of $n_+(\ell)$ by the splitting rule yields that

$$\ell^\star \leq \frac{1}{d+\gamma}\log_2\left(\frac{k}{\phi}\right) + 2.$$

Then by plugging this in we get that

$$|\mathcal{P}_h^k| \leq 2^{d\ell^\star} \leq 2^{\frac{d}{d+\gamma}\log_2(k/\phi)+2d} = 4^d\left(\frac{k}{\phi}\right)^{d/(d+\gamma)}.$$

$\qquad\square$

In other words, the worst case partition size is determined by a *uniform* scattering of samples, wherein the entire space is partitioned up to equal granularity (in other words, a uniform $\epsilon$-net).

More generally, we can use Lemma F.1 to bound various functions of counts over balls in $\mathcal{P}_h^k$. In Appendix J we use this to bound various terms in our regret expansion.

**Corollary F.3.** *For any $h \in [H]$, consider any sequence of partitions $\mathcal{P}_h^k, k \in [K]$ induced under* ADAMB *with splitting thresholds $n_+(\ell) = \phi 2^{\gamma\ell}$. Then, for any $h \in [H]$ we have:*

- *For any $\alpha, \beta \geq 0$ s.t. $\alpha \leq 1$ and $\alpha\gamma - \beta \geq 1$, we have*

$$\sum_{k=1}^K \frac{2^{\beta\ell(B_h^k)}}{\left(n_h^k(B_h^k)\right)^\alpha} = O\left(\phi^{\frac{-(d\alpha+\beta)}{d+\gamma}}K^{\frac{d+(1-\alpha)\gamma+\beta}{d+\gamma}}\right)$$

- *For any $\alpha, \beta \geq 0$ s.t. $\alpha \leq 1$ and $\alpha\gamma - \beta/\ell^\star \geq 1$ (where $\ell^\star = 2 + \frac{1}{d+\gamma}\log_2\left(\frac{K}{\phi}\right)$), we have*

$$\sum_{k=1}^K \frac{\ell(B_h^k)^\beta}{\left(n_h^k(B_h^k)\right)^\alpha} = O\left(\phi^{\frac{-d\alpha}{d+\gamma}}K^{\frac{d+(1-\alpha)\gamma}{d+\gamma}}(\log_2 K)^\beta\right)$$

The proof of both the inequalities follows from a direct application of Lemma F.1 (and in fact, using the same $\ell^\star$ as in Corollary F.2), after first rewriting the summation over balls in $\mathcal{P}_h^k$ as a summation over active balls in $\mathcal{P}_h^K$. The complete proof is deferred to Appendix K.

**Algorithm 2** Model-Based Reinforcement Learning with Adaptive Partitioning (ADAMB)

1: **procedure** ADAMB($\mathcal{S}, \mathcal{A}, \mathcal{D}, H, K, \delta$)
2:    Initialize partitions $\mathcal{P}_h^0 = \mathcal{S} \times \mathcal{A}$ for $h \in [H]$, estimates $\overline{\mathbf{Q}}_h^0(\cdot) = \overline{\mathbf{V}}_h^k(\cdot) = H - h + 1$
3:    **for** each episode $k \leftarrow 1, \ldots K$ **do**
4:        Receive starting state $X_1^k$
5:        **for** each step $h \leftarrow 1, \ldots, H$ **do**
6:            Observe $X_h^k$ and determine $\text{RELEVANT}_h^k(X_h^k) = \{B \in \mathcal{P}_h^{k-1} \mid X_h^k \in B\}$
7:            Greedy selection rule: pick $B_h^k = \arg\max_{B \in \text{RELEVANT}_h^k(X_h^k)} \overline{\mathbf{Q}}_h^{k-1}(B)$
8:            Play action $A_h^k = \tilde{a}(B_h^k)$ associated with ball $B_h^k$; receive $R_h^k$ and transition to $X_{h+1}^k$
9:            Update counts for $n_h^k(B_h^k), \hat{\mathbf{r}}_h^k(B_h^k)$, and $\hat{\mathbf{T}}_h^k(\cdot \mid B_h^k)$ via:
10:            $n_h^k(B_h^k) \leftarrow n_h^{k-1}(B_h^k) + 1$
11:            $\hat{\mathbf{r}}_h^k(B_h^k) \leftarrow \frac{(n_h^k(B_h^k) - 1)\hat{\mathbf{r}}_h^k(B_h^k) + R_h^k}{n_h^k(B_h^k)}$
12:            $\hat{\mathbf{T}}_h^k(A \mid B_h^k) = \frac{(n_h^k(B_h^k) - 1)\hat{\mathbf{T}}_h^{k-1}(A \mid B_h^k) + \mathbb{1}_{\left[X_{h+1}^k \in A\right]}}{n_h^k(B_h^k)}$ for $A \in \square_{\ell(B_h^k)}$
13:            **if** $n_h^k(B_h^k) + 1 \geq n_+(B_h^k)$ **then** REFINE PARTITION$(B_h^k)$
            COMPUTE ESTIMATES$(B_h^k, R_h^k, X_{h+1}^k)_{h=1}^H$
14: **procedure** REFINE PARTITION($B, h, k$)
15:    Construct $\mathcal{P}(B) = \{B_1, \ldots, B_{2^d}\}$ a $2^{-(\ell(B)+1)}$-dyadic partition of $B$
16:    Update $\mathcal{P}_h^k = \mathcal{P}_h^{k-1} \cup \mathcal{P}(B) \setminus B$
17:    For each $B_i$, initialize $n_h^k(B_i) = 0$, $\hat{\mathbf{r}}_h^k(B_i) = 0$ and $\hat{\mathbf{T}}_h^k(B_i) = 0$
18: **procedure** COMPUTE ESTIMATES$((B_h^k, R_h^k, X_{h+1}^k)_{h=1}^H)$
19:    **for** each $h \leftarrow 1, \ldots H$ and $B \in \mathcal{P}_h^k$ **do**
20:        Construct $\overline{\mathbf{r}}_h^k(B)$ and $\overline{\mathbf{T}}_h^k(B)$ by
21:        $\overline{\mathbf{r}}_h^k(B) = \frac{\sum_{B' \supseteq B} \hat{\mathbf{r}}_h^k(B') n_h^k(B')}{\sum_{B' \supseteq B} n_h^k(B')}$
22:        $\overline{\mathbf{T}}_h^k(A \mid B) = \frac{\sum_{B' \supseteq B} \sum_{A' \in \square_{\ell(B')}: A \subset A'} 2^{-d_{\mathcal{S}}(\ell(B') - \ell(B))} n_h^k(B') \hat{\mathbf{T}}_h^k(A' \mid B')}{\sum_{B' \supseteq B} n_h^k(B')}$ for $A \in \square_{\ell(B)}$
23:        Solve for $\overline{\mathbf{V}}_{h+1}^{k-1}(A)$ for every $A \in \square_{\ell(B)}$ by

$$\overline{\mathbf{V}}_{h+1}^{k-1}(A) = \min_{A' \in \mathcal{S}(\mathcal{P}_{h+1}^{k-1})} \widetilde{\mathbf{V}}_{h+1}^{k-1}(A') + L_V \mathcal{D}_{\mathcal{S}}(\tilde{x}(A), \tilde{x}(A'))$$

24:        Set $\overline{\mathbf{Q}}_h^k(B) = \overline{\mathbf{r}}_h^k(B) + \text{RUCB}_h^k(B) + \mathbb{E}_{A \sim \overline{\mathbf{T}}_h^k(\cdot \mid B)}[\overline{\mathbf{V}}_{h+1}^{k-1}(A)] + \text{TUCB}_h^k(B)$
25:    **for** each $h \leftarrow 1, \ldots H$ and $A \in \mathcal{S}(\mathcal{P}_h^k)$ **do**
26:        Set $\widetilde{\mathbf{V}}_h^k(A) = \min\{\widetilde{\mathbf{V}}_h^{k-1}(A), \max_{B \in \mathcal{P}_h^k: \mathcal{S}(B) \supseteq A} \overline{\mathbf{Q}}_h^k(B)\}$

## G  Algorithm and Implementation

In this section we give the full pseudocode for implementing the algorithm, discuss the run-time and space complexity, and provide some discussion on other heuristic approaches to discretization.

### G.1  Implementation and Running Time

Here we briefly discuss the oracle assumptions required for implementing the algorithm, and analyze the run-time and storage complexity.

**Oracle Assumptions**: There are three main oracle assumptions needed to execute the algorithm. In line 14 of Algorithm 2 we need access to a "covering oracle" on the metric space. This oracle takes as input a ball $B \subset \mathcal{S} \times \mathcal{A}$ and outputs an $r$-covering of $B$. This subroutine is easy in many metrics of interest (e.g. the Euclidean norm or any equivalent norms in $\mathbb{R}^d$) by just splitting each of the principle dimensions in half. Second, we need to be able to compute $\mathcal{S}(B)$ for any $B \in \mathcal{S} \times \mathcal{A}$. As our

algorithm is maintaining a dyadic partition of the space, this subroutine is also simple to implement as each ball $B$ is of the form $\mathcal{S}(B) \times \mathcal{S}(A)$ and so the algorithm can store the two components separately. Lastly, we require computing $\text{RELEVANT}_h^k(X)$. By storing the partition as a tree, this subroutine can be implementing by traversing down the tree and checking membership at each step. See the Github repository at `https://github.com/seanrsinclair/AdaptiveQLearning` for examples of implementing these methods. Storing the discretization as a hash function would allow some of these access steps to be implemented in $O(1)$ time, with the downside being that splitting a region has a larger computational requirement.

**Storage Requirements**: The algorithm maintains a partition $\mathcal{P}_h^k$ of $\mathcal{S}_h \times \mathcal{A}_h$ for every $h$, and the respective induced partition $\mathcal{S}(\mathcal{P}_h^k)$ whose size is trivially upper bounded by the size of the total partition. Each element $B \in \mathcal{P}_h^k$ maintains four estimates. The first three ($n_h^k(B)$, $\hat{\mathbf{r}}_h^k(B)$, and $\overline{\mathbf{Q}}_h^k(B)$) are linear with respect to the size of the partition. The last one, $\hat{\mathbf{T}}_h^k(\cdot \mid B)$ has size $|\square_{\ell(B)}| \lesssim O(2^{d_{\mathcal{S}} \ell(B)})$. Moreover, the algorithm also maintains estimate $\widetilde{\mathbf{V}}_h^k(\cdot)$ over $\mathcal{S}(\mathcal{P}_h^k)$. Clearly we have that the worst-case storage complexity arises from maintaining estimates of the transition kernels over each region in $\mathcal{P}_h^k$. Thus we have that the total storage requirement of the algorithm is bounded above by

$$\sum_{h=1}^{H} \sum_{B \in \mathcal{P}_h^K} 2^{d_{\mathcal{S}} \ell(B)}.$$

Utilizing Lemma F.1 with $a_\ell = 2^{d_{\mathcal{S}} \ell}$ we find that the sum is bounded above by

$$\sum_{h=1}^{H} \sum_{B \in \mathcal{P}_h^K} 2^{d_{\mathcal{S}} \ell(B)} \leq \sum_{h=1}^{H} 2^{d\ell^\star} a_{\ell^\star}$$

$$\lesssim H K^{\frac{d+d_{\mathcal{S}}}{d+\gamma}}.$$

Plugging in the definition of $\gamma$ from the splitting rule yields the results in Table 1.

**Run-Time**: We assume that the oracle access discussed occurs in constant time. The inner loop of Algorithm 2 has four main steps. Finding the set of relevant balls for a given state can be implemented in $\log_d(|\mathcal{P}_h^k|)$ time by traversing through the tree structure. Updating the estimates and refining the partition occur in constant time by assumption on the oracle. Lastly we need to update the estimates for $\overline{\mathbf{Q}}_h^k$ and $\overline{\mathbf{V}}_h^k$. Since the update only needs to happen for a constant number of regions (as only one ball is selected per step episode pair) the dominating term arises from computing the expectation over $\overline{\mathbf{T}}_h^k(\cdot \mid B_h^k)$. Noting that the support of the distribution is $|\square_{\ell(B_h^k)}| = 2^{d_{\mathcal{S}} \ell(B_h^k)}$ the total run-time of the algorithm is upper bounded by

$$\sum_{h=1}^{H} \sum_{k=1}^{K} 2^{d_{\mathcal{S}} \ell(B_h^k)}.$$

Rewriting the sum we have

$$\sum_{h=1}^{H} \sum_{k=1}^{K} 2^{d_{\mathcal{S}} \ell(B_h^k)} \leq \sum_{h=1}^{H} \sum_{\ell \in \mathbb{N}} \sum_{B \in \mathcal{P}_h^K : \ell(B) = \ell} 2^{d_{\mathcal{S}} \ell} \sum_{k \in [K] : B_h^k = B} 1$$

$$\lesssim \sum_{h=1}^{H} \sum_{\ell \in \mathbb{N}} \sum_{B \in \mathcal{P}_h^K : \ell(B) = \ell} 2^{d_{\mathcal{S}} \ell} n_+(B)$$

$$\lesssim \sum_{h=1}^{H} \sum_{\ell \in \mathbb{N}} \sum_{B \in \mathcal{P}_h^K : \ell(B) = \ell} 2^{d_{\mathcal{S}} \ell} \phi 2^{\gamma \ell}.$$

Utilizing Lemma F.1 with $a_\ell = 2^{(d_{\mathcal{S}} + \gamma)\ell}$ we find that the sum is bounded above by $H\phi 2^{d\ell^\star} a_{\ell^\star} \lesssim H K^{1 + \frac{d_{\mathcal{S}}}{d+\gamma}}$. Plugging in $\gamma$ from the splitting rule yields the result in Table 1.

Figure 3: Comparison of the observed rewards, size of the partition, and resulting discretization for the four algorithms on the one-dimensional oil problem with no noise and survey function $f(x,a) = 1 - (x - .7)^2$ and $\alpha = 1$. The colours correspond to the estimated $\overline{\mathbf{Q}}_h^k(B)$ values, where green corresponds to a larger estimated $Q$ value.

**Monotone Increasing Run-Time and Storage Complexity**: The run-time and storage complexity guarantees presented are monotonically increasing with respect to the number of episodes $K$. However, to get sublinear minimax regret in a continuous setting for nonparametric Lipschitz models, the model complexity must grow over episodes. In practice, one would run ADAMB until running out of space - and our experiments show that ADAMB uses resources (storage and computation) much better than a uniform discretization. We are not aware of any storage-performance lower bounds, so this is an interesting future direction.

# H  Experiments

In this section we give full details on the experiments and simulations performed. For full code implementation and more results please see the Github repository at `https://github.com/seanrsinclair/AdaptiveQLearning`.

For the experiments we were motivated to work on ambulance routing and the oil discovery problem as efficient algorithms for reinforcement learning in operations tasks is still largely unexplored. It is, however, a very natural objective in designing systems where agents must learn to navigate an uncertain environment to maximize their utility. These experiments can have broader implications in planning effective public transportation, stationing medics at events, or even cache management (which technically is a discrete measurement, but is most usefully talked about in a continuous manner due to the magnitude of memory units).

The main objective for continuous space problems in reinforcement learning is to meaningfully store continuous data in a discrete manner while still producing optimal results in terms of performance and reward. We find that the oil discovery and ambulance routing problems are simple enough that we can realistically produce uniform discretization benchmarks to test our adaptive algorithm against.

Figure 4: Comparison of the observed rewards, size of the partition, and resulting discretization for the four algorithms on the one-dimensional oil problem in the "laplace-reward" setting with $\alpha = 1$ and $\lambda = 10$. The colours correspond to the estimated $\overline{\mathbf{Q}}_h^k(B)$ values, where green corresponds to a larger estimated $Q$ value.

At the same time, they provide interesting continuous space scenarios that suggest there can be substantial improvements when using adaptive discretization in real world problems. The ambulance routing problem also allows us to naturally increase the state and action space dimensionality by adding another ambulance and consequently test our algorithms in a slightly more complex setting. In particular, we compare ADAPTIVE Q-LEARNING[38], MODEL-FREE $\epsilon$-NET[42], ADAMB (Algorithm 1), and a $\epsilon$-net variant of UCBVI [5]. We refer to the simulations as ADAQL, EPSILONQL, ADAMB, and EPSILONMB respectively in the figures.

## H.1 Oil Discovery

This problem, adapted from [24] is a continuous variant of the "Grid World" environment. It comprises of an agent surveying a 1D map in search of hidden "oil deposits". The world is endowed with an unknown survey function which encodes the probability of observing oil at that specific location. For agents to move to a new location they pay a cost proportional to the distance moved, and surveying the land produces noisy estimates of the true value of that location. In addition, due to varying terrain the true location the agent moves to is perturbed as a function of the state and action.

To formalize the problem, here the state space $\mathcal{S} = [0, 1]$ and action space $\mathcal{A} = [0, 1]$, where the product space is endowed with the $\ell_\infty$ metric. The reward function is defined as

$$r_h(x, a) = \max\{\min\{f_h(x, a) - \alpha|x - a| + \epsilon, 1\}, 0\}$$

where $f_h(x, a)$ is the survey function, corresponding to the probability of observing an oil deposit at that specific location and $\alpha$ is a parameter used to govern the transportation cost and $\epsilon$ is independent Gaussian noise. The transition function is defined as

$$\mathbb{P}_h(\cdot \mid x, a) = \max\{\min\{\delta_a + N(0, \sigma_h(x, a)^2), 1\}, 0\}$$

Figure 5: Plot of discretized approximation to $Q_h^\star$ for the one-dimensional oil problem in the "laplace" ($\lambda = 10$) and "quadratic" ($\lambda = 1$) setting. Note that here the $x$-axis corresponds to states and the $y$-axis to actions. The colour corresponds to the true $Q_2^\star$ value where green corresponds to a larger value.

where again we have truncated the new state to fall within $[0, 1]$ and the noise function $\sigma_h(x, a)$ allows for varying terrain in the environment leading to noisy transitions. Clearly if we take $\sigma_h(x, a) = 0$ we recover deterministic transitions from a state $x$ taking action $a$ to the next state being $a$.

We performed three different simulations, where we took $f_h(x, a)$ and $\sigma_h(x, a)$ as follows:

**Noiseless Setting**: $\sigma_h(x, a) = 0$ and the reward function $f_h(x, a) = 1 - \lambda(x - c)^2$ or $f_h(x, a) = 1 - e^{-\lambda|x-c|}$ where $c$ is the location of the oil deposit and $\lambda$ is a tunable parameter.

**Sparse-Reward Setting**: $\sigma_h(x, a) = .025(x + a)^2$ and the survey function is defined via:

$$f_h(x, a) = \begin{cases} \frac{1}{h}\left(1 - e^{-\lambda|x-.5|}\right) & h = 1 \\ \frac{1}{h}\left(1 - e^{-\lambda|x-.25|}\right) & h = 2 \\ \frac{1}{h}\left(1 - e^{-\lambda|x-.5|}\right) & h = 3 \\ \frac{1}{h}\left(1 - e^{-\lambda|x-.75|}\right) & h = 4 \\ \frac{1}{h}\left(1 - e^{-\lambda|x-1|}\right) & h = 5 \end{cases}$$

**Discussion.** We can see in Figure 4 and in Figure 3 that the EPSILONQL algorithm takes much longer to learn the optimal policy than its counterpart EPSILONMB and both model-based algorithms. Seeing improved performance of model-based algorithms over model-free with a uniform discretization is unsurprising, as it is folklore that model-based algorithms perform better than model-free in discrete spaces.

The two adaptive algorithms also offer a significantly smaller partition size than the corresponding uniform discretization. After comparing the adaptive algorithms' discretization of estimated $Q$-values with the true $Q_2^\star$-values in the state-action space, we find that the adaptive algorithms closely approximate the underlying $Q$ function (see Figure 5). This is as the adaptive algorithms maintain a much finer partition in regions of the space where the underlying $Q^\star$ values are large, thus reducing unnecessary exploration (hence reducing the size of the partition), and allowing the algorithm to learn the optimal policy faster (low regret). This demonstrates our algorithms' effectiveness in allocating space only to where it is advantageous to exploit more rigorously. Interestingly, we see that the model-free algorithm is able to more closely resemble the underlying $Q^\star$ values than the model-based algorithm. This affirms recent work showing instance-dependent bounds for model-free algorithms [7], and our discussion on the drawback of model-based algorithms storing estimates of the transition kernel.

Moreover, in the attached github repository we include code testing the necessity of the splitting rule in the model based algorithm being of the form $n_+(B) = \phi 2^{\gamma \ell(B)}$ for various forms of $\gamma$. While the theoretical results indicate that $\gamma = d_\mathcal{S}$ is necessary for convergence, experimentally we see that $\gamma = 2$ matching the model-free algorithm also suffices.

Figure 6: Comparison of the observed rewards, size of the partition, and resulting discretization for the four algorithms on the one ambulance problem with $\alpha = 1$ and arrivals $\mathcal{F}_h = \text{Beta}(5, 2)$. The colors correspond to the estimated $\overline{\mathbf{Q}}_h^k(B)$ values, where green corresponds to a larger estimated $Q$ value.

Figure 7: Comparison of the observed rewards and the size of the partition for the four algorithms on the two ambulance problem with $\alpha = 1$ and arrivals $\mathcal{F}_h = \text{Beta}(5, 2)$. We ommit confidence bars in this plot to help with readability.

## H.2 Ambulance Routing

This problem is a widely studied question in operations research and control, and is closely related to the $k$-server problem. A controller positions a fleet of $k$ ambulances over $H$ time periods, so as to minimize the transportation costs and time to respond to incoming patient requests. In our setting, the controller first chooses locations to station the ambulances. Next, a single request is realized drawn

from a fixed $h$-dependent distribution. Afterwards, one ambulance is chosen to travel to meet the demand, while other ambulances can re-position themselves.

Here the state space $\mathcal{S} = [0,1]^k$ and action space $\mathcal{A} = [0,1]^k$ where $k$ is the number of ambulances, and the product space is endowed with the $\ell_\infty$ metric. The reward function and transition is defined as follows. First, all ambulances travel from their initial state $x_i$ to their desired location $a_i$, paying a transportation cost to move the ambulance to location $a_i$. Afterwards, a patient request location $p_h \sim \mathcal{F}_h$ is drawn i.i.d. from a fixed distribution $\mathcal{F}_h$. The closest ambulance to $p_h$ is then selected to serve the patient, i.e. let

$$i^\star = \operatorname*{argmin}_{i \in [k]} |a_i - p_h|$$

denote the ambulance traveling to serve the patient. The rewards and transitions are then defined via:

$$x_i^{new} = \begin{cases} a_i & i \neq i^\star \\ p_h & i = i^\star \end{cases}$$

$$r_h(x,a) = 1 - \left( \frac{\alpha}{k} \|x - a\|_1 + (1-\alpha)|a_{i^\star} - p_h| \right)$$

where $\alpha$ serves as a tunable parameter to relate the cost of initially traveling from their current location $x$ to the desired location $a$, and the cost of traveling to serve the new patient $p_h$. We tested values of $\alpha$ in $\{0, .25, 1\}$ where $\alpha = 1$ corresponds to only penalizing the ambulances for traveling to the initial location, $\alpha = 0$ only penalizes agents for traveling to serve the patient, and $\alpha = 0.25$ interpolates between these two settings.

For the arrival distributions, we took $\mathcal{F}_h = \text{Beta}(5,2)$, $\mathcal{F}_h = \text{Uniform}(0,1)$ and a time-varying arrival distribution:

$$\mathcal{F}_h = \begin{cases} \text{Uniform}(0,.25) & h = 1 \\ \text{Uniform}(.25,.3) & h = 2 \\ \text{Uniform}(.3,.5) & h = 3 \\ \text{Uniform}(.5,.6) & h = 4 \\ \text{Uniform}(.6,.65) & h = 5 \end{cases}$$

**Discussion.** In both the single ambulance case (Figure 6) and two-ambulance (Figure 7) we see that the uniform discretization algorithms are outperformed by their adaptive counterparts. Unsurprisingly, the partition size of both adaptive algorithms is significantly smaller than the epsilon algorithms, with ADAQL being slightly more efficient. We also see that both adaptive algorithms perform similarly in terms of rate of convergence and observed rewards for both the two and one ambulance problem. Again, this is because the adaptive algorithms maintain a finer partition in regions of the space where the underlying $Q^\star$ values are large, thus reducing the size of the partition and leading the algorithm to learn the optimal policy faster. When looking at the resulting discretizations in Figure 6 we observe similar results to the oil problem, where the model-free algorithm exhibits a finer partition than the model-based algorithm.

# I  Experiment Setup and Computing Infrastructure

**Experiment Setup**: Each experiment was run with 200 iterations where the relevant plots are taking the mean and a standard-normal confidence interval of the related quantities. We picked a fixed horizon of $H = 5$ and ran it to $K = 2000$ episodes. As each algorithm uses bonus terms of the form $c/\sqrt{t}$ where $t$ is the number of times a related region has been visited, we tuned the constant $c$ separately for each algorithm (for $c \in [.001, 10]$) and plot the results on the performance of the algorithm for the best constant $c$.

**Fixed Discretization UCBVI**: We bench marked our adaptive algorithm against a fixed-discretization model-based algorithm with full and one-step planning. In particular, we implemented UCBVI from [5] using a fixed discretization of the state-action space. The algorithm takes as input a parameter $\epsilon$ and constructs an $\epsilon$-covering of $\mathcal{S}$ and $\mathcal{A}$ respectively. It then runs the original UCBVI algorithm over

Table 3: Comparison of the average running time (in seconds) of the four different algorithms considered in the experimental results: ADAMB (Algorithm 1), ADAPTIVE Q-LEARNING [38], NET-BASED Q-LEARNING [42], and a FIXED DISCRETIZATION UCBVI [5].

| Problem | ADAMB | ADAQL | EPSILONQL | EPSILONMB |
|---|---|---|---|---|
| 1 AMBULANCE | 8.07 | 0.90 | 1.10 | 16.59 |
| 2 AMBULANCES | 22.92 | 1.57 | 9.54 | 90.92 |
| OIL PROBLEM | 5.63 | 1.31 | 2.21 | 20.27 |

this discrete set of states and actions. The only difference is that when visiting a state $x$, as feedback to the algorithm, the agent snaps the point to its closest neighbour in the covering.

UCBVI has a regret bound of $H^{3/2}\sqrt{SAK} + H^4 S^2 A$ where $S$ and $A$ are the size of the state and action spaces. Replacing these quantities with the size of the covering, we obtain

$$H^{3/2}\sqrt{\epsilon^{-d_\mathcal{S}}\epsilon^{-d_\mathcal{A}}K} + H^4\epsilon^{-2d_\mathcal{S}}\epsilon^{-d_\mathcal{A}}.$$

A rough calculation also shows that the discretization error is proportional to $HLK\epsilon$. Tuning $\epsilon$ so as to balance these terms, we find that the regret of the algorithm can be upper bounded by

$$LH^2 K^{2d/(2d+1)}.$$

The major difference in this approach versus a uniform discretization of a model-free algorithm (e.g. [42]) is that in model-based algorithms the lower-order terms scale quadratically with the size of the state space. In tabular settings, this term is independent of the number of episodes $K$. However, in continuous settings the discretization depends on the number of episodes $K$ in order to balance the approximation error from discretizing the space uniformly. See [11] for a discussion on this dependence.

Obtaining better results for model-based algorithms with uniform discretization requires better understanding the complexity in learning the transition model, which ultimately leads to the terms which depend on the size of the state space. The theoretical analysis of the concentration inequalities for the transitions in Appendix D are min-max, showing that worst case dependence on the dimension of the state space is inevitable. However, potential approaches could instead model bonuses over the value function instead of the transitions would lead to better guarantees [3]. Our concentration inequalities on the transition kernels is a first-step at understanding this feature in continuous settings.

**Computing Infrastructure and Run-Time**: The experiments were conducted on a personal computer with an AMD Ryzen 5 3600 6-Core 3.60 GHz processor and 16.0GB of RAM. No GPUs were harmed in these experiments. The average computation time for running a single simulation of an algorithm is listed in Table 3. As different hyperparameter settings result in similar run-times, we only show the three major simulations conducted with fixed bonus scaling $c = 1$. As to be expected, the adaptive algorithms ran much faster than their uniform discretization counterparts. Moreover, the model-free methods have lower running time than the model-based algorithms. These results mimic the run-time and space complexity discussed in Table 1.

## J  Regret Derivation

In this section we combine all of the previous results to derive a final regret bound. We first provide a bound on the expected regret for ADAMB, before using a simple concentration inequality to obtain a high probability result.

**Theorem J.1.** *Let* $d = d_A + d_S$, *then the expected regret of* ADAMB *for any sequence of starting states* $\{X_1^k\}_{k=1}^K$ *is upper bounded by*

$$\mathbb{E}[R(K)] \lesssim \begin{cases} LH^{1+\frac{1}{d+1}} K^{\frac{d+d_\mathcal{S}-1}{d+d_\mathcal{S}}} & d_S > 2 \\ LH^{1+\frac{1}{d+1}} K^{1-\frac{1}{d+d_\mathcal{S}+2}} & d_S \leq 2 \end{cases}$$

*where* $L = 1 + L_r + L_V + L_V L_T$ *and* $\lesssim$ *omits poly-logarithmic factors of* $\frac{1}{\delta}, H, K, d$, *and any universal constants.*

*Proof.* Using Lemma E.2 we have that

$$\mathbb{E}[R(K)] \leq \sum_{k=1}^{K}\sum_{h=1}^{H} \mathbb{E}\Big[\widetilde{\mathbf{V}}_h^{k-1}(\mathcal{S}(\mathcal{P}_h^{k-1}, X_h^k)) - \widetilde{\mathbf{V}}_h^k(\mathcal{S}(\mathcal{P}_h^k, X_h^k))\Big]$$

$$+ \sum_{h=1}^{H}\sum_{k=1}^{K} \mathbb{E}\big[2\text{RUCB}_h^k(B_h^k)\big] + \sum_{h=1}^{H}\sum_{k=1}^{K} \mathbb{E}\big[2\text{TUCB}_h^k(B_h^k)\big] + \sum_{k=1}^{K}\sum_{h=1}^{H} L_V \mathbb{E}\big[\mathcal{D}(B_h^k)\big].$$

We ignore the expectations, arguing a worst-case problem-independent bound on each of the quantities which appear in the summation. At the moment, we leave the splitting rule defined in the algorithm description as $n_+(\ell) = \phi 2^{\gamma\ell}$, where we specialize the regret bounds for the two cases at the end. We also ignore all poly-logarithmic factors of $H$, $K$, $d$, and absolute constants in the $\lesssim$ notation.

First note that via the splitting rule the algorithm maintains that for any selected ball $B$ we have that $\mathcal{D}(B) \leq (\phi/n_h^k(B))^{1/\gamma}$.

**Term One**: Using Lemma E.3 we have that

$$\sum_{k=1}^{K}\sum_{h=1}^{H} \widetilde{\mathbf{V}}_h^{k-1}(\mathcal{S}(\mathcal{P}_h^{k-1}, X_h^k)) - \widetilde{\mathbf{V}}_h^k(\mathcal{S}(\mathcal{P}_h^k, X_h^k)) \leq H^2 \max_h |\mathcal{S}(\mathcal{P}_h^k)|.$$

However, using Corollary F.2 we have that $|\mathcal{S}(\mathcal{P}_h^k)| \leq |\mathcal{P}_h^k| \leq 4^d\left(\frac{K}{\phi}\right)^{d/(d+\gamma)}$. Thus we can upper bound this term by $H^2 4^d\left(\frac{K}{\phi}\right)^{d/(d+\gamma)} \lesssim H^2 K^{d/(d+\gamma)} \phi^{-d/(d+\gamma)}$.

**Term Two and Four**:

$$\sum_{h=1}^{H}\sum_{k=1}^{K} \text{RUCB}_h^k(B_h^k) + L_V \mathcal{D}(B_h^k) = \sum_{h=1}^{H}\sum_{k=1}^{K} \sqrt{\frac{8\log(2HK^2/\delta)}{\sum_{B' \supseteq B_h^k} n_h^k(B')}} + 4L_r \mathcal{D}(B_h^k) + L_V \mathcal{D}(B_h^k)$$

$$\lesssim \sum_{h=1}^{H}\sum_{k=1}^{K} \sqrt{\frac{1}{n_h^k(B_h^k)}} + (L_r + L_V)\left(\frac{\phi}{n_h^k(B_h^k)}\right)^{\frac{1}{\gamma}}$$

where we used the definition of $\text{RUCB}_h^k(B)$ and the splitting rule.

Next we start by considering the case when $d_{\mathcal{S}} > 2$.

**Term Three**:

$$\sum_{h=1}^{H}\sum_{k=1}^{K} \text{TUCB}_h^k(B_h^k) = \sum_{h=1}^{H}\sum_{k=1}^{K} (L_T + 1)L_V 4\mathcal{D}(B) + 4L_V \sqrt{\frac{\log(HK^2/\delta)}{\sum_{B' \subseteq B} n_h^k(B')}}$$

$$+ \sum_{h=1}^{H}\sum_{k=1}^{K} L_T L_V \mathcal{D}(B) + cL_V\left(\sum_{B' \subseteq B} n_h^k(B')\right)^{-1/d_{\mathcal{S}}}$$

$$\lesssim \sum_{h=1}^{H}\sum_{k=1}^{K} (L_V L_T + L_V)\left(\frac{\phi}{n_h^k(B_h^k)}\right)^{\frac{1}{\gamma}} + L_V \sqrt{\frac{1}{n_h^k(B_h^k)}} + L_V\left(n_h^k(B_h^k)\right)^{-1/d_{\mathcal{S}}}.$$

where we used the definition of $\text{TUCB}_h^k(B_h^k)$.

**Combining Terms**: We will take $\phi \geq 1$ in order to tune the regret bound in terms of $H$ and $\gamma = d_{\mathcal{S}}$ in this situation. Using this we find that the dominating term is of the form $(\phi/n_h^k(B_h^k))^{1/\gamma}$. Thus we get that for $L = 1 + L_r + L_V + L_V L_T$,

$$R(K) \lesssim H^2 \phi^{-\frac{d}{d+\gamma}} K^{\frac{d}{d+\gamma}} + L\phi^{\frac{1}{\gamma}} \sum_{h=1}^{H}\sum_{k=1}^{K}\left(\frac{1}{n_h^k(B_h^k)}\right)^{\frac{1}{\gamma}}.$$

We now use Corollary F.3 for the case when $\alpha = \frac{1}{\gamma}$ and $\beta = 0$. This satisfies the required conditions of the result and we get:

$$R(K) \lesssim H^2 \phi^{-\frac{d}{d+\gamma}} K^{\frac{d}{d+\gamma}} + HL\phi^{\frac{1}{\gamma}} \phi^{-\frac{d}{\gamma(d+\gamma)}} K^{\frac{d+\gamma-1}{d+\gamma}}$$

$$= H^2\phi^{-\frac{d}{d+\gamma}}K^{\frac{d}{d+\gamma}} + HL\phi^{\frac{1}{d+\gamma}}K^{\frac{d+\gamma-1}{d+\gamma}}.$$

Taking $\phi$ as $\phi = H^{\frac{d+\gamma}{d+1}} \geq 1$ and plugging in $\gamma = d_{\mathcal{S}}$ we see that

$$R(K) \lesssim LH^{1+\frac{1}{d+1}}K^{\frac{d+d_{\mathcal{S}}-1}{d+d_{\mathcal{S}}}}.$$

Next we consider the case when $d_S \leq 2$. The first two terms and the fourth term remain the same, whereby now in the third term we have:

$$\sum_{h=1}^{H}\sum_{k=1}^{K}\mathrm{TUCB}_h^k(B_h^k) = \sum_{h=1}^{H}\sum_{k=1}^{K}L_V\left((5L_T+6)\mathcal{D}(B_h^k) + 4\sqrt{\frac{\log(HK^2)}{\sum_{B'\supseteq B_h^k}n_h^k(B')}} + c\sqrt{\frac{2^{d_S}\ell(B_h^k)}{\sum_{B'\supseteq B_h^k}n_h^k(B')}}\right)$$

$$\lesssim \sum_{h=1}^{H}\sum_{k=1}^{K}L_V(1+L_T)\left(\frac{\phi}{n_h^k(B_h^k)}\right)^{1/\gamma} + L_V\sqrt{\frac{1}{n_h^k(B_h^k)}} + L_V\sqrt{\frac{2^{d_S}\ell(B_h^k)}{n_h^k(B_h^k)}}.$$

**Combining Terms**: Again using that we take $\phi \geq 1$ we can combine terms to get:

$$R(K) \lesssim H^2\phi^{-\frac{d}{d+\gamma}}K^{\frac{d}{d+\gamma}} + L\sum_{h=1}^{H}\sum_{k=1}^{K}\left(\frac{1}{n_h^k(B_h^k)}\right)^{\frac{1}{\gamma}} + L\sum_{h=1}^{H}\sum_{k=1}^{K}\sqrt{\frac{2^{d_S}\ell(B_h^k)}{n_h^k(B_h^k)}}.$$

Again using Corollary F.3 for the case when $\gamma = d_{\mathcal{S}} + 2$ which satisfies the requirements we get

$$R(K) \lesssim H^2\phi^{-\frac{d}{d+\gamma}}K^{\frac{d}{d+\gamma}} + LH\phi^{\frac{1}{d+\gamma}}K^{\frac{d+\gamma-1}{d+\gamma}} + LH\phi^{-\frac{d}{2(d+\gamma)}}K^{\frac{d+\frac{1}{2}\gamma+\frac{1}{2}}{d+\gamma}}$$

$$\lesssim H^2\phi^{-\frac{d}{d+\gamma}}K^{\frac{d}{d+\gamma}} + LH\phi^{\frac{1}{d+\gamma}}K^{\frac{d+\gamma-1}{d+\gamma}}.$$

where we used the fact that the second term dominates the third when $\gamma = d_{\mathcal{S}} + 2$. Taking $\phi$ the same as the previous case we get:

$$R(K) \lesssim LH^{1+\frac{1}{d+1}}K^{\frac{d+d_{\mathcal{S}}+1}{d+d_{\mathcal{S}}+2}}.$$

$\square$

Using this bound on the expected regret and a straightforward use of Azuma-Hoeffding's inequality we can show the following:

**Theorem J.2.** *Let* $d = d_A + d_S$, *then the regret of* ADAMB *for any sequence of starting states* $\{X_1^k\}_{k=1}^K$ *is upper bounded with probability at least* $1 - \delta$ *by*

$$R(K) \lesssim \begin{cases} LH^{1+\frac{1}{d+1}}K^{\frac{d+d_{\mathcal{S}}-1}{d+d_{\mathcal{S}}}} & d_S > 2 \\ LH^{1+\frac{1}{d+1}}K^{\frac{d+d_{\mathcal{S}}+1}{d+d_{\mathcal{S}}+2}} & d_S \leq 2 \end{cases}$$

*where* $L = 1 + L_r + L_V + L_V L_T$ *and* $\lesssim$ *omits poly-logarithmic factors of* $\frac{1}{\delta}$, $H, K$, $d$, *and any universal constants.*

*Proof.* Let $R(K) = \sum_{k=1}^{K}V_1^\star(X_1^k) - V_1^{\pi^k}(X_1^k)$ be the true regret of the algorithm. We apply Azuma-Hoeffding's inequality, where we use Theorem J.1 to find a bound on its expectation. Keeping the same notation as before, let $Z_\tau = \sum_{k=1}^{\tau}V_1^\star(X_1^k) - V_1^{\pi^k} - \mathbb{E}\left[\sum_{k=1}^{\tau}V_1^\star(X_1^k) - V_1^{\pi^k}(X_1^k)\right]$. Clearly we have that $Z_\tau$ is adapted to the filtration $\mathcal{F}_\tau$, and has finite absolute moments. Moreover, using the fact that the value function is bounded above by $H$ then

$$|Z_\tau - Z_{\tau-1}| = |V_1^\star(X_1^\tau) - V_1^{\pi^\tau}(X_1^\tau) - \mathbb{E}\left[V_1^\star(X_1^\tau) - V_1^{\pi^\tau}(X_1^\tau)\right]|$$

$$\leq 4H.$$

Thus we get, via a straightforward application of Azuma-Hoeffding's that with probability at least $1 - \delta$,

$$R(K) \leq \mathbb{E}[R(K)] + \sqrt{32H^2K\log(1/\delta)}$$

$$\lesssim \begin{cases} LH^{1+\frac{1}{d+1}}K^{\frac{d+d_{\mathcal{S}}-1}{d+d_{\mathcal{S}}}} & d_S > 2 \\ LH^{1+\frac{1}{d+1}}K^{1-\frac{1}{d+d_{\mathcal{S}}+2}} & d_S \leq 2. \end{cases}$$

$\square$

# K Proofs for Technical Results

Finally we provide some additional proofs of the technical results we use in our regret analysis.

*Proof of Lemma D.1.* Recall we want to show that for any ball $B$ and $h, k \in [H] \times [K]$ we have

$$\frac{\sum_{B' \supseteq B} \mathcal{D}(B') n_h^k(B')}{\sum_{B' \supseteq B} n_h^k(B')} \leq 4\mathcal{D}(B).$$

First notice that the term on the left hand side can be rewritten as:

$$\frac{\sum_{B' \supseteq B} \mathcal{D}(B') n_h^k(B')}{\sum_{B' \supseteq B} n_h^k(B')} = \frac{1}{t} \sum_{i=1}^{t} \mathcal{D}(B_h^{k_i})$$

where $t = \sum_{B' \supseteq B} n_h^k(B')$ is the number of times $B$ or its ancestors were selected and $k_1, \ldots, k_t$ are the episodes for which they were selected. Using the fact that $\mathcal{D}(B_h^{k_i})$ are decreasing over time as the partition is refined, this average can be upper bounded by only averaging over the ancestors of $B$, i.e.

$$\frac{\sum_{B' \supseteq B} \mathcal{D}(B') n_h^k(B')}{\sum_{B' \supseteq B} n_h^k(B')} \leq \frac{\sum_{B' \supsetneq B} \mathcal{D}(B') n_h^k(B')}{\sum_{B' \supsetneq B} n_h^k(B')}.$$

Using the splitting threshold $n_+(B) = \phi 2^{\gamma \ell(B)}$, we can upper bound this quantity by

$$
\begin{aligned}
\frac{\sum_{B' \supsetneq B} n_h^k(B') \mathcal{D}(B')}{\sum_{B' \supsetneq B} n_h^k(B')} &= \frac{\sum_{i=0}^{\ell(B)-1} 2^{-i} \phi 2^{\gamma i}}{\sum_{i=0}^{\ell(B)-1} \phi 2^{\gamma i}} \\
&\leq \frac{2^{(\gamma-1)(\ell(B)-1)} \sum_{i=0}^{\infty} 2^{-(\gamma-1)i}}{2^{\gamma(\ell(B)-1)}} \\
&\leq \frac{2 \cdot 2^{(\gamma-1)(\ell(B)-1)}}{2^{\gamma(\ell(B)-1)}} \qquad \text{because } 2^{-(\gamma-1)} \leq \frac{1}{2} \\
&= 4 \cdot 2^{-\ell(B)} = 4\mathcal{D}(B).
\end{aligned}
$$

$\square$

*Proof of Lemma D.3, for $d_\mathcal{S} > 2$.* Let $h, k \in [H] \times [K]$ and $B \in \mathcal{P}_h^k$ be fixed and $(x, a) \in B$ be arbitrary. We use a combination of Proposition 10 and 20 from [46]. Let $P_0 = T_h(\cdot \mid x_0, a_0)$ where $(x_0, a_0) = (\tilde{x}(B), \tilde{a}(B))$ is the center of the ball $B$. Our goal then is to come up with concentration between the one-Wasserstein metric of $\overline{\mathbf{T}}_h^k(\cdot \mid B)$ and $T_h(\cdot \mid x, a)$. We break the proof down into four stages, where we show concentration between the one-Wasserstein distance of various measures. As defined, $\overline{\mathbf{T}}_h^k(\cdot \mid B)$ is a distribution over $\square_{\ell(B)}$, the uniform discretization of $\mathcal{S}$ at over balls with diameter $2^{-\ell(B)}$. However, we will view $\overline{\mathbf{T}}_h^k(\cdot \mid B)$ as a distribution over a set of finite points in $\mathcal{S}$, where

$$\overline{\mathbf{T}}_h^k(x \mid B) = \overline{\mathbf{T}}_h^k(A \mid B) \quad \text{if } x = \tilde{x}(A).$$

**Step One**: Let $\tilde{T}_h^k(\cdot \mid B)$ be the true empirical distribution of all samples collected from $B'$ for any $B'$ which is an ancestor of $B$, i.e.

$$\tilde{T}_h^k(\cdot \mid B) = \frac{\sum_{B' \supseteq B} \sum_{k' \leq k} \delta_{X_{h+1}^{k'}} \mathbb{1}_{[B_h^{k'} = B']}}{\sum_{B' \supseteq B} n_h^k(B')}. \tag{7}$$

Let $A_{h+1}^{k'}$ denote the region in $\square_{\ell(B_h^{k'})}$ containing the point $X_{h+1}^{k'}$. Recall $\overline{\mathbf{T}}_h^k(\cdot \mid B)$ is the distribution defined according to:

$$\overline{\mathbf{T}}_h^k(\cdot \mid B) = \frac{\sum_{B' \supseteq B} \sum_{k' \leq k} \mathbb{1}_{[B_h^{k'} = B']} \sum_{A \in \square_{\ell(B)}: A \subseteq A_{h+1}^{k'}} 2^{-d_\mathcal{S}(\ell(B') - \ell(B))} \delta_{\tilde{x}(A)}}{\sum_{B' \supseteq B} n_h^k(B')}.$$

We can verify that $\sum_{A\in\square_{\ell(B)}:A\subseteq A_{h+1}^{k'}} 2^{-d_{\mathcal{S}}(\ell(B')-\ell(B))} = 1$ as the number of regions in $\square_{\ell(B)}$ which contain any region in $\square_{\ell(B')}$ is exactly $2^{d_{\mathcal{S}}(\ell(B')-\ell(B))}$. Furthermore $X_{h+1}^{k'}$ and $\tilde{x}(A)$ are both contained in $A_{h+1}^{k'}$ so that $\mathcal{D}_{\mathcal{S}}(X_{h+1}^{k'}, \tilde{x}(A)) \leq \mathcal{D}_{S}(A_{h+1}^{k'}) \leq \mathcal{D}(B_h^{k'})$, where we use the definition of $\square_{\ell(B_h^{k'})}$ for the last inequality. Using these observations, it follows that

$$d_W(\overline{\mathbf{T}}_h^k(\cdot \mid B), \tilde{T}_h^k(\cdot \mid B)) \leq \frac{\sum_{B'\supseteq B}\sum_{k'\leq k}\mathbb{1}_{[B_h^{k'}=B']}}{\sum_{B'\supseteq B} n_h^k(B')} \sum_{A\in\square_{\ell(B)}:A\subseteq A_{h+1}^{k'}} 2^{-d_{\mathcal{S}}(\ell(B')-\ell(B))}\mathcal{D}_{\mathcal{S}}(X_{h+1}^{k'}, \tilde{x}(A))$$

$$\leq \frac{\sum_{B'\supseteq B}\sum_{k'\leq k}\mathbb{1}_{[B_h^{k'}=B']}}{\sum_{B'\supseteq B} n_h^k(B')} \sum_{A\in\square_{\ell(B)}:A\subseteq A_{h+1}^{k'}} 2^{-d_{\mathcal{S}}(\ell(B')-\ell(B))}\mathcal{D}_{\mathcal{S}}(A_{h+1}^{k'})$$

$$\leq \frac{\sum_{B'\supseteq B}\sum_{k'\leq k}\mathbb{1}_{[B_h^{k'}=B']}\mathcal{D}(B_h^{k'})}{\sum_{B'\supseteq B} n_h^k(B')}$$

$$\leq \frac{\sum_{B'\supseteq B}\mathcal{D}(B')n_h^k(B')}{\sum_{B'\supseteq B} n_h^k(B')}$$

**Step Two**: Next we bound the difference between $\tilde{T}_h^k(\cdot \mid B)$ and $\tilde{T}_h(\cdot \mid x_0, a_0)$ where $\tilde{T}_h(\cdot \mid x_0, a_0)$ is a 'ghost empirical distribution' of samples whose marginal distribution is $T_h(\cdot \mid x_0, a_0)$. By Lipschitzness of the transition kernels, for every $x, a, x_0, a_0$,

$$d_W(T_h(\cdot \mid x, a), T_h(\cdot \mid x_0, a_0)) \leq L_T \mathcal{D}((x,a), (x_0, a_0)).$$

Using the coupling definition of the Wasserstein metric, there exists a family of distributions $\xi(\cdot, \cdot | x, a, x_0, a_0)$ parameterized by $x, a, x_0, a_0$ such that

$$\mathbb{E}_{(Z,Y)\sim\xi(\cdot,\cdot|x,a,x_0,a_0)}[\mathcal{D}_{\mathcal{S}}(Z,Y)] = d_W(T_h(\cdot \mid x, a), T_h(\cdot \mid x_0, a_0)) \leq L_T\mathcal{D}((x,a),(x_0,a_0)),$$

whose marginals are

$$\int_{\mathcal{S}} \xi(z,y|x,a,x_0,a_0)dy = T_h(z \mid x, a) \text{ and } \int_{\mathcal{S}} \xi(z,y|x,a,x_0,a_0)dz = T_h(y \mid x_0, a_0).$$

For $(Z,Y) \sim \xi(\cdot, \cdot | x, a, x_0, a_0)$, let $\xi'(\cdot | z, x, a, x_0, a_0)$ denote the conditional distribution of $Y$ given $Z$, such that

$$\xi(z,y|x,a,x_0,a_0) = T_h(z \mid x, a)\xi'(y|z,x,a,x_0,a_0). \tag{8}$$

For ease of notation let us denote $t = \sum_{B'\supseteq B} n_h^k(B')$ and let the indexing $k_1, \ldots, k_t$ be the episodes for which $B$ or its ancestors were selected by the algorithm. For the sequence of samples $\{(X_h^{k_i}, A_h^{k_i}, X_{h+1}^{k_i})\}_{i\in[t]}$ realized by our algorithm, consider a 'ghost sample' $Y_1, \ldots, Y_t$ such that $Y_i \sim \xi'(\cdot | X_{h+1}^{k_i}, X_h^{k_i}, A_h^{k_i}, x_0, a_0)$ for $i \in [t]$. Let $\tilde{T}_h(\cdot \mid x_0, a_0)$ denote the empirical distribution of these samples such that

$$\tilde{T}_h(\cdot \mid x_0, a_0) = \frac{1}{t}\sum_{i=1}^{t} \delta_{Y_i} \text{ and recall by definition } \tilde{T}_h^k(\cdot \mid B) = \frac{1}{t}\sum_{i=1}^{t} \delta_{X_{h+1}^{k_i}}.$$

Using the definition of the Wasserstein distance we have that

$$d_W(\tilde{T}_h^k(\cdot \mid B), \tilde{T}_h(\cdot \mid x_0, a_0)) \leq \frac{1}{t}\sum_{i=1}^{t} \mathcal{D}_{\mathcal{S}}(X_{h+1}^{k_i}, Y_i). \tag{9}$$

We will use Azuma-Hoeffding's to provide a high probability bound on this term by its expectation. For any $\tau \leq K$ define the quantity

$$Z_\tau = \sum_{i=1}^{\tau} \mathcal{D}_{\mathcal{S}}(X_{h+1}^{k_i}, Y_i) - \mathbb{E}\left[\mathcal{D}_{\mathcal{S}}(X_{h+1}^{k_i}, Y_i)\right].$$

Let $\mathcal{F}_i$ be the filtration containing $\mathcal{F}_{k_i+1} \cup \{Y_j\}_{j \leq i}$. It follows that $Z_\tau$ is a martingale with respect to $\mathcal{F}_\tau$. The process is adapted to the filtration by construction, has finite first moment, and we have that

$$\mathbb{E}[Z_\tau \mid \mathcal{F}_{\tau-1}] = Z_{\tau-1} + \mathbb{E}\big[\mathcal{D}_\mathcal{S}(X_{h+1}^\tau, Y_\tau)\big] - \mathbb{E}\big[\mathcal{D}_\mathcal{S}(X_{h+1}^\tau, Y_\tau)\big] = Z_{\tau-1}.$$

Moreover, we also have the differences are bounded by

$$|Z_\tau - Z_{\tau-1}| = \left|\mathcal{D}_\mathcal{S}(X_{h+1}^{k_\tau}, Y_\tau) - \mathbb{E}\big[\mathcal{D}_\mathcal{S}(X_{h+1}^{k_\tau}, Y_\tau)\big]\right| \leq 2$$

since by assumption $\mathcal{D}_\mathcal{S}(\mathcal{S}) \leq 1$. By Azuma-Hoeffding's inequality, with probability at least $1 - \frac{\delta}{HK^2}$,

$$\frac{1}{\tau}\sum_{i=1}^\tau \mathcal{D}_\mathcal{S}(Y_i, X_{h+1}^{k_i}) \leq \mathbb{E}\left[\frac{1}{\tau}\sum_{i=1}^\tau \mathcal{D}_\mathcal{S}(Y_i, X_{h+1}^{k_i})\right] + \sqrt{\frac{8\log(HK^2/\delta)}{\tau}}. \tag{10}$$

Moreover, by construction of the ghost samples we have that

$$\frac{1}{\tau}\sum_{i=1}^\tau \mathbb{E}\Big[\mathcal{D}_\mathcal{S}(Y_i, X_{h+1}^{k_i})\Big] = \frac{1}{\tau}\sum_{i=1}^\tau \mathbb{E}\Big[d_W(T_h(\cdot \mid X_h^{k_i}, A_h^{k_i}), T_h(\cdot \mid x_0, a_0))\Big]$$

$$\leq \frac{1}{\tau}\sum_{i=1}^\tau L_V \mathcal{D}(B_{h+1}^{k_i})$$

since $x_0, a_0$ is in the ball $B$ which is contained in the ball $B_{h+1}^{k_i}$. By plugging this into Eq. (10), taking a union bound over the number of steps $H$, the number of episodes $K$, the number of potential stopping times $K$, and combining it with Eq. (9) and using the construction of $t$, it follows that with probability at least $1 - \delta$, for all $h, k, B$

$$d_W(\tilde{T}_h^k(\cdot \mid B), \tilde{T}_h(\cdot \mid x_0, a_0)) \leq L_T \frac{\sum_{B' \supseteq B} n_h^k(B')\mathcal{D}(B')}{\sum_{B' \supseteq B} n_h^k(B')} + \sqrt{\frac{8\log(HK^2/\delta)}{\sum_{B' \supseteq B} n_h^k(B')}}.$$

Note that we do not need to union bound over all balls $B \in \mathcal{P}_h^k$ as the estimate of only one ball is changed per (step, episode) pair, i.e. $\hat{\mathbf{T}}_h^k(B)$ and correspondingly $\overline{\mathbf{T}}_h^k(B)$ is changed for only a single ball $B = B_h^k$ per episode. For all balls not selected, it inherits the concentration of the good event from the previous episode because its estimate does not change. Furthermore, even if ball $B$ is "split" in episode $k$, all of its children inherit the value of the parent ball, and thus also inherits the good event, so we still only need to consider the update for $B_h^k$ itself.

**Step Three**: Next we bound $d_W(\tilde{T}_h(\cdot \mid x_0, a_0), T_h(\cdot \mid x_0, a_0))$. Recall $\mathcal{F}_i$ is the filtration containing $\mathcal{F}_{k_i+1} \cup \{Y_j\}_{j \leq i}$. Note that the joint distribution over $\{(X_h^{k_i}, A_h^{k_i}, X_{h+1}^{k_i}, Y_i)\}_{i \in [t]}$ is given by

$$G_t(\{(X_h^{k_i}, A_h^{k_i}, X_{h+1}^{k_i}, Y_i)\}_{i \in [t]}) = \prod_{i=1}^t (P(X_h^{k_i}, A_h^{k_i} \mid \mathcal{F}_{i-1})T_h(X_{h+1}^{k_i}|X_h^{k_i}, A_h^{k_i})\xi'(Y_i|X_{h+1}^{k_i}, X_h^{k_i}, A_h^{k_i}, x_0, a_0),$$

where $P(X_h^{k_i}, A_h^{k_i} \mid \mathcal{F}_{i-1})$ is given by the dynamics of the MDP along with the policy that the algorithm plays. Then we have

$$\int_{\mathcal{S} \times \mathcal{A} \times \mathcal{S}} G_t(\{(X_h^{k_i}, X_h^{k_i}, X_{h+1}^{k_i}, Y_i)\}_{i \in [t]})dX_h^{k_t}dA_h^{k_t}dX_{h+1}^{k_t}$$

$$= G_{t-1}(\{(X_h^{k_i}, X_h^{k_i}, X_{h+1}^{k_i}, Y_i)\}_{i \in [t-1]})$$

$$\cdot \int_{\mathcal{S} \times \mathcal{A}} P(X_h^{k_t}, A_h^{k_t} \mid \mathcal{F}_{k_{t-1}})\left(\int_\mathcal{S} \xi(X_{h+1}^{k_i}, Y_i|X_h^{k_i}, A_h^{k_i}, x_0, a_0)dX_{h+1}^{k_t}\right)dX_h^{k_t}dA_h^{k_t}$$

$$= G_{t-1}(\{(X_h^{k_i}, X_h^{k_i}, X_{h+1}^{k_i}, Y_i)\}_{i \in [t-1]})T_h(Y_i|x_0, a_0)\int_{\mathcal{S} \times \mathcal{A}} P(X_h^{k_t}, A_h^{k_t} \mid \mathcal{F}_{k_{t-1}})dX_h^{k_t}dA_h^{k_t}$$

$$= G_{t-1}(\{(X_h^{k_i}, X_h^{k_i}, X_{h+1}^{k_i}, Y_i)\}_{i \in [t-1]})T_h(Y_i|x_0, a_0).$$

By repeating this calculation, we can verify that the marginal distribution of $Y_1 \ldots Y_t$ is $\prod_{i \in [t]} T_h(Y_i | x_0, a_0)$. Following Proposition 10 and 20 from [46] for the case when $d_S > 2$ we have that with probability at least $1 - \delta/HK^2$ for some universal constant $c$,

$$d_W(\tilde{T}_h(\cdot \mid x_0, a_0), T_h(\cdot \mid x_0, a_0)) \leq \mathbb{E}\Big[d_W(\tilde{T}_h(\cdot \mid x_0, a_0), T_h(\cdot \mid x_0, a_0)\Big] + \sqrt{\frac{\log(HK^2/\delta)}{\sum_{B' \subseteq B} n_h^k(B')}}$$

$$\leq c\left(\sum_{B' \subseteq B} n_h^k(B')\right)^{-1/d_S} + \sqrt{\frac{\log(HK^2/\delta)}{\sum_{B' \subseteq B} n_h^k(B')}}.$$

**Step Four**: Using the assumption that $T_h$ is Lipschitz and $(x_0, a_0)$ and $(x, a) \in B$ we have that

$$d_W(T_h(\cdot \mid x, a), T_h(\cdot \mid x_0, a_0)) \leq L_T \mathcal{D}((x, a), (x_0, a_0)) \leq L_T \mathcal{D}(B).$$

Putting all of the pieces together we get that

$$d_W(\overline{\mathbf{T}}_h^k(\cdot \mid B), T_h(\cdot \mid x, a))$$

$$\leq d_W(\overline{\mathbf{T}}_h^k(\cdot \mid B), \tilde{T}_h^k(\cdot \mid B)) + d_W(\tilde{T}_h^k(\cdot \mid B), \tilde{T}_h(\cdot \mid x_0, a_0))$$

$$\quad + d_W(\tilde{T}_h(\cdot \mid x_0, a_0), T_h(\cdot \mid x_0, a_0)) + d_W(T_h(\cdot \mid x_0, a_0), T_h(\cdot \mid x, a))$$

$$\leq \frac{\sum_{B' \supseteq B} n_h^k(B') \mathcal{D}(B')}{\sum_{B' \supseteq B} n_h^k(B')} + \frac{\sum_{B' \subseteq B} L_T n_h^k(B') \mathcal{D}(B')}{\sum_{B' \subseteq B} n_h^k(B')} + \sqrt{\frac{8 \log(HK^2/\delta)}{\sum_{B' \subseteq B} n_h^k(B')}}$$

$$\quad + L_T \mathcal{D}(B) + c\left(\sum_{B' \subseteq B} n_h^k(B')\right)^{-1/d_S} + \sqrt{\frac{\log(HK^2/\delta)}{\sum_{B' \subseteq B} n_h^k(B')}}$$

$$= (L_T + 1)\frac{\sum_{B' \supseteq B} n_h^k(B') \mathcal{D}(B')}{\sum_{B' \supseteq B} n_h^k(B')} + 4\sqrt{\frac{\log(HK^2/\delta)}{\sum_{B' \subseteq B} n_h^k(B')}} + L_T \mathcal{D}(B) + c\left(\sum_{B' \subseteq B} n_h^k(B')\right)^{-1/d_S}$$

$$\leq (5L_T + 4)\mathcal{D}(B) + 4\sqrt{\frac{\log(HK^2/\delta)}{\sum_{B' \subseteq B} n_h^k(B')}} + c\left(\sum_{B' \subseteq B} n_h^k(B')\right)^{-1/d_S} \quad \text{by Lemma D.1}$$

The result then follows via a union bound over $H, K$, the $K$ possible values of the random variable $n_h^k(B)$. Per usual we do not need to union bound over the number of balls as the estimate of only one ball is updated per iteration. $\qquad\square$

The second concentration inequality deals with the case when $d_S \leq 2$. The constant $c$ in Proposition 10 from [46] becomes very large when $d_S \to 2$, and thus we instead use the fact that $\overline{\mathbf{T}}_h^k(\cdot \mid B)$ has finite support over $2^{d_S \ell(B)}$ points and consider Wasserstein convergence of empirical distributions sampled from discrete distributions. $T_h(\cdot \mid x, a)$ is still a (potentially) continuous distribution so we need to change Step 3 of the above argument slightly.

*Proof of Lemma D.3, for $d_S \leq 2$.* Let $h, k \in [H] \times [K]$ and $B \in \mathcal{P}_h^k$ be fixed with $(x, a) \in B$ arbitrary. We use a combination of Proposition 10 and 20 from [46] for the case when when the distributions have finite support. As before, let $(x_0, a_0) = (\tilde{x}(B), \tilde{a}(B))$ be the center of the ball $B$. We again break the proof down into several stages, where we show concentration between the Wasserstein distance of various measures. In order to obtain bounds that scale with the support of $\overline{\mathbf{T}}_h^k(\cdot \mid B)$ we consider "snapped" versions of the distributions, where we snap the resulting random variable to its point in the discretization of $\square_{\ell(B)}$. We repeat the same first two steps as Lemma D.3 which are restated again here for completeness.

**Step One**: Let $\tilde{T}_h^k(\cdot \mid B)$ be the true empirical distribution of all samples collected from $B'$ for any $B'$ which is an ancestor of $B$, formally defined in Eq. (7). By the same argument as Step 1 in the

proof of Lemma D.3 it follows that

$$d_W(\overline{\mathbf{T}}_h^k(\cdot \mid B), \tilde{T}_h^k(\cdot \mid B)) \leq \frac{\sum_{B' \supseteq B} n_h^k(B') \mathcal{D}(B')}{\sum_{B' \supseteq B} n_h^k(B')}$$

**Step Two**: Let $\tilde{T}_h(\cdot \mid x_0, a_0)$ be a 'ghost empirical distribution' of samples whose marginal distribution is $T_h(\cdot \mid x_0, a_0)$. It consists of $t = \sum_{B' \supseteq B} n_h^k(B')$ samples drawn from $Y_i \sim \xi'(\cdot | X_{h+1}^{k_i}, X_h^{k_i}, A_h^{k_i}, x_0, a_0)$ as constructed in Eq. (8). By the same argument from Step 2 of the proof of Lemma D.3, with probability at least $1 - \delta$, for all $h, k, B$

$$d_W(\tilde{T}_h^k(\cdot \mid B), \tilde{T}_h(\cdot \mid x_0, a_0)) \leq L_T \frac{\sum_{B' \supseteq B} n_h^k(B') \mathcal{D}(B')}{\sum_{B' \supseteq B} n_h^k(B')} + \sqrt{\frac{8 \log(HK^2/\delta)}{\sum_{B' \supseteq B} n_h^k(B')}}.$$

**Step Three**: Next we let $\tilde{T}_h^{\ell(B)}(\cdot \mid x_0, a_0)$ to be the snapped empirical distribution of the ghost samples $Y_1 \ldots Y_t$ to their nearest point in $\square_{\ell(B)}$. Denote $\tilde{Y}_i$ as $\tilde{x}(A_i)$ where $A_i \in \square_{\ell(B)}$ is the region containing the point $Y_i$. It follows that:

$$\tilde{T}_h^{\ell(B)}(\cdot \mid x_0, a_0) = \frac{1}{t} \sum_{i=1}^t \sum_{A \in \square_{\ell(B)}} \mathbb{1}_{[Y_i \in A]} \delta_{\tilde{x}(A)} = \frac{1}{t} \sum_{i=1}^t \delta_{\tilde{Y}_i}.$$

Since each of the points are moved by at most $\mathcal{D}_\mathcal{S}(A_i) \leq \mathcal{D}(B)$ by construction of $\tilde{T}_h^{\ell(B)}$ and $\square_{\ell(B)}$, we have that $d_W(\tilde{T}_h^{\ell(B)}(\cdot \mid x_0, a_0), \tilde{T}_h(\cdot \mid x_0, a_0)) \leq \mathcal{D}(B)$.

Define the snapped distribution $T_h^{\ell(B)}(\cdot \mid x_0, a_0)$ according to

$$T_h^{\ell(B)}(x \mid x_0, a_0) = \sum_{A \in \square_{\ell(B)}} \mathbb{1}_{[x = \tilde{x}(A)]} \int_A T_h(y \mid x_0, a_0) dy$$

where we note that this distribution has finite support of size $2^{-d_S \ell(B)}$ over the set $\{\tilde{x}(A)\}_{A \in \square_{\ell(B)}}$.

By the same argument from Step 3 of the proof of Lemma D.3, it holds that by construction, the marginal distribution of $Y_1 \ldots Y_t$ denoted $f_{Y_1 \ldots Y_t}$ is $\prod_{i \in [t]} T_h(Y_i | x_0, a_0)$. Furthermore, conditioned on $(Y_1 \ldots Y_t)$, the snapped samples $(\tilde{Y}_1 \ldots \tilde{Y}_t)$ are fully determined. Recall that $\tilde{Y}_i$ can only take values in $\{\tilde{x}(A)\}_{A \in \square_{\ell(B)}}$. If $A_i$ refers to the set in $\square_{\ell(B)}$ for which $\tilde{Y}_i = \tilde{x}(A_i)$, then

$$P(\tilde{Y}_1 \ldots \tilde{Y}_t) = P(Y_1 \in A_1, \ldots Y_t \in A_t)$$
$$= \int_{A_1} \int_{A_2} \cdots \int_{A_t} f_{Y_1 \ldots Y_t}(y_1 \ldots y_t) dy_t \cdots dy_1$$
$$= \int_{A_1} \int_{A_2} \cdots \int_{A_t} \prod_{i \in [t]} T_h(Y_i | x_0, a_0) dy_t \cdots dy_1$$
$$= \prod_{i \in [t]} \int_{A_i} T_h(Y_i | x_0, a_0) dy_i$$
$$= T_h^{\ell(B)}(\tilde{Y}_i | x_0, a_0).$$

such that the marginal distribution of $\tilde{Y}_1 \ldots \tilde{Y}_t$ is equivalent to that of a set of $t$ i.i.d. samples from $T_h^{\ell(B)}(\cdot | x_0, a_0)$.

By Proposition [13] and [20] from from [46], for some universal constant $c$, with probability at least $1 - \frac{\delta}{HK^2}$,

$$d_W(\tilde{T}_h^{\ell(B)}(\cdot \mid x_0, a_0), T_h^{\ell(B)}(\cdot \mid x_0, a_0))$$
$$\leq \mathbb{E}\left[d_W(\tilde{T}_h^{\ell(B)}(\cdot \mid x_0, a_0), \tilde{T}_h^{\ell(B)}(\cdot \mid x_0, a_0))\right] + \sqrt{\frac{\log(HK^2/\delta)}{t}}$$

$$\leq c\sqrt{\frac{2^{d_S \ell(B)}}{t}} + \sqrt{\frac{\log(HK^2/\delta)}{t}}.$$

**Step Four**: Next we construct a coupling to show that $d_W(T_h^{\ell(B)}(\cdot \mid x_0, a_0), T_h(\cdot \mid x_0, a_0)) \leq \mathcal{D}(B)$. For a coupling we define a family of distributions $\Gamma(\cdot, \cdot | x_0, a_0, \ell)$ parameterized by $x_0, a_0, \ell$ such that

$$\Gamma(x_{snap}, x_{orig} | x_0, a_0, \ell) = T_h(x_{orig} \mid x_0, a_0) \sum_{A \in \mathcal{S}_\ell} \mathbb{1}_{[x_{snap} = \tilde{x}(A)]} \mathbb{1}_{[x_{orig} \in A]}.$$

First notice that the marginals of these distributions match $T_h^\ell$ and $T_h$ respectively since:

$$\int_{\mathcal{S}} \Gamma(x_{snap}, x \mid x_0, a_0, \ell) dx = \sum_{A \in \mathcal{S}_\ell} \mathbb{1}_{[x_{snap} = \tilde{x}(A)]} \int_A T_h(x \mid x_0, a_0) dx = T_h^\ell(x_{snap} \mid x_0, a_0)$$

and

$$\int_{\mathcal{S}} \Gamma(x, x_{orig} \mid x_0, a_0, \ell) dx = \sum_{A \in \mathcal{S}_\ell} \Gamma(\tilde{x}(A), x_{orig} \mid x_0, a_0, \ell) = T_h(x_{orig} \mid x_0, a_0).$$

Using this coupling $\Gamma$ it follows by definition of Wasserstein distance that

$$d_W(T_h^\ell(\cdot \mid x_0, a_0), T_h(\cdot \mid x_0, a_0)) \leq \mathbb{E}_{X_{snap}, X_{orig} \sim \Gamma(\cdot | x_0, a_0, \ell(B))}[\mathcal{D}_{\mathcal{S}}(X_{snap}, X_{orig})]$$
$$\leq \mathcal{D}(B)$$

where we used that $X_{snap}$ and $X_{orig}$ have distance bounded by $\mathcal{D}_{\mathcal{S}}(A)$ for some $A \in \square_{\ell(B)}$, and by construction of $\square_{\ell(B)}$, $\mathcal{D}_{\mathcal{S}}(A) = \mathcal{D}(B)$.

**Step Five**: Using the assumption that $T_h$ is Lipschitz and $(x_0, a_0)$ and $(x, a) \in B$ we have that

$$d_W(T_h(\cdot \mid x, a), T_h(\cdot \mid x_0, a_0)) \leq L_T \mathcal{D}((x, a), (x_0, a_0)) \leq L_T \mathcal{D}(B).$$

Putting all of the pieces together and a union bound over $H$, $K$, the possible values of the random variables $t$, and the number of balls $B \in \mathcal{P}_h^K$ we get that:

$$d_W(\overline{\mathbf{T}}_h^k(\cdot \mid B), T_h(\cdot \mid x_0, a_0))$$
$$\leq d_W(\overline{\mathbf{T}}_h^k(\cdot \mid B), \tilde{T}_h^k(\cdot \mid B)) + d_W(\tilde{T}_h^k(\cdot \mid B), \tilde{T}_h(\cdot \mid x_0, a_0)) + d_W(\tilde{T}_h(\cdot \mid x_0, a_0), \tilde{T}_h^{\ell(B)}(\cdot \mid x_0, a_0))$$
$$\quad + d_W(\tilde{T}_h^{\ell(B)}(\cdot \mid x_0, a_0), T_h(\cdot \mid x_0, a_0)) + d_W(T_h(\cdot \mid x_0, a_0), T_h(\cdot \mid x, a))$$
$$\leq \frac{\sum_{B' \supseteq B} n_h^k(B') \mathcal{D}(B')}{\sum_{B' \supseteq B} n_h^k(B')} + \frac{\sum_{B' \supseteq B} L_T n_h^k(B') \mathcal{D}(B')}{\sum_{B' \supseteq B} n_h^k(B')} + \sqrt{\frac{8 \log(HK^2/\delta)}{\sum_{B' \supseteq B} n_h^k(B')}}$$
$$\quad + c\sqrt{\frac{2^{d_S \ell(B)}}{\sum_{B' \supseteq B} n_h^k(B')}} + \sqrt{\frac{\log(HK^2)}{\sum_{B' \supseteq B} n_h^k(B')}} + 2\mathcal{D}(B) + L_T \mathcal{D}(B)$$
$$= (1 + L_T) \frac{\sum_{B' \supseteq B} n_h^k(B') \mathcal{D}(B')}{\sum_{B' \supseteq B} n_h^k(B')} + 4\sqrt{\frac{\log(HK^2/\delta)}{\sum_{B' \supseteq B} n_h^k(B')}} + (2 + L_T) \mathcal{D}(B) + c\sqrt{\frac{2^{d_S \ell(B)}}{\sum_{B' \supseteq B} n_h^k(B')}}$$
$$\leq (5 L_T + 4) \mathcal{D}(B) + 4\sqrt{\frac{\log(HK^2/\delta)}{\sum_{B' \supseteq B} n_h^k(B')}} + c\sqrt{\frac{2^{d_S \ell(B)}}{\sum_{B' \supseteq B} n_h^k(B')}} \quad \text{by Lemma D.1.}$$

which is $\frac{1}{L_V} \text{TUCB}_h^k(B)$ as needed. $\qquad\square$

*Proof of Corollary F.3.* The proof of both the inequalities follows from a direct application of Lemma F.1, after first rewriting the summation over balls in $\mathcal{P}_h^k$ as a summation over active balls in $\mathcal{P}_h^K$.

**First Inequality**: First, observe that we can write

$$\sum_{k=1}^K \frac{2^{\beta \ell(B_h^k)}}{\left(n_h^k(B_h^k)\right)^\alpha} = \sum_{\ell \in \mathbb{N}_0} \sum_{B : \ell(B) = \ell} \sum_{k=1}^K \mathbb{1}_{[B_h^k = B]} \frac{2^{\beta \ell(B)}}{\left(n_h^k(B)\right)^\alpha}$$

Now, in order to use Lemma F.1, we first need to rewrite the summation as over 'active balls' in the terminal partition $\mathcal{P}_h^K$ (i.e., balls which are yet to be split). Expanding the above, we get

$$\sum_{k=1}^{K} \frac{2^{\beta\ell(B_h^k)}}{\left(n_h^k(B_h^k)\right)^\alpha} = \sum_{\ell\in\mathbb{N}_0} \sum_{B\in\mathcal{P}_h^K:\ell(B)=\ell} \sum_{B'\supseteq B} 2^{d(\ell(B')-\ell(B))} \sum_{k=1}^{K} \mathbb{1}_{\left[B_h^k=B'\right]} \frac{2^{\beta\ell(B')}}{\left(n_h^k(B')\right)^\alpha}$$

$$\leq \sum_{\ell\in\mathbb{N}_0} \sum_{B\in\mathcal{P}_h^K:\ell(B)=\ell} \sum_{B'\supseteq B} 2^{d(\ell(B')-\ell(B))} 2^{\beta\ell(B')} \sum_{j=1}^{n_+(\ell(B'))} \frac{1}{j^\alpha}$$

$$\leq \frac{\phi^{1-\alpha}}{1-\alpha} \sum_{\ell\in\mathbb{N}_0} \sum_{B\in\mathcal{P}_h^K:\ell(B)=\ell} \sum_{B'\supseteq B} 2^{d(\ell(B')-\ell(B))} 2^{\beta\ell(B')} 2^{\gamma\ell(B')(1-\alpha)}.$$

where we used the fact that once a ball has been partitioned it is no longer chosen by the algorithm and an integral approximation to the sum of $1/j^\alpha$ for $\alpha\leq 1$. Next, we plug in the levels to get

$$\sum_{k=1}^{K} \frac{2^{\beta\ell(B_h^k)}}{\left(n_h^k(B_h^k)\right)^\alpha} \leq \frac{\phi^{1-\alpha}}{1-\alpha} \sum_{\ell\in\mathbb{N}_0} \sum_{B\in\mathcal{P}_h^K:\ell(B)=\ell} \sum_{j=0}^{\ell} 2^{d(j-\ell)} 2^{\beta j} 2^{\gamma j(1-\alpha)}$$

$$= \frac{\phi^{1-\alpha}}{1-\alpha} \sum_{\ell\in\mathbb{N}_0} \sum_{B\in\mathcal{P}_h^K:\ell(B)=\ell} \frac{1}{2^{d\ell}} \sum_{j=0}^{\ell} 2^{j(d+\beta+\gamma(1-\alpha))}$$

$$\leq \frac{\phi^{1-\alpha}}{(2^{d+\beta+\gamma(1-\alpha)}-1)(1-\alpha)} \sum_{\ell\in\mathbb{N}_0} \sum_{B\in\mathcal{P}_h^K:\ell(B)=\ell} \frac{1}{2^{d\ell}} 2^{(\ell+1)(d+\beta+\gamma(1-\alpha))}$$

$$\leq \frac{2\phi^{1-\alpha}}{(1-\alpha)} \sum_{\ell\in\mathbb{N}_0} \sum_{B\in\mathcal{P}_h^K:\ell(B)=\ell} 2^{\ell(\beta+\gamma(1-\alpha))}.$$

We set $a_\ell = 2^{\ell(\beta+\gamma(1-\alpha))}$. Clearly we have that $a_\ell$ are increasing with respect to $\ell$. Moreover,

$$\frac{2a_{\ell+1}}{a_\ell} = \frac{2\cdot 2^{(\ell+1)(\beta+\gamma(1-\alpha))}}{2^{(\ell)(\beta+\gamma(1-\alpha))}} = 2^{1+\beta+\gamma(1-\alpha)}.$$

Setting this quantity to be less than $n_+(\ell)/n_+(\ell-1) = 2^\gamma$ we require that

$$2^{1+\beta+\gamma(1-\alpha)} \leq 2^\gamma \Leftrightarrow 1+\beta-\alpha\gamma \leq 0$$

Now we can apply Lemma F.1 to get that

$$\sum_{k=1}^{K} \frac{2^{\beta\ell(B_h^k)}}{\left(n_h^k(B_h^k)\right)^\alpha} \leq \frac{2\phi^{1-\alpha}}{(1-\alpha)} 2^{d\ell^\star} a_{\ell^\star}$$

$$= \frac{2^{2(d+\beta+\gamma(1-\alpha))}\phi^{1-\alpha}}{(1-\alpha)} \left(\frac{K}{\phi}\right)^{\frac{d+\beta+\gamma(1-\alpha)}{d+\gamma}}$$

$$= O\left(\phi^{\frac{-(d\alpha+\beta)}{d+\gamma}} K^{\frac{d+(1-\alpha)\gamma+\beta}{d+\gamma}}\right).$$

**Second Inequality**: As in the previous part, we can rewrite as the summation we have

$$\sum_{k=1}^{K} \frac{\ell(B_h^k)^\beta}{\left(n_h^k(B_h^k)\right)^\alpha} = \sum_{\ell\in\mathbb{N}_0} \sum_{B:\ell(B)=\ell} \sum_{k=1}^{K} \mathbb{1}_{\left[B_h^k=B\right]} \frac{\ell(B)^\beta}{\left(n_h^k(B)\right)^\alpha}.$$

$$\leq \sum_{\ell\in\mathbb{N}_0} \sum_{B\in\mathcal{P}_h^K:\ell(B)=\ell} \sum_{B'\supseteq B} 2^{d(\ell(B')-\ell(B))} \ell(B')^\beta \sum_{j=1}^{n_+(\ell(B'))} \frac{1}{j^\alpha}$$

$$\leq \sum_{\ell\in\mathbb{N}_0} \sum_{B\in\mathcal{P}_h^K:\ell(B)=\ell} \sum_{B'\supseteq B} 2^{d(\ell(B')-\ell(B))} \ell(B')^\beta \frac{n_+(\ell(B'))^{1-\alpha}}{1-\alpha}$$

$$= \frac{\phi^{1-\alpha}}{1-\alpha} \sum_{\ell \in \mathbb{N}_0} \sum_{B \in \mathcal{P}_h^K : \ell(B) = \ell} \sum_{B' \supseteq B} 2^{d(\ell(B') - \ell(B))} \ell(B')^\beta 2^{\ell(B')\gamma(1-\alpha)}$$

As before, we plug in the levels to get

$$\sum_{k=1}^K \frac{\ell(B_h^k)^\beta}{\left(n_h^k(B_h^k)\right)^\alpha} = \frac{\phi^{1-\alpha}}{1-\alpha} \sum_{\ell \in \mathbb{N}_0} \sum_{B \in \mathcal{P}_h^K : \ell(B) = \ell} \sum_{j=0}^\ell 2^{d(j-\ell)} j^\beta 2^{j\gamma(1-\alpha)}$$

$$\leq \frac{\phi^{1-\alpha}}{1-\alpha} \sum_{\ell \in \mathbb{N}_0} \sum_{B \in \mathcal{P}_h^K : \ell(B) = \ell} \frac{\ell^\beta}{2^{d\ell}} \sum_{j=0}^\ell 2^{j(d+\gamma(1-\alpha))}$$

$$\leq \frac{2\phi^{1-\alpha}}{(1-\alpha)} \sum_{\ell \in \mathbb{N}_0} \sum_{B \in \mathcal{P}_h^K : \ell(B) = \ell} \ell^\beta 2^{\ell\gamma(1-\alpha)}.$$

We take the term $a_\ell = \ell^\beta 2^{\ell\gamma(1-\alpha)}$. Clearly we have that $a_\ell$ are increasing with respect to $\ell$. Moreover,

$$\frac{2a_{\ell+1}}{a_\ell} = \left(1 + \frac{1}{\ell}\right)^\beta 2^{1+\gamma(1-\alpha)}.$$

We require that this term is less than $n_+(\ell+1)/n_+(\ell) = 2^\gamma$ for all $\ell \geq \ell^\star$ (see note after Lemma F.1). This yields the following sufficient condition (after dividing through by $2^\gamma$)

$$\left(1 + \frac{1}{\ell}\right)^\beta 2^{1-\alpha\gamma} \leq 1 \,\forall \ell \geq \ell^\star$$

or equivalently, $\alpha\gamma - \beta \log_2(1 + 1/\ell^\star) \geq 1$. Finally note that $\log_2(1+x) \leq x/\ln 2 \leq x$ for all $x \in [0,1]$. Thus, we get that a sufficient condition is that $\alpha\gamma - \beta/\ell^\star \geq 1$. Assuming this holds, we get by Lemma F.1 that

$$\sum_{k=1}^K \frac{\ell(B_h^k)^\beta}{\left(n_h^k(B_h^k)\right)^\alpha} \leq \left(\frac{2\phi^{1-\alpha}}{(1-\alpha)}\right) 2^{d\ell^\star} a_{\ell^\star}$$

$$= \left(\frac{2\phi^{1-\alpha}}{1-\alpha}\right) 4^{d+\gamma(1-\alpha)} \left(\frac{K}{\phi}\right)^{\frac{d+\gamma(1-\alpha)}{d+\gamma}} \left(\frac{\log_2(K/\phi)}{d+\gamma} + 2\right)^\beta$$

$$= O\left(\phi^{\frac{-d\alpha}{d+\gamma}} K^{\frac{d+(1-\alpha)\gamma}{d+\gamma}} (\log_2 K)^\beta\right).$$

$\square$

*Proof of Lemma E.1.* We use the notation $B_{h'}^k$ to denote the active ball containing the point $(X_{h'}^k, A_{h'}^k)$. Under this we have by the update rule on $\widetilde{\mathbf{V}}_{h'}^k$ that for any $h' \geq h$

$$\mathbb{E}^{k-1}\left[\widetilde{\mathbf{V}}_{h'}^k(\mathcal{S}(\mathcal{P}_{h'}^k, X_{h'}^k)) \mid X_h^k\right] \leq \mathbb{E}^{k-1}\left[\overline{\mathbf{Q}}_{h'}^k(B_{h'}^k) \mid X_h^k\right]$$

$$= \mathbb{E}^{k-1}\left[\overline{\mathbf{r}}_{h'}^k(B_{h'}^k) + \mathbb{E}_{x \sim \overline{\mathbf{T}}_{h'}^k(\cdot|B)}[\overline{\mathbf{V}}_{h'+1}^{k-1}(x)] + \text{RUCB}_{h'}^k(B_{h'}^k) + \text{TUCB}_{h'}^k(B_{h'}^k) \mid X_h^k\right] \text{ (via update rule for } \overline{\mathbf{Q}}_h^k)$$

$$= \mathbb{E}^{k-1}\left[\overline{\mathbf{r}}_{h'}^k(B_{h'}^k) - r_{h'}(X_{h'}^k, A_{h'}^k) + \text{RUCB}_{h'}^k(B_{h'}^k) \mid X_h^k\right] + \mathbb{E}^{k-1}\left[r_{h'}(X_{h'}^k, A_{h'}^k) \mid X_h^k\right]$$

$$+ \mathbb{E}^{k-1}\left[\mathbb{E}_{x \sim \overline{\mathbf{T}}_{h'}^k(\cdot|B_{h'}^k)}[\overline{\mathbf{V}}_{h'+1}^{k-1}(x)] - \mathbb{E}_{x \sim T_h(\cdot|X_{h'}^k, A_{h'}^k)}[\overline{\mathbf{V}}_{h'+1}^{k-1}(x)] + \text{TUCB}_{h'}^k(B_{h'}^k) \mid X_h^k\right]$$

$$+ \mathbb{E}^{k-1}\left[\mathbb{E}_{x \sim T_{h'}(\cdot|X_{h'}^k, A_{h'}^k)}[\overline{\mathbf{V}}_{h'+1}^{k-1}(x)] \mid X_h^k\right]$$

$$= \mathbb{E}^{k-1}\left[\overline{\mathbf{r}}_{h'}^k(B_{h'}^k) - r_h(X_{h'}^k, A_{h'}^k) + \text{RUCB}_{h'}^k(B_{h'}^k) \mid X_h^k\right] + \mathbb{E}^{k-1}\left[r_h(X_{h'}^k, A_{h'}^k) \mid X_h^k\right]$$

$$+ \mathbb{E}^{k-1}\left[\mathbb{E}_{x \sim \overline{\mathbf{T}}_{h'}^k(\cdot|B_{h'}^k)}[\overline{\mathbf{V}}_{h'+1}^{k-1}(x)] - \mathbb{E}_{x \sim T_h(\cdot|X_{h'}^k, A_{h'}^k)}[\overline{\mathbf{V}}_{h'+1}^{k-1}(x)] + \text{TUCB}_{h'}^k(B_{h'}^k) \mid X_h^k\right]$$

$$+ \mathbb{E}^{k-1}\left[\overline{\mathbf{V}}_{h'+1}^{k-1}(X_{h'+1}^k) \mid X_h^k\right] \qquad (\text{ as } X_{h'+1}^k \sim T_h(\cdot \mid X_{h'}^k, A_{h'}^k))$$

$$\leq \mathbb{E}^{k-1}\left[\overline{\mathbf{r}}_{h'}^k(B_{h'}^k) - r_h(X_{h'}^k, A_{h'}^k) + \text{RUCB}_{h'}^k(B_{h'}^k) \mid X_h^k\right] + \mathbb{E}^{k-1}\left[r_h(X_{h'}^k, A_{h'}^k) \mid X_h^k\right]$$

$$+ \mathbb{E}^{k-1}\left[\mathbb{E}_{x \sim \overline{\mathbf{T}}_{h'}^k(\cdot | B_{h'}^k)}[\overline{\mathbf{V}}_{h'+1}^{k-1}(x)] - \mathbb{E}_{x \sim T_{h'}(\cdot | X_{h'}^k, a_{h'}^k)}[\overline{\mathbf{V}}_{h'+1}^{k-1}(x)] + \text{TUCB}_{h'}^k(B_{h'}^k) \mid X_h^k\right]$$

$$+ \mathbb{E}^{k-1}\left[\widetilde{\mathbf{V}}_{h'+1}^{k-1}(\mathcal{S}(\mathcal{P}_{h'+1}^{k-1}, X_{h'+1}^k)) + L_V \mathcal{D}(B_{h'+1}^k) \mid X_h^k\right] \qquad \text{(via update rule for } \overline{\mathbf{V}}_h^k)$$

Taking this inequality and summing from $h' = h$ up until $H$ we find that $\sum_{h'=h}^H \mathbb{E}^{k-1}\left[r_{h'}(X_{h'}^k, A_{h'}^k) \mid X_h^k\right] = V_h^{\pi^k}(X_h^k)$. Moreover, by changing the index in the sum and using the fact that $V_{H+1} = 0$, it follows that

$$\sum_{h'=h}^H \mathbb{E}^{k-1}\left[\widetilde{\mathbf{V}}_{h'+1}^{k-1}(\mathcal{S}(\mathcal{P}_{h'+1}^{k-1}, X_{h'+1}^k)) \mid X_h^k\right] = \sum_{h'=h}^H \mathbb{E}^{k-1}\left[\widetilde{\mathbf{V}}_{h'}^{k-1}(\mathcal{S}(\mathcal{P}_{h'}^{k-1}, X_{h'}^k)) \mid X_h^k\right] - \widetilde{\mathbf{V}}_h^{k-1}(\mathcal{S}(\mathcal{P}_h^{k-1}, X_h^k)).$$

Rearranging the inequalities gives the desired results. $\qquad\square$

*Proof of Lemma E.2.* We condition on the good events from Lemmas D.2 and D.3 by taking $\delta = 1/HK$. Using the definition of regret and the law of total expectation we have that:

$$\mathbb{E}[R(K)] = \mathbb{E}\left[\sum_{k=1}^K V_1^\star(X_1^k) - V_1^{\pi^k}(X_1^k)\right]$$

$$\lesssim \mathbb{E}\left[\sum_{k=1}^K \overline{\mathbf{V}}_1^{k-1}(X_1^k) - V_1^{\pi^k}(X_1^k)\right] \qquad \text{(via the optimism principle, Lemma D.4)}$$

$$\lesssim \mathbb{E}\left[\sum_{k=1}^K \widetilde{\mathbf{V}}_1^{k-1}(\mathcal{S}(\mathcal{P}_1^{k-1}, X_1^k)) - V_1^{\pi^k}(X_1^k) + L_V \mathcal{D}_\mathcal{S}(\mathcal{S}(\mathcal{P}_1^{k-1}, X_1^k))\right] \quad \text{(update rule for } \widetilde{\mathbf{V}}_h^k)$$

$$\lesssim \mathbb{E}\left[\sum_{k=1}^K \widetilde{\mathbf{V}}_1^{k-1}(\mathcal{S}(\mathcal{P}_1^{k-1}, X_1^k)) - V_1^{\pi^k}(X_1^k) + L_V \mathcal{D}(B_1^k)\right]$$

Next, define $\mathbb{E}^{k-1}[\cdot] \triangleq \mathbb{E}[\cdot \mid \mathcal{F}_{k-1}]$. Now using Lemma E.1, and the tower rule for conditional expectations, we get

$$\mathbb{E}[R(K)] \lesssim \mathbb{E}\left[\sum_{k=1}^K \mathbb{E}^{k-1}\left[\widetilde{\mathbf{V}}_1^{k-1}(\mathcal{S}(\mathcal{P}_1^{k-1}, X_1^k)) - V_1^{\pi^k}(X_1^k)\right] + L_V \mathcal{D}(B_1^k)\right]$$

$$\lesssim \sum_{k=1}^K \sum_{h=1}^H \mathbb{E}\left[\mathbb{E}^{k-1}\left[\widetilde{\mathbf{V}}_h^{k-1}(\mathcal{S}(\mathcal{P}_h^{k-1}, X_h^k)) - \widetilde{\mathbf{V}}_h^k(\mathcal{S}(\mathcal{P}_h^k, X_h^k))\right]\right]$$

$$+ \sum_{h=1}^H \sum_{k=1}^K \mathbb{E}\left[\mathbb{E}^{k-1}\left[\overline{\mathbf{r}}_h^k(B_h^k) - r_h(X_h^k, A_h^k) + \text{RUCB}_h^k(B_h^k)\right]\right]$$

$$+ \sum_{h=1}^H \sum_{k=1}^K \mathbb{E}\left[\mathbb{E}^{k-1}\left[\mathbb{E}_{x \sim \overline{\mathbf{T}}_h^k(\cdot | B_h^k)}[\overline{\mathbf{V}}_{h+1}^{k-1}(x)] - \mathbb{E}_{x \sim T_h(\cdot | X_h^k, A_h^k)}[\overline{\mathbf{V}}_{h+1}^{k-1}(x)] + \text{TUCB}_h^k(B_h^k)\right]\right]$$

$$+ \sum_{k=1}^K \sum_{h=1}^H L_V \mathbb{E}\left[\mathcal{D}(B_h^k)\right]$$

$$= \sum_{k=1}^K \sum_{h=1}^H \mathbb{E}\left[\widetilde{\mathbf{V}}_h^{k-1}(\mathcal{S}(\mathcal{P}_h^{k-1}, X_h^k)) - \widetilde{\mathbf{V}}_h^k(\mathcal{S}(\mathcal{P}_h^k, X_h^k))\right]$$

$$+ \sum_{h=1}^H \sum_{k=1}^K \mathbb{E}\left[2\text{RUCB}_h^k(B_h^k)\right] + \sum_{h=1}^H \sum_{k=1}^K \mathbb{E}\left[2\text{TUCB}_h^k(B_h^k)\right] + \sum_{k=1}^K \sum_{h=1}^H L_V \mathbb{E}\left[\mathcal{D}(B_h^k)\right]$$

where in the last line we used the definition of the good event. $\qquad\square$