[Reviews · NeurIPS 2020]

Review 1

Summary and Contributions: The paper introduces a technique for partitioning a large state space into a set of discrete bins in the context of model-based reinforcement learning. The partitioning is done adaptively, and the paper provides theoretical guarantees which are comparable to those of existing methods. They highlight that on the practical implementation side, their approach is computationally more efficient, and empirically performs well compared to their fixed-discretization counterparts.

Strengths: Much of the paper's strengths lie in its theoretical grounding, where they thoroughly characterized and contextualized their method in terms of its worst case regret, and computational complexities. While adaptive discretization of state spaces has been done before in reinforcement learning settings, the novelty lies in the extension to the model-based setting. It further emphasizes its focus on methods for computation-limited applications, that it is likely to be of interest to those looking to apply reinforcement learning on small devices.

Weaknesses: The main weakness is in the strength of the empirical results, as it seems that the adaptive discretization had no benefit over using a fixed discretization in the model-based case (apart from using fewer partitions of the space). It's possible that it might just be the case for these environments, e.g. the environments might be too simple/easy, but only two environments were tested. The number of environments is alright given the theoretical focus of the paper, but could the authors comment on whether it's a representative sample of environments for assessing each method's performance? Are there any possible explanations for why the two model-based methods performed similarly in the two problems despite the big difference in the number of state partitions?

Correctness: The experimental details in the appendix suggest that the plots are supposed to contain confidence intervals. I apologize if I had simply missed them, but they seem to be missing from the plots. Could the authors confirm whether these statistics were computed, and comment on the significance of the results? Apart from this, the empirical methodology appears sensible, e.g., a reasonable number of runs, hyperparameter tuning via cross-validation.

Clarity: The paper is very well written.

Relation to Prior Work: Thorough summary of related work, and how this work fits into their contexts. A minor note about the connection to CMACs, there has been conceptually similar work on adapting the discretizations of CMACs (rebranded as tile coding), e.g., Pyeatt et al. (2001) and Whiteson et al. (2006). They similarly represent the partitioning as a (decision) tree, but focused more on the model-free setting with more heuristically motivated splitting rules. While the prior work was less theoretical, it might still be interesting to compare the intutions behind the methods, or their empirical performances.

Reproducibility: Yes

Additional Feedback: My questions for the authors can be found in the Weaknesses and Correctness sections, and am willing to adjust my score if they are addressed. ----- I have read the rebuttal and the other reviews. Thank you for the clarifications regarding the statistical significance testing. The intervals being less than a line width was surprising from how oscillatory the lines were, suggesting that I was misunderstanding "25 iterations" to mean results across 25 random seeds, when it seems to instead be 1 run evaluated at 25 different time points. This is minor given the paper's theoretical focus, but I think it would paint a clearer picture of how one might expect the algorithm to perform if the former was reported. Much of my concern with the empirical evaluation was that as presented, the results did not seem to support the rest of the paper. I believe this was largely due to a lack of discussion around experimental design choices made, and around potentially surprising results. This was clarified a bit in the rebuttal, and I encourage the authors to incorporate more of such discussion in their paper. I acknowledge that the primary contributions are theoretical, and why my score still leaned toward acceptance in light of the above. I've raised my score, but I believe that rounding out some of these empirical concerns to better support the theoretical results or shed additional insight beyond the theory would really strengthen the paper and make the difference for a stronger accept!


Review 2

Summary and Contributions: This paper introduces an algorithm for adaptive discretization of continuous state-action spaces for model-based reinforcement learning. The paper also provides worst-case regret bounds for the algorithm.

Strengths: - This paper clear theoretical analysis on the worst-case regret bounds of the introduced algorithm. - The experiments show that AdaMB performs similarly to epsilonMB with less storage and computation requirements. This could be important in applications with strict resource limitations.

Weaknesses: The experiments are performed on two simple environments: 1) an oil discovery problem which involves the agent minimizing costs in an unknown noisy 1D cost function, and 2) an ambulance relocation problem which involves controlling a fleet of k ambulances, to minimize costs (movement and time) to respond to incoming requests. While AdaMB performs similarly to epsilonMB with less resources in these problems, it is not clear if the proposed method would scale to more complex problems with larger state-action spaces. The proposed method AdaMB does not seem to perform any better than AdaQL. Both perform similarly and maintain a similar number of regions on average. The paper lacks a thorough comparison of AdaMB and AdaQL and it is not clear if AdaMB is better than AdaQL in any aspect.

Correctness: Yes, the empirical methodology is similar to (Sinclar et al., 2019).

Clarity: The motivation for the work and how it relates to prior works is not clear.

Relation to Prior Work: It is not clear how the adaptive discretization algorithm used in this paper is different from that of AdaQL (Sinclair et al., 2019).

Reproducibility: Yes

Additional Feedback: - How do the storage and computation requirements of AdaMB compare to AdaQL? Isn't AdaQL lighter since it does not maintain a transition model? - What are the advantages of using adaptive discretization in a tabular setting as opposed to using a function approximator? Update after author response: I acknowledge that I have read the rebuttal. Based on other reviews and the author response, I have updated my score above the acceptance threshold.


Review 3

Summary and Contributions: The papers presents a method of state space simplification that adaptively discretizes a large (or continuous) state space, with a regret that scales favorably (compared to baseline methods) w.r.t. the time horizon H and number of episodes K.

Strengths: The main strengths was the clarity or writing, and the equations and algorithm which described the method unambiguously, in addition to the regret bound this algorithm achieves.

Weaknesses: The three main weaknesses I found was: 1) Lack of strong motivation for why adaptive discretization is needed in the first place? Not all RL algorithms need assume a discrete state space, and perform quite well in continuous spaces, so I do not know why there would be a need to "simplify" such continuous spaces in the first place. 2) This paper could have been put into greater context w.r.t. related literature, especially all the literature on this topic that's older than the last couples years. See the "Relation to prior work:" section below. 3) The experiments appears to show no difference between the methods' performances (except for the red baseline)?

Correctness: As far as I know.

Clarity: Yes, the writing is very clear.

Relation to Prior Work: Much prior work is discussed, but could be broken down more (opposed to long lists of citations). However, I wonder if more can be said w.r.t. what other adaptive partitionings that could be used beyond square/cube partitions, like Voronoi cells? [1] Or adaptive soft state aggregations [2] Or basis functions + linear function approximation: - neighbourhood component analysis [3] - radial basis functions [4] I realize parametric algorithms were mentioned on line 80, but those references are very recent. It might be worth citing some more original works too, such as [3], [4], and/or others. [1] Lee, I. S. and Lau, H. Y. (2004). Probabilistic State Representations Adaptive state space partitioning for reinforcement learning. Engineering applications of artificial intelligence, 17(6):577–588. [2] Singh, S. P., Jaakkola, T., and Jordan, M. I. (1995). Reinforcement learning with soft state aggregation. NeurIPS, pages 361–368. [3] Keller, P. W., Mannor, S., and Precup, D. (2006). Automatic basis function construction for approximate dynamic programming and reinforcement learning. ICML, pages 449–456. ACM. [4] Menache, I., Mannor, S., and Shimkin, N. (2005). Basis function adaptation in temporal difference reinforcement learning. Annals of Operations Research, 134(1):215–238.

Reproducibility: Yes

Additional Feedback: medium points: table 1: the "Lower Bounds" method doesn't have "Time complexity" or "Space complexity" entries? Also why is it separated from the other prior work? line 50: explain "product metric"? line 51: comment on when Lipschitz assumptions might be broken? Is this assuming something the others baselines in Table 1 aren't? If so, is this a fair comparison then? line 59: mentions "our bounds are uniformly better (in terms of dependence on K and H", but Table 1 shows AdaMB has Regret proportional to H^{>1} while some baselines are H^{1}? line 64-69: while interesting, is the relevant to the paper or proposed AdaMB algorithm? line 180: "if needed", please elaborate? line 279: did they? Everything except red (epsilonQL) seems to perform the same. small points: line 74: maybe divide up this string of 12 citations into meaningful sub-categories? Seeing twelve bunched together doesn't help the reader understand the landscape. Or reduce. Same thing for line 80. line 74: rather than using "[5]" as a noun, mentioning the name of the method would be more helpful. line 117: should the distribution "R_h" be indexed by "h", or its inputs and samples? Same thing with "T_h" line 118. line 112: maybe "\sim" instead of "=" if "A" is a random sample ? figure 1 caption: "{B_1, B_23, B_24}" --> "{B_4, B_23, B_24}" ? (B_1 is not blue) line 360: "pac" --> "PAC" line 383, 396: "q-learning" --> "Q-learning" (i.e. use {ABC} to capitalize names and acronyms in bibtex) ====== After author response ====== I thank the author's for their rebuttal, but this did not address my three concerns, so I keep my score the same.


Review 4

Summary and Contributions: The paper studies model-based RL in continuous domains. It introduces and evaluates an adaptive discretization technique which focuses on high-reward arras of the joint state-action space. It proves worst-case regret bounds for the method. I have read the author feedback and discussions.

Strengths: Well written, well-motivated by clear examples. The work is quite far outside my expertise but it seems to be a carefully conducted piece of research and valuable contribution. It is intuitive that adaptive discretization should have advantages over "blind" discretization with the same granularity everywhere.

Weaknesses: The experiments in Fig. 2 show superiority over simple uniform "epsilon-nets", and comparable performance to other methods tested. I did not understand very well whether this is also a positive result or not, i.e. did they achieve similar results with less resource use? That seems to be claimed in general terms, but what about the specific experiment? This is important enough to be in the main paper.

Correctness: I am far from an expert, but the setting was clear and the method/approach seemed correct an the test cases seemed reasonable. There is only a proof sketch in the main paper.

Clarity: Yes, I found the paper very well written overall. I did read "lightly" over some of the more technical developments in Sections 3 and 4 for which I lack the background knowledge.

Relation to Prior Work: Yes, it is very clearly discussed.

Reproducibility: Yes

Additional Feedback: I wondered how these techniques are related to algorithms for function optimization such as Munos' SOO, DOO, HOO.


Review 5

Summary and Contributions: The paper proposed and studied a new MBRL algorithm for episodic finite horizon continuous state-action space MDP. The proposed algorithm combines a classic discretization technique (adaptive discretization) and a classic regret minimization MBRL algorithm (UCBVI). Theoretical results showed that the adaptive discretization technique achieves lower regret bound compared to fixed discretization (the other classic discretization technique) when these two discretization techniques are combined with the same MBRL algorithm (UCBVI). Empirical results on two low dimensional (1 or 2) problems provided positive evidence for the theoretical claims.

Strengths: The problem being studied is an interesting open question. The proposed algorithm has a lower regret bound, compared to existing methods. Theoretical results are sound.

Weaknesses: The key disadvantage is that the proposed algorithm has unbounded monotonically increasing space complexity as the number of simulated episodes increases. Therefore it is hard to me to imagine using the proposed algorithm in practical large scale problems where typically new experiences are generated continuously. The reason for this unbounded monotonically increasing space complexity is that the algorithm doesn't have a mechanism to combine small balls (elements of the partition) to produce a large ball, instead, it only has mechanism to split a large ball into several smaller balls. Empirical result in the appendix also seems to suggest this point. Empirical results are limited in several aspects: 1) only domains with 1 and 2-dimensional space are used 2) the horizon of the algorithm being applied is short (only 5), 3) no faster convergence compared to UCBVI with fixed discretization is shown.

Correctness: I agree with most of the claims. But I don't agree with that the algorithm is a practical RL algorithm with low space complexity, as claimed in the paper.

Clarity: I would give 5 out of 10 for writing. There are many issues. Here are several examples: There is no conclusion section. No limitation of the proposed algorithm is discussed. line 179 \tilde a is not defined line 164 I don't understand why the complex space is used here. line 159 I guess max should be sup line 155 "partition" is not rigorously defined or at least is confusing. line 153 I don't know why it uses bold symbol + bar to denote estimates. line 60 The cited paper [21] is not in table 1. line 26 Not all model-free algorithms are action-value based and use upper-confidence bounds.

Relation to Prior Work: Yes

Reproducibility: Yes

Additional Feedback:

[Author Response · NeurIPS 2020]

**Overall Comments to Program Chair and Reviewers**: All the reviewers have praised the paper for the theoretical contributions, and have questions primarily about the empirics. We address these below in detail, but first want to emphasize that *our simulations were chosen not to compare our algorithm to others, but rather, to illustrate our theory* in 2 ways: (i) the 'small' examples (2-dimensional, small $H$) let us control the $Q$-function and exactly show how ADAMB 'zooms' in to regions of higher $Q$-values, and (ii) the examples show that even in small settings, ADAMB gets optimal regret with *much lower storage and time complexity* than naive discretization.

We do provide more simulations in the appendix (which answer some of the reviewer questions), and we have code implementing ADAMB on a larger problem with a 10-dimensional joint state and action space showing similar insights that we can include. While comparing with other heuristics is clearly important, doing this in a fair empirical way requires extensive engineering and detailed simulations, and also we feel may distract from our main contributions:

• We give the *best available* regret guarantees for model-based RL with continuous state-action spaces and Lipschitz $Q^\star$ functions, and the first with adaptive function approximation, improving on [21,11] with a much simpler algorithm.
• New technique for transition-kernel concentration which bypasses more complex covering arguments over function spaces (see [11, 41, 18]), and a new LP duality argument for adaptive discretization (simplifying [35,36]).
• We provide storage, time, and regret guarantees for our algorithm. Our regret bound is novel vs. prior work with discretization, and as far as we know, we get the *first sublinear dependence on storage and time complexity* as well.

**Adaptive vs Fixed Discretization**: (**R1, R2, R4**) Empirically, we do show settings where ADAMB performs better than fixed discretization (see Fig 7 in appendix); this gap can be made larger when $Q^\star$ is more peaked (as suggested by our theory), and we can include these in our camera-ready. We also choose the optimal fixed discretization in our experiments, based on knowing the $Q^\star$, while ADAMB has minimal tuning, and needs much less storage. (**R5**) Finally, our theory explains why adaptive and fixed discretization get the same minimax regret bounds (see Corollary D2).

**Adaptive Discretization versus Function Approximation** (**R3, R5**): Adaptive discretization is a *form of function approximation* (using step-functions as a kernel) – in particular, it is a natural basis for Lipschitz functions. While other function approximation techniques are clearly important in practice, our approach has the following benefits:

• Standard algorithms based on function approximators (linear/polynomial/RBF kernel) use a fixed number of parameters (degree, bandwidth), and have guarantees that depend on some parametric model structure (i.e., how well these functions approximate the $Q$-function). Our guarantees (also [11,21]) hold uniformly for the larger class of Lipschitz functions, which is suited to many more applications where special parametric model structures may not hold.
• In contrast to standard function approximation techniques (including [11,21]), *our algorithm adaptively chooses number of parameters based on data*. Also, by using a simpler basis (step functions) we get lower storage, runtime.
• Overparametrized models like deep-RL enjoy success in experiments, but require *much* larger storage and training; comparing to ADAMB under resource constraints (i.e., on-policy settings) would be unfair.

**Model-Based vs. Model-Free Algorithms**: (**R2, R4**) Comparing model-based and model-free algorithms is an important open problem in RL. Folklore says model-based methods outperform model-free ones in practice – however, these comparisons involve very different styles of algorithms. By giving a model-based equivalent for a state-of-the-art model-free method (using adaptive discretization in continuous metric spaces) we can make a direct comparison. Surprisingly, our bounds show ADAMB has *worse* theoretical storage, time, and regret compared to ADAQL from [35] (Table 1). Moreover, simulations show that even in practice, ADAMB *and* ADAQL *have similar performance*, which is a sharp contrast to $\epsilon$-net algorithms (where model-based is much better as folklore suggests).

Our results suggest that $Q$-learning with adaptive discretization is good for resource-constrained RL in continuous settings, as they learn an efficient discretization of the space subject to the constraints. Moreover, in larger dimensional settings (ambulance problem with $\geq 5$ ambulances), ADAMB does better than ADAQL, but also maintains a much larger partition and so needs higher storage. We can include these experiments in the camera ready.

**Experimental Results** (**R1, R2, R5**): We chose our instances as *proofs of concept* to illustrate the theory (how the algorithm intrinsically partitions the regions across near-optimal parts of the space, in comparison to a fixed discretization) – in particular, using a 2-dimensional problem lets us plot the $Q$ value-estimates and true $Q^\star$ values. Making the algorithm scale to larger problems is an interesting future direction outside the scope of this paper.

(**R5**) The point that ADAMB's space complexity is monotonically increasing is valid; however, to get sublinear minimax regret in a continuous setting for nonparametric Lipschitz models, the model complexity must grow over episodes. In practice, one would run our algorithm until we run out of space – our experiments show that ADAMB uses resources (storage and computation) much better than a uniform discretization (see Figures 3 and 4 in appendix). We are not aware of any storage-performance lower bounds, so that is also an interesting future direction.

(**R1**) We omitted confidence intervals as they were so small that they were not visible given the figure size. In the camera ready, we will include bigger plots with CIs, and more fine-grained comparisons of the various algorithms.

[Meta-Review · NeurIPS 2020]

The work has clear positives: + paper presents a novel algorithm that achieves low regret + novel consideration of adaptive discretization for model-based RL (prior work focuses on the model-free case) + important practical focus on computational resources + novel theory However, there are significant issues as well: - the experiments are not were presented or discussed within the context of the rest of the paper: they appear to contradict the main messages of the paper - there is confusion over how the experiments were implemented and evaluated (e.g., proper averaging over independent runs, fair treatment of hyperparameters etc). See R1 for more details - the proposed algorithm exhibits space complexity that monotonically increases; the authors suggested to just cap it - poor discussion of model-based RL with function approximation (linear dyna, recent deep learning approaches etc) - related no clear argument why we would explore adaptive discretization approaches compared to other approaches. The paper is doing something different than the majority of the community---that can be good but it should be directly addressed Summary of the discussion. The reviewers thought the experiments considerably weakened the paper, and it would be best if they were removed from the paper. The strongest advocate of the paper had low confidence and not much to say much during discussion. The biggest critic also had low confidence. The most knowledgable reviewer R5 landed at weak accept concluding the theory is interesting on its own, but the paper needs significant revision to deal with the experiment issue. The meta reviewer agrees. The authors seem to agree that the paper could be much better if more strongly pitched towards the theory side.